# Depthwise Hyperparameter Transfer in Residual Networks: Dynamics and Scaling Limit

**Blake Bordelon**[* ¶], **Lorenzo Noci**[* ♯], **Mufan (Bill) Li**[§], **Boris Hanin**[† §] **& Cengiz Pehlevan**[† ¶]

¶ Harvard University
♯ ETH Zürich
§ Princeton University

## Abstract

The cost of hyperparameter tuning in deep learning has been rising with model sizes, prompting practitioners to find new tuning methods using a proxy of smaller networks. One such proposal uses $\mu$P parameterized networks, where the optimal hyperparameters for small width networks *transfer* to networks with arbitrarily large width. However, in this scheme, hyperparameters do not transfer across depths. As a remedy, we study residual networks with a residual branch scale of $1/\sqrt{\text{depth}}$ in combination with the $\mu$P parameterization. We provide experiments demonstrating that residual architectures including convolutional ResNets and Vision Transformers trained with this parameterization exhibit transfer of optimal hyperparameters across width and depth on CIFAR-10 and ImageNet. Furthermore, our empirical findings are supported and motivated by theory. Using recent developments in the dynamical mean field theory (DMFT) description of neural network learning dynamics, we show that this parameterization of ResNets admits a well-defined feature learning joint infinite-width and infinite-depth limit and show convergence of finite-size network dynamics towards this limit.

## 1 Introduction

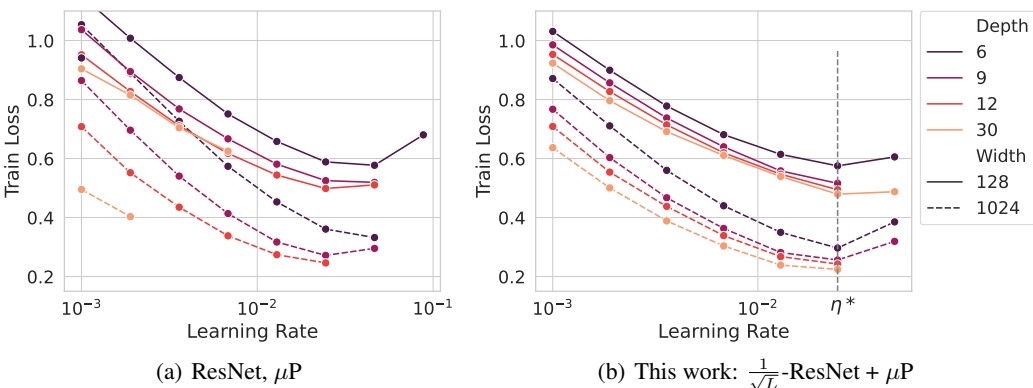

(a) ResNet, $\mu$P  (b) This work: $\frac{1}{\sqrt{L}}$-ResNet + $\mu$P

Figure 1: The optimal learning rate $\eta^*$ transfers across both depth and width in our proposed parameterization but not in $\mu$P or standard parameterization (Fig. B.2). Loss is plotted after 20 epochs on CIFAR-10. *All the missing datapoints indicate that the corresponding run diverged.*

Increasing the number of parameters in a neural network has led to consistent and often dramatic improvements in model quality (Kaplan et al., 2020; Hoffmann et al., 2022; Zhai et al., 2022; Klug & Heckel, 2023; OpenAI, 2023). To realize these gains, however, it is typically necessary to conduct by trial-and-error a grid search for optimal choices of hyperparameters, such as learning rates. Doing so

---

[*]Equal first authors.
[†]Equal senior authors.

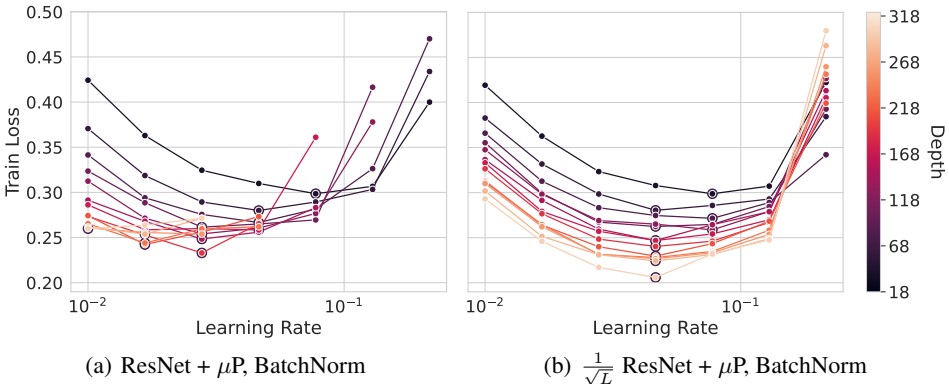

Figure 2: Effect of batch normalization on a standard Resnet18-type architecture (He et al., 2016). The above examples are taken after 20 epochs on CIFAR-10. Normalization layers can slightly improve consistency and trainability across depths in standard ResNets, but consistency is much more reliable with the $1/\sqrt{L}$ scaling (precisely, we use $\beta_\ell = 3/\sqrt{L}$ here to increase feature learning at finite depth). Runs that exceed a target loss of 0.5 are removed from the plot for visual clarity.

directly in SOTA models, which may have hundreds of billions (Brown et al., 2020) or even trillions of parameters (Fedus et al., 2022), often incurs prohibitive computational costs.

To combat this, an influential recent line of work by Yang & Hu (2021) proposed the so-called $\mu$P parameterization, which seeks to develop principles by which optimal hyperparameters from small networks can be reused — or *transferred* — to larger networks (Yang et al., 2021). The $\mu$P prescription focuses on transfer from narrower to wider models, but does not always transfer across depth (see Figure 1(a)). This leads us to pose the following problem:

**Question:** *Can we transfer hyperparameters simultaneously across depth and width?*

In this work, we provide an affirmative answer for a particular flexible class of residual architectures (including ResNets and transformers) with residual branches scaled like $1/\sqrt{\text{depth}}$ (see Equation 1 and Table 1 for the exact parameterization). More specifically, our empirical contributions can be summarized as follows:

- We show that a simple variant of the $\mu$P parameterization allows for transfer of learning rates simultaneously across depth and width in residual networks with $1/\sqrt{\text{depth}}$ scaling on the residual branches (see Equation 1 and Figure 1(b)). Moreover, we observe transfer not only for training with a fixed learning rate, but also with complex learning rate schedules (Figure 3(a-d)).

- We show that the lack of learning rate transfer across depth in $\mu$P ResNets can be partly overcome by adding normalization layers (Figure 2). However, we find on CIFAR-10, Tiny ImageNet, ImageNet, that the transfer can be significantly improved with our $1/\sqrt{\text{depth}}$ residual branches (Figures 3 and B.6).

- While most of our experiments concern residual convolutional networks, such as wide ResNets, we also present evidence that our prescription leads to learning rate transfer in Vision Transformers (ViTs), both with and *without* normalization (Figure 3).

- We demonstrate transfer for learning rates, as well as other hyperparameters such as momentum coefficients and regularization strengths in convolutional residual networks (Figure 4).

The reason hyperparameters can transfer is due to the existence of a scaling limit where the network predictions and internal features of the model move at a rate independent of the size of the network. The key underlying idea is that two neural networks will exhibit similar training dynamics (and hence similar optimal hyperparameters) if they are both finite size approximations to, or discretizations of, such a consistent scaling limit. In fact, the success of the mean field or $\mu$P scaling (Yang & Hu, 2021; Bordelon & Pehlevan, 2022b) is precisely due to the width-independent scale of feature updates. In the same spirit, we propose our parameterization so that the rate of feature up-

|  | SP (PyTorch) | $\mu$P / Mean Field | $\mu$P+1/$\sqrt{L}$-Residual (Ours) |
|---|---|---|---|
| Branch Scale $\beta_\ell$ | 1 | $\begin{cases} N^{-1/2}, & \ell > 0 \\ D^{-1/2}, & \ell = 0 \end{cases}$ | $\begin{cases} N^{-1/2}, & \ell = L \\ (LN)^{-1/2}, & 0 < \ell < L \\ D^{-1/2}, & \ell = 0 \end{cases}$ |
| Output Scale $\gamma$ | 1 | $\gamma_0 N^{1/2}$ | $\gamma_0 N^{1/2}$ |
| LR Schedule $\eta(t)$ | $\eta_0(t)$ | $\eta_0(t)\gamma_0^2 N$ | $\eta_0(t)\gamma_0^2 N$ |
| Weight Variance $\sigma_\ell^2$ | $\begin{cases} N^{-1}, & \ell > 0 \\ D^{-1}, & \ell = 0 \end{cases}$ | 1 | 1 |

Table 1: In the framework of Equation 1, different choices of coefficients leads to standard parameterization (SP, PyTorch default (Paszke et al., 2019)), $\mu$P / Mean Field parameterization, and our $\mu$P+1/$\sqrt{L}$-residual branch scaling which transfers over width and depth.

dates and prediction updates are also independent of the *depth* of the network. Our main theoretical contributions can be summarized as follows:

- Extending the infinite-width limit results of Bordelon & Pehlevan (2022b), we show the infinite-width-and-depth limit of a residual network with $1/\sqrt{\text{depth}}$ residual branch scaling and $\mu$P initialization can be characterized by a dynamical mean field theory (DMFT) (Section 4, 4.2) where the layer index takes on a continuous "layer time." In general, the network's internal features evolve during training in this limit. We show that a lazy limit of the DMFT dynamics, where features are frozen, can also be analyzed as a special case. See Section 4.3.1 and Proposition 2.
- We provide exact solutions of the DMFT dynamics in the rich (i.e. non-kernel) regime for a simple deep linear network. See Section 4.3.2.

The rest of the article is organized as follows. In Section 2, we define our parameterization and scaling rules. In Section 3 demonstrate our empirical results on hyperparameter transfer. Then in Section 4, we proceed to characterize the limiting process at infinite width and depth and the approximation error at finite $N, L$. Next we provide a brief overview of related works in Section 5, before concluding with a discussion of limitations and future directions in Section 6.

## 2 FRAMEWORK FOR DEPTHWISE HYPERPARAMETER TRANSFER

We consider the following type of residual network (and suitable architectural/optimization generalizations, see Appendix K) of width $N$ and depth $L$ with inputs $\boldsymbol{x} \in \mathbb{R}^D$ which are mapped to $N$-dimensional preactivations $\boldsymbol{h}^\ell \in \mathbb{R}^N$ and outputs $f(\boldsymbol{x}) \in \mathbb{R}$:

$$f(\boldsymbol{x}) = \frac{\beta_L}{\gamma} \boldsymbol{w}^L \cdot \phi(\boldsymbol{h}^L(\boldsymbol{x})), \quad \boldsymbol{h}^{\ell+1}(\boldsymbol{x}) = \boldsymbol{h}^\ell(\boldsymbol{x}) + \beta_\ell \boldsymbol{W}^\ell \phi(\boldsymbol{h}^\ell(\boldsymbol{x})), \quad \boldsymbol{h}^1(\boldsymbol{x}) = \beta_0 \boldsymbol{W}^0 \boldsymbol{x}, \quad (1)$$

where $\gamma$ is a scaling factor, $\phi(\cdot)$ the activation function, and the weights are initialized as $\boldsymbol{W}_{ij}^\ell \sim \mathcal{N}(0, \sigma_\ell^2)$ with a corresponding learning rate $\eta(t)$. See Table 1.

Compared to what is commonly done in practical residual architectures, such as ResNets and Transformers, our parameterization uses the $\sqrt{L}$ factor proposed in Hayou et al. (2021); Fischer et al. (2023), and the coefficient $\gamma$ from $\mu$P, but do not use normalization layers (Table 1). The parameters $\boldsymbol{\theta} = \{\boldsymbol{W}^0, ..., \boldsymbol{w}^L\}$ are updated with a gradient based learning rule. For concreteness consider SGD which we use in many experiments (we also allow for gradient flow and other optimizers in Appendix K):

$$\boldsymbol{\theta}(t+1) = \boldsymbol{\theta}(t) - \eta(t) \, \mathbb{E}_{\boldsymbol{x} \sim \mathfrak{B}_t} \nabla_{\boldsymbol{\theta}} \, \mathcal{L}[f(\boldsymbol{\theta}, \boldsymbol{x})], \quad (2)$$

where $\mathfrak{B}_t$ is the minibatch at iteration $t$, $\eta(t)$ is the learning rate schedule, and the loss function $\mathcal{L}$ depends on parameters through the predictions $f$ of the model.

The choice of how $(\gamma, \eta(t))$ should be scaled as width $N$ and depth $L$ go to infinity determines the stability and feature learning properties of the network. If $\gamma, \eta \sim \mathcal{O}(1)$[1] (*kernel regime*), then the

---

[1] We use $\mathcal{O}(1)$ to denote a constant independent of width $N$ and depth $L$.

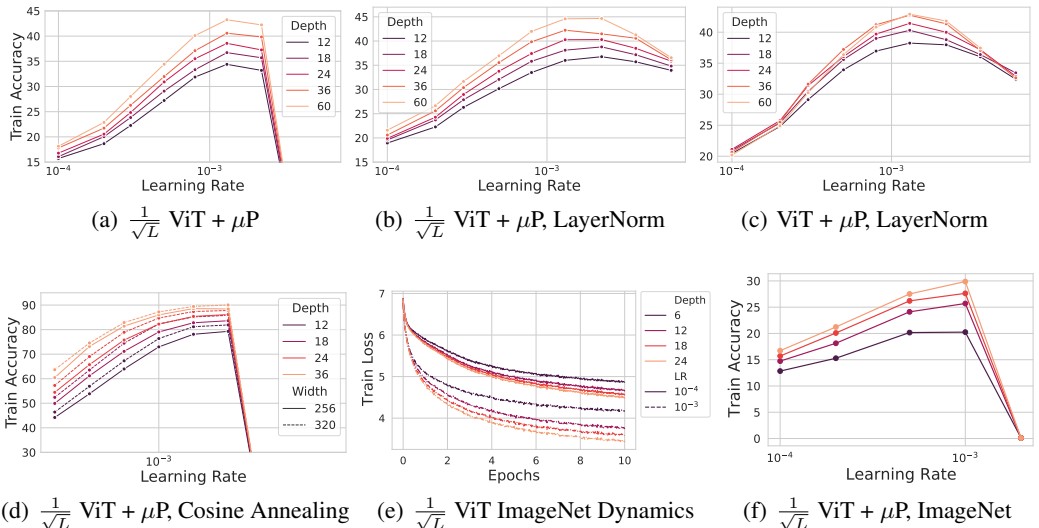

Figure 3: ViTs trained with Adam also exhibit learning rate transfer, with or without LayerNorm and $1/\sqrt{L}$-scaling. The above examples are taken after 20 epochs on Tiny ImageNet for (a)-(d) and ImageNet after 10 epochs for (e)-(f). In (a)-(c), the learning rate is linearly increased to the target in the first 1000 steps. In (d), after 2000 warm-up steps, the learning rate is decreased to zero with a cosine schedule in 100 epochs. Notice that in addition to the transfer across all the settings, the $1/\sqrt{L}$ models show a stronger non-saturating benefit of deeper models. (e)-(f) ViTs on first few epochs of ImageNet. Also, we note (a), (d), (e) and (f) refer to models *without normalization layers*.

network evolution is stable under gradient descent, but the network does not learn features at infinite width (Jacot et al., 2018; Lee et al., 2019; Yang & Hu, 2021). In this work, we study the richer feature learning (*mean field*) parameterization $\gamma = \gamma_0 \sqrt{N}$, $\eta(t) = \eta_0(t)\gamma_0^2 N$, where $\gamma_0, \eta_0(t)$ are independent of $N$. In this regime, the scaling factor of $1/\gamma$ in the last layer enables feature learning at infinite width, and the constant $\gamma_0$ controls the rate of feature learning. At fixed depth $L$, this mean field parameterization is capable of generating consistent feature learning dynamics across network widths $N$ (Yang & Hu, 2021; Bordelon & Pehlevan, 2022b).

One contribution of this work is identifying how $\eta_0, \gamma_0$ should be scaled with depth $L$ in order to obtain a stable feature-learning large $L$ limit. We argue that after the $\frac{1}{\sqrt{L}}$ factors have been introduced the correct scaling is simply

$$\eta = \eta_0 \gamma_0^2 N \ , \ \gamma = \gamma_0 \sqrt{N} \ , \ \eta_0, \gamma_0 \sim \mathcal{O}(1). \tag{3}$$

We show both theoretically and experimentally that this parameterization exhibits consistent learning dynamics and hyperparameter optima across large network depths $L$. To justify this choice, we will characterize the large width and depth $N, L \to \infty$ limit of this class of models.

## 3 EXPERIMENTAL VERIFICATION OF HYPERPARAMETER TRANSFER

We report here a variety of experiments on hyperparameter transfer in residual networks. We begin with a basic modification of equation 1 by replacing fully connected layers with convolutional layers. Apart from the proposed $1/\sqrt{L}$ scaling, notice that this convolutional residual model is similar to widely adopted implementations of ResNets (He et al., 2016), with the main difference being the absence of normalization layers, and a single convolution operator per residual block (details in Appendix A). We use this residual convolutional architecture in figures 1, 4, B.2 and B.1. We also experiment with a practical ResNet (in figures 2, B.4, B.6), which has two convolutional layers per block, and normalization layers (see Appendix A.2). Finally, we also perform selected experiments on Vision Transformers (Dosovitskiy et al., 2020), to demonstrate the broad applicability of our theoretical insights (Figure 3). Below we list the the main takeaways from our experiments.

**µP does not Transfer at Large Depths.** While µP exhibits transfer over network widths, it does not immediately give transfer over network depths. In Figure 1(a), we show the train loss after 20

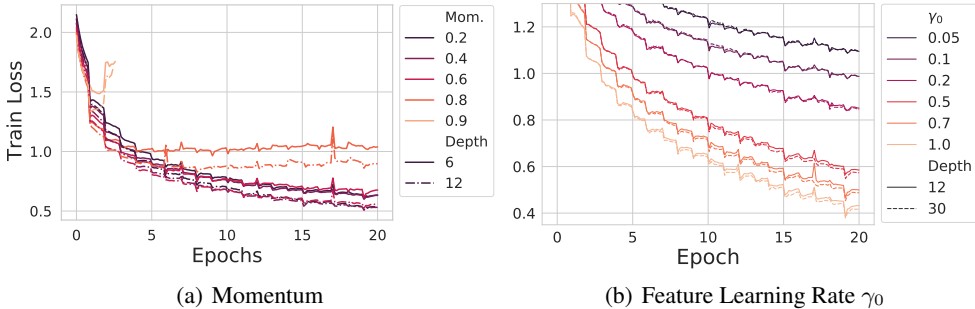

(a) Momentum

(b) Feature Learning Rate $\gamma_0$

Figure 4: Other hyperparameters also transfer. This example shows training dynamics during the first 20 epochs on CIFAR-10 (architecture details in Appendix A). The dynamics at two different depths are provided for (a) momentum (b) feature learning rate $\gamma_0$.

epochs on CIFAR-10. Notice how the network transfers over widths but not depths for $\mu$P. For completeness, in Fig. B.2 we show that also SP does not transfer.

$1/\sqrt{L}$**-scaled Residual Networks Transfer**. We repeat the same experiment, but now scale the residual branch by $1/\sqrt{L}$. The networks transfer over both width and depth (Figure 1 (b)).

**The Role of Normalization Layers**. Normalization layers, such as LayerNorm and BatchNorm, are commonly used instead of the $1/\sqrt{L}$-scaling in residual architectures. In Fig. 2, we repeat the learning rate transfer experiment with BatchNorm placed after the convolutional layer. In Fig. B.4 we repeat the same experiment with LayerNorm. In both cases, while it is known that the presence of normalization layers certainly helps the trainability of the model at larger depth (Daneshmand et al., 2021; Joudaki et al., 2023), we observe that the $1/\sqrt{L}$-scaled version exhibits a more consistent transfer across both width and depth.

**Empirical Verification on Vision Transformers with Adam.** Despite our theory being primarily on (S)GD-based training, we empirically test whether the transfer results can be extended to Vision Transformers (ViTs) trained with $\mu$P-parameterized Adam optimizer (Yang et al., 2021). In particular, we consider Pre-LN Transformers (Xiong et al., 2020), and adapt it to our setting by scaling all the residual branches by $1/\sqrt{L}$ (as in Noci et al. (2022)), and make the architecture compatible with the $\mu$P framework of Yang et al. (2021) (see Appendix A.3 for details). We test the variants both with and without LayerNorm. Results for 20 epochs training with the Tiny ImageNet dataset are shown in Fig. 3 (a-c), where we also see a good transfer across all the settings. The observation of a good transfer with depth for Pre-LN Transformers was empirically observed for translation tasks in Yang et al. (2021). Here, we additionally notice that while the performance benefits of Pre-LN Transformers saturates at large depth, the models with $1/\sqrt{L}$-scaling exhibits a more consistent improvement with depth. Finally, in Fig. 3(e-f) we successfully test scaling the learning rate transfer experiment of $1/\sqrt{L}$-scaled ViTs (without LayerNorm) to ImageNet (see Appendix B for ImageNet experiments with ResNets).

**Other Hyperparameters and Learning Rate Schedules also Transfer**. Given the standard practice of using learning rate schedulers in optimizing Transformers (Dosovitskiy et al., 2020; Touvron et al.), we show that linear warm-up (Fig 3(a-c)), optionally followed by a cosine decay (Fig 3(d)) also transfer consistently. Finally, in Figure 4, we plot the learning dynamics for momentum, weight decay, and the feature learning rate $\gamma_0$, which interpolates between the kernel and feature learning regimes (Bordelon & Pehlevan, 2022b). Despite the usual correct transfer, we highlight the *consistency* of the dynamics, in the sense that learning curves with the same hyperparameter but different depths are remarkably similar, especially at the beginning of training.

## 4 CONVERGENCE TO THE LARGE WIDTH AND DEPTH LIMIT

We argued that width/depth-invariant feature and prediction updates are crucial for hyperparameter transfer. To formalize this, let $f_{N,L}(t)$ be the network output after $t$ steps of optimization, where we explicitly write the dependence on width $N$ and depth $L$. For hyperparameter transfer we want

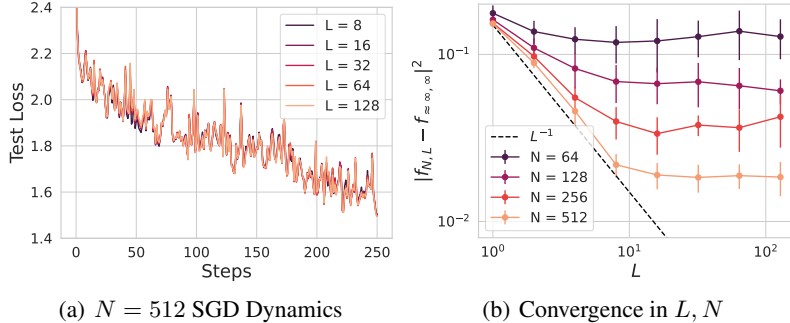

(a) $N = 512$ SGD Dynamics  (b) Convergence in $L, N$

Figure 5: Approximation of the joint $N \to \infty$, $L \to \infty$ limit requires both sufficiently large $N$ and $L$. CNNs are compared after 250 steps on CIFAR-10 with batchsize 32 with $\gamma_0 = 1.0$. Since we cannot compute the infinite width and depth predictor, we use a proxy $f_{\approx\infty,\infty}$: the ensemble averaged (over random inits) predictor of networks with $N = 512$ and $L = 128$. Error bars are standard deviation computed over 15 random initializations. (a) SGD dynamics at large width are strikingly consistent across depths. (b) The convergence of $f_{N,L}$ to a large width and depth proxy $f_{\approx\infty,\infty}$ is bottlenecked by width $N$ for small $N$ while for large $N$, it decays like $\mathcal{O}(L^{-1})$.

$f_{N,L}(t)$ to be close to $f_{N',L'}(t)$. One strategy to achieve this is to adopt a parameterization which makes finite networks as close as possible to a limit $f_{\infty,\infty}(t)$. Following this line of thought, we desire that a parameterization:

1. Have a well-defined large depth and width limit, i.e. $\lim_{N,L\to\infty} f_{N,L}(t)$.

2. Have internal features / kernels evolve asymptotically *independent of width $N$ or depth $L$*.

3. Have minimal finite size approximation error.

These desiderata[2] motivated the $\frac{1}{\sqrt{L}}$-$\mu$P ResNet in Equation 1 and Table 1. The existence of the limit for the forward pass at initialization $t = 0$ was shown in Hayou & Yang (2023); Cirone et al. (2023). Here, we study the learning dynamics in this model with $\mu$P parameterization, establish that the limit exists and show that feature updates are width and depth independent. Further, we characterize the limit throughout training in Appendices D, E, F. We attempt to visualize convergence to the limit in Figure 5 and Appendix Figure B.7. See Appendix H and I for theoretical analysis of finite size approximation error and Figure B.7 for additional tests of the sources of finite width/depth approximation error.

## 4.1 PRIMER ON THE DMFT IN THE $N \to \infty$ LIMIT

To derive our theoretical results, we use a technique known as dynamical mean field theory (DMFT) Bordelon & Pehlevan (2022b). DMFT describes the dynamics of neural networks at (or near) infinite width (preserving feature learning). In this setting one tracks a set of kernel order parameters

$$\Phi^\ell(\boldsymbol{x}, \boldsymbol{x}'; t, s) = \frac{1}{N}\phi(\boldsymbol{h}^\ell(\boldsymbol{x}, t)) \cdot \phi(\boldsymbol{h}^\ell(\boldsymbol{x}', s)), \quad G^\ell(\boldsymbol{x}, \boldsymbol{x}'; t, s) = \frac{1}{N}\boldsymbol{g}^\ell(\boldsymbol{x}, t) \cdot \boldsymbol{g}^\ell(\boldsymbol{x}', s), \quad (4)$$

where $\boldsymbol{g}^\ell(\boldsymbol{x}, t) \equiv N\gamma_0 \frac{\partial f(\boldsymbol{x},t)}{\partial \boldsymbol{h}^\ell(\boldsymbol{x},t)}$ are the back-propagation signals. DMFT reveals the following facts about the infinite width limit:

1. The dynamical kernels $\{\Phi^\ell, G^\ell\}$ and network outputs $f$ concentrate over random initializations of network weights throughout training.

2. The preactivations and gradient fields of each neuron $\{h_i^\ell, g_i^\ell\}$ become i.i.d. random variables drawn from a *single-site* (i.e. neuron marginal) density which depends on the kernels.

3. The kernels can be computed as averages over this *single-site* density.

---

[2]In addition, an ideal parameterization would also have the property that scaling up the width and depth of the model would monotonically improve performance, which empirically seems to hold in our parameterization.

The last two facts provide a self-consistency interpretation: given the kernels, one can compute or sample the distribution of preactivations $\{h^\ell, g^\ell\}$. Given the density of $\{h^\ell, g^\ell\}$, one can compute the kernels. Prior work on this limit has demonstrated that these equations can be approximately solved at finite $L$ with a Monte Carlo procedure (but with computational complexity cubic in the training iterations) (Bordelon & Pehlevan, 2022b).

## 4.2 THE LARGE-WIDTH-AND-DEPTH LIMIT DMFT EQUATIONS

From the finite-$L$ DMFT equations (App. D), we calculate the large depth $L \to \infty$ limit, generating a continuum process over the *layer time* $\tau = \frac{\ell}{L} \in [0, 1]$, a concept introduced in Li et al. (2022). [3]

**Proposition 1 (Informal)** *Consider network training dynamics of a ResNet as in Equation 1 with our $\mu P$ $\frac{1}{\sqrt{L}}$ scaling and the appropriate choice of learning rate and $\gamma_0$ in the infinite width and depth limit. The preactivation $h(\tau; \boldsymbol{x}, t)$ drawn from the marginal density of neurons in a layer at layer-time $\tau$, data point $\boldsymbol{x}$ and training time $t$ obeys a stochastic integral equation*

$$h(\tau; \boldsymbol{x}; t) = h(0; \boldsymbol{x}; t) + \int_0^\tau du(\tau'; \boldsymbol{x}; t) + \eta_0 \gamma_0 \int_0^\tau d\tau' \int_0^t ds \int d\boldsymbol{x}' C_h(\tau'; \boldsymbol{x}, \boldsymbol{x}'; t, s) g(\tau'; \boldsymbol{x}'; s)$$

(5)

*where $du(\tau; \boldsymbol{x}; t)$ is zero mean Brownian motion. The covariance of $du$ is the feature kernel $\Phi$ and the deterministic operator $C_h$ can be computed from the deterministic limiting kernels $\Phi, G$. The gradients $g(\tau)$ satisfy an analogous integral equation (See App. E.7 for formulas for $C_h$ and $g$). The stochastic variables $h, g$ and the kernels $\Phi, G$ satisfy a closed system of equations.*

A full proposition can be found in Appendix E.7. We derive this limit in the Appendix E, F. These equations are challenging to solve in the general case, as the $h$ and $g$ distributions are non-Gaussian. Its main utility in the present work is demonstrating that there is a well-defined feature-learning infinite width and depth limit and that, in the limit, the predictions, and kernels evolve by $\mathcal{O}(1)$.

## 4.3 SPECIAL EXACTLY SOLVEABLE CASES OF THE $N, L \to \infty$ DMFT EQUATIONS

Though the resulting DMFT equations for the kernels are intractable in the general case, we can provide analytical solutions in special cases to verify our approach. We first write down the training dynamics in the lazy limit by characterizing the neural tangent kernel (NTK) at initialization. Next, we discuss deep linear ResNets where the feature learning dynamics gives ODEs that can be closed while preserving Gaussianity of preactivations.

### 4.3.1 THE LAZY LEARNING LIMIT

A special case of the above dynamics potentially of interest is the limit of $\gamma_0 \to 0$ where the feature kernels are frozen. In this limit the NTK $K$ is static and can be computed at initialization and used to calculate the full trajectory of the network dynamics (see Appendix G.1).

**Proposition 2 (Informal)** *Consider a ResNet with the $\frac{1}{\sqrt{L}}$-$\mu P$ parameterization in the $N, L \to \infty$ limit. In the $\gamma_0 \to 0$ limit, the feature kernel $\Phi(\tau)$, gradient kernel $G(\tau)$ and NTK $K$ do not evolve during training and satisfy the following differential equations over layer time $\tau$*

$$\partial_\tau H(\tau; \boldsymbol{x}, \boldsymbol{x}') = \Phi(\tau; \boldsymbol{x}, \boldsymbol{x}') \, , \; \Phi(\tau; \boldsymbol{x}, \boldsymbol{x}') = \langle \phi(h)\phi(h') \rangle_{h,h' \sim \mathcal{N}(0, \boldsymbol{H}_{\boldsymbol{x},\boldsymbol{x}'}(\tau))}$$

$$\partial_\tau G(\tau; \boldsymbol{x}, \boldsymbol{x}') = -G(\tau; \boldsymbol{x}, \boldsymbol{x}') \left\langle \dot\phi(h)\dot\phi(h') \right\rangle_{h,h' \sim \mathcal{N}(0, \boldsymbol{H}_{\boldsymbol{x},\boldsymbol{x}'}(\tau))}$$

$$\boldsymbol{H}_{\boldsymbol{x},\boldsymbol{x}'}(\tau) = \begin{bmatrix} H(\tau; \boldsymbol{x}, \boldsymbol{x}) & H(\tau; \boldsymbol{x}, \boldsymbol{x}') \\ H(\tau; \boldsymbol{x}', \boldsymbol{x}) & H(\tau; \boldsymbol{x}', \boldsymbol{x}') \end{bmatrix} \, , \quad K(\boldsymbol{x}, \boldsymbol{x}') = \int_0^1 d\tau \, G(\tau, \boldsymbol{x}, \boldsymbol{x}') \Phi(\tau, \boldsymbol{x}, \boldsymbol{x}').$$

(6)

*The dynamics of the neural network outputs $f(\boldsymbol{x}, t)$ can be computed from $K(\boldsymbol{x}, \boldsymbol{x}')$.*

The forward ODE for $\Phi$ at initialization was studied by Hayou & Yang (2023); Cirone et al. (2023). Figure 6 shows this limiting NTK for a ReLU network. The finite depth kernels converge to the limiting large $L$ kernel (blue line in (a)) at a rate $\mathcal{O}(L^{-2})$ in square error.

---

[3]Alternatively, one can also take the large-$L$ limit to get a limiting *distribution* over order parameters (kernels, output logits, etc) at fixed $N$, which yields the same limiting equations when $N \to \infty$ (see App. F).

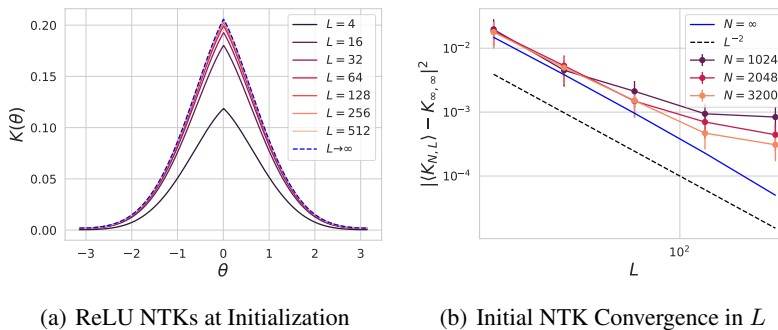

(a) ReLU NTKs at Initialization      (b) Initial NTK Convergence in $L$

Figure 6: Convergence of the depth $L$ NTK to the limiting kernel at initialization. (a) The infinite width NTK for a ReLU residual multilayer perpectron (MLP) for two inputs $\boldsymbol{x}, \boldsymbol{x}'$ separated by angle $\theta \in [-\pi, \pi]$ over varying depth $L$. The kernel converges as $L \to \infty$. (b) The convergence rate of the initialization averaged NTK $\langle K \rangle$ with network widths $N$ and network depths $L$ at initialization. For sufficiently large $N$, the *initial* NTK will converge at rate $\mathcal{O}(L^{-2})$ in square error.

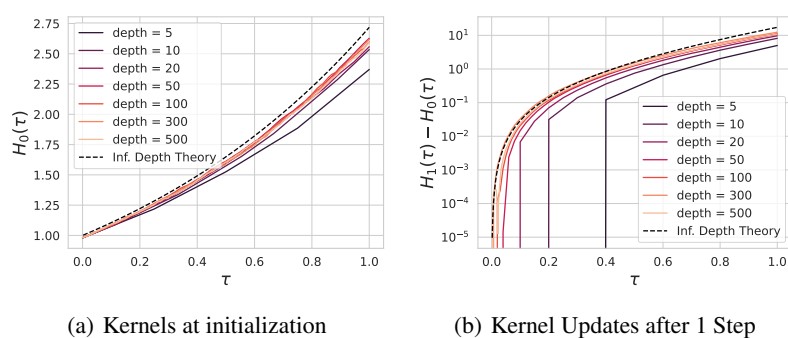

(a) Kernels at initialization      (b) Kernel Updates after 1 Step

Figure 7: We plot the feature kernels $H(\tau) = H^\ell|_{\ell = \tau L}$ in width $N = 1000$ residual linear networks before and after taking a gradient step in our parameterization. The kernel profiles at initialization $H_0(\tau)$ and the kernel changes $H_1(\tau) - H_0(\tau)$ converge for large $L$. We compare to the theoretical result expected from the $L \to \infty$ limit of the DMFT equations (dashed black).

### 4.3.2 DEEP LINEAR RESNETS

Next we explore the dynamics of feature learning in a solveable infinite depth ResNet with linear activations. In this case, the preactivations $h$ and gradient fields $g$ remain Gaussian over the full time-course of learning, rendering the DMFT equations tractable without any need for Monte Carlo methods. As an example, in Figure 7 we show the kernel updates in a deep linear ResNet after a single training step on a single training example $(\boldsymbol{x}, y)$. The theoretical solution to the kernel ODE at initialization $H_0(\tau) = e^\tau H_0(0)$ and after one step $H_1(\tau)$ is (see Appendix G.2.2 for derivation)

$$H_1(\tau) = e^\tau H_1(0) + 2\eta_0^2 \gamma_0^2 y^2 e^\tau \left[ \frac{1}{3}\tau^3 + \frac{1}{2}\tau^2 + (1 + e^{-1})(\tau^2 e^\tau - \tau e^\tau + e^\tau - 1) \right] \quad (7)$$

For networks with depths on the order of $\sim 100$ the infinite depth solution gives an accurate description of the profile of the kernel updates. More general equations for infinite depth linear networks are provided in Appendix G.2.

## 5 RELATED WORKS

The $1/\sqrt{L}$-scaled residual model (He et al., 2016) adopted in this work belongs to a rapidly expanding research domain. This approach has several advantageous characteristics in signal propagation, in terms of dynamical isometry (Tarnowski et al., 2019), as well as stable backward and forward kernels (Balduzzi et al., 2017; Hayou et al., 2021; Fischer et al., 2023). As a result, this has paved the

way for the development of reliable initialization methods (Taki, 2017; Zhang et al., 2019), including Transformers without normalization layers (Huang et al., 2020; Noci et al., 2022).

The study of neural network scaling limits started with MLPs in the infinite-width kernel regime (Jacot et al., 2018; Neal, 2012; Matthews et al., 2018; Lee et al., 2017), where the preactivations are also Gaussian with deterministic covariance. However, the joint depth and width limit has been shown to be non-Gaussian (Hanin & Nica, 2020; Noci et al., 2021) and have a stochastic covariance (Li et al., 2021). In particular, the feature covariance of particular classes of networks have a limit described by stochastic differential equations (SDEs) (Li et al., 2022; Noci et al., 2023), where the layer time is the depth-to-width ratio in this case. On the other hand, Hayou & Yang (2023) study the forward signal propagation in $1/\sqrt{L}$-scaled ResNets at initialization, where the preactivations converge to a Gaussian distribution with a *deterministic* covariance described by an ODE.

Transfer across width and depth was achieved on non-residual networks with the Automatic Gradient Descent algorithm proposed by Bernstein et al. (2020; 2023). Successful hyperparameter transfer across network widths in $\mu$P was shown in Yang et al. (2021). Width-consistent early time prediction and representation dynamics were also observed in an empirical study of Vyas et al. (2023). The infinite width limits of feature learning networks have been characterized with a mean-field PDE in two layer networks (Chizat & Bach, 2018; Mei et al., 2019; Chizat & Bach, 2020), Tensor programs (Yang, 2020; Yang & Hu, 2021), and DMFT techniques (Bordelon & Pehlevan, 2022b;a; 2023), which have been proven useful to understand the gradient dynamics in other settings as well (Mignacco et al., 2020; Gerbelot et al., 2022). This present work extends the DMFT technique to ResNets to analyze their feature learning behavior at large depth $L$.

Understanding feature learning in non-residual feedforward networks close to the asymptotic regime has also been studied with finite width and depth corrections from the corresponding infinite limit, both in the lazy (Chizat et al., 2019) NTK (Hanin, 2018; Hanin & Nica, 2019) and NNGP limits (Antognini, 2019; Yaida, 2020; Li & Sompolinsky, 2021; Zavatone-Veth et al., 2021; Segadlo et al., 2022; Hanin, 2022), and more recently in the rich feature learning regime (Bordelon & Pehlevan, 2023). At any finite width and depth the analysis is even more challenging and has been studied in special cases using combinatorial approaches and hypergeometric functions (Noci et al., 2021; Zavatone-Veth & Pehlevan, 2021; Hanin & Zlokapa, 2023).

## 6 DISCUSSION

In this work, we have proposed a simple parameterization of residual networks. In this parameterization, we have seen empirically that hyperparameter transfer is remarkably consistent across both width and depth for a variety of architectures, hyperparameters, and datasets. The experimental evidence is supported by theory, which shows that dynamics of all hidden layers are both non-trivial and independent of width and depth while not vanishing in the limit. We believe that these results have the potential to reduce computational costs of hyperparameter tuning, allowing the practitioner to train a large depth and width model only once with near optimal hyperparameters.

**Limitations.** While we demonstrate the existence of an infinite width and depth feature learning limit of neural network dynamics using statistical field theory methods and empirical tests, our results are derived at the level of rigor of physics, rather than formal proof. We also do not consider joint scaling limits in which the dataset size and number of optimization steps can scale with width $N$ or depth $L$. Instead, we treat time and sample size as $\mathcal{O}(1)$ quantities. Due to computational limits, most of our experiments are limited to 10-20 epochs on each dataset.

**Future Directions.** This paper explores one possible scaling limit of infinite width and depth networks. Other parameterizations and learning rate scaling rules may also be worth exploring in future work (Jelassi et al., 2023; Li et al., 2022; Noci et al., 2023). It would be especially interesting to see if an stochastic limit for the kernels can be derived for the training dynamics in ResNets without the $1/\sqrt{L}$ scaling factor. This parameterization makes it especially convenient to study depth-width trade-offs while scaling neural networks since hyperparameter dependencies are consistent across widths/depths (see Figure B.1). It would also be interesting to study our limiting dynamics in further detail to characterize what kinds of features are learned. Finally, it remains an open problem theoretically why hyperparameters transfer across widths and depths even when the actual predictions/losses differ across width and depth.

ACKNOWLEDGEMENTS

BB is supported by a Google PhD research fellowship. BB and CP are also supported by NSF Award DMS-2134157. This work has been made possible in part by a gift from the Chan Zuckerberg Initiative Foundation to establish the Kempner Institute for the Study of Natural and Artificial Intelligence. BH and MBL are supported by NSF grant DMS-2133806. BH is further supported by NSF CAREER grant DMS-2143754, NSF grant DMS-1855684 and an ONR MURI on Foundations of Deep Learning. We would like to thank Jacob Zavatone-Veth, Mary Letey, Alex Atanasov for comments on early versions of this manuscript.

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

APPENDIX

# A   EXPERIMENTAL DETAILS

## A.1   SIMPLIFIED RESNET

We design a residual architecture following the footprint of Eq. 1, and replacing the weight matrices with 1-strided convolutional layers with $3 \times 3$ kernel size. We max-pool the feature map at three selected layers equally distributed across depth. For instance, for a depth=12 model, we perform average pooling after the third, 6-th and 9-th layer. After each average pooling, we apply an extra convolutional layer to double the number of channels, which play the role of width in a convolutional network. Hence, also the width doubled three times across the depth (one time per average pooling). The final feature map is then flattened and passed through a final readout layer. The reported width in all the figures is the last width before the flattening operations. In the experiment with the $1/\sqrt{L}$-scaling, we divide by the total number of layers instead of the number of residual blocks (i.e. $L = 3 + \#$ blocks). This extra factor does not affect the scaling limit.

## A.2   RESNET

Compared to the simplified ResNet architecture of the previous section, the ResNet family as proposed in He et al. (2016) are augmented with two convolutional layers per residual block, each followed by batch normalization normalization. The subsampling is performed by 2-strided convolutional layers, for a total of $4$ times across the depth. For further details, we refer to He et al. (2016). In some experiments (e.g. Fig. B.4), we replace batch norm with LayerNorm applied across each two dimensional channel.

## A.3   VISION TRANSFORMER

We use the standard ViT architecture as in Dosovitskiy et al. (2020), i.e. we tokenize the input images by splitting it into patches of fixed size, and embed it to $N$ dimension using a linear layer. Then we apply a number of the usual Transformer blocks, each one composed of two residual blocks, a Softmax-based self attention block, and an MLP block with GeLU activation function. If used, we place LayerNorm at the beginning of each residual branch (Pre-LN configuration). Finally, we apply the following modification to make it compatible with $\mu$P (Yang et al., 2021):

- We re-scale the Softmax logits (i.e. the product of queries and keys) of each attention layer by $1/N_q$. The usual convention would be to use instead $1/\sqrt{N_q}$.

- Initialize the queries to zero.

- Initialize all the other weights to have standard deviation of $1/\sqrt{N}$, where $N$ here is the model dimension.

## A.4   DETAILS

We always use data augmentation, in particular the following random transformations:

- CIFAR-10: horizontal flips, 10-degree rotations, affine transformations, color jittering.

- Tiny ImageNet and ImageNet: crops, horizontal flips.

Across all the experiments, we fix the batch size to 64. Unless stated otherwise, all the hyperparameters that are not tuned are set to a default value: $\beta, \gamma_0, \sigma_\ell = 1$. For ViTs, we use a patch size We use the $\mu$P implementation of the $\mu$-Readout layer, and optimizers (SGD, and Adam under $\mu$P parametrizstion) as in the released $\mu$P package.

# B  ADDITIONAL EXPERIMENTS

## B.1  DEPTH WIDTH TRADE-OFF

While tuning the architecture at small widths/depths, it is interesting to study along which direction — width or depth — the architecture should be scaled. To test this, we train the residual convolutional model (Sec. A.1) at relatively large scale (up to almost a billion parameters), for 20 epochs on CIFAR-10 with fixed learning rate of $0.046$ (using data augmentation, see Sec. A). We report the results in Fig. B.1, where we plot the train loss as a function of parameters. In particular, in Fig. B.1(a) we highlight models of same widths, while in Fig. B.1(a) models of equal depth. We find that the Pareto optimal curve coincides by the depth 12 model for the range of widths/depths used for the experiment. As shown in Fig. B.1, we then fit power laws to the curves of constant depth, which seems to predict an optimal depth 30 at very large widths. Notice that we fix the number of epochs. We hypothesize the optimal curve to shift to different trade-offs as the number of epochs increases. We also expect different trade-offs for different datasets and models. However, extensively studying width-depth trade-off is beyond the scope of this work, thus we leave it as a potential direction for future work.

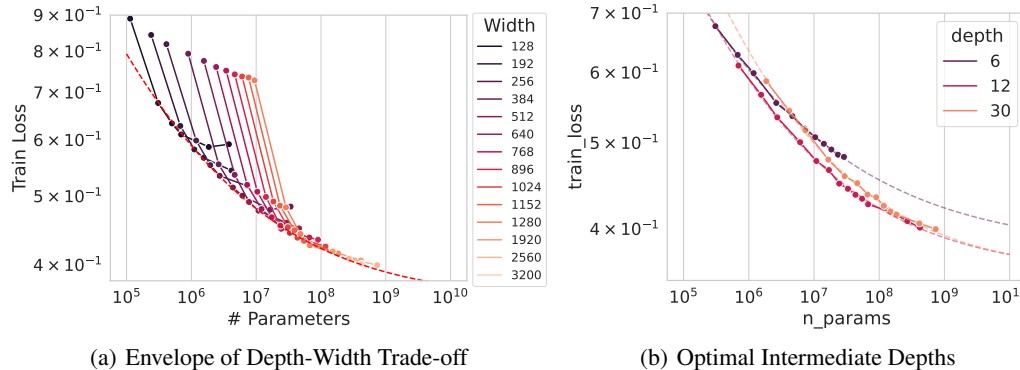

(a) Envelope of Depth-Width Trade-off  (b) Optimal Intermediate Depths

Figure B.1:  Since all models have hyperparameter transfer across depths and widths, one can explore a trade-off between depth and width at the optimal learning rate. We illustrate this idea with a simple experiment on CIFAR-10. (a) We plot the final train loss as a function of parameters for different widths. (b) intermediate depth of $\sim 9 - 12$ is preferred at fixed parameter count.

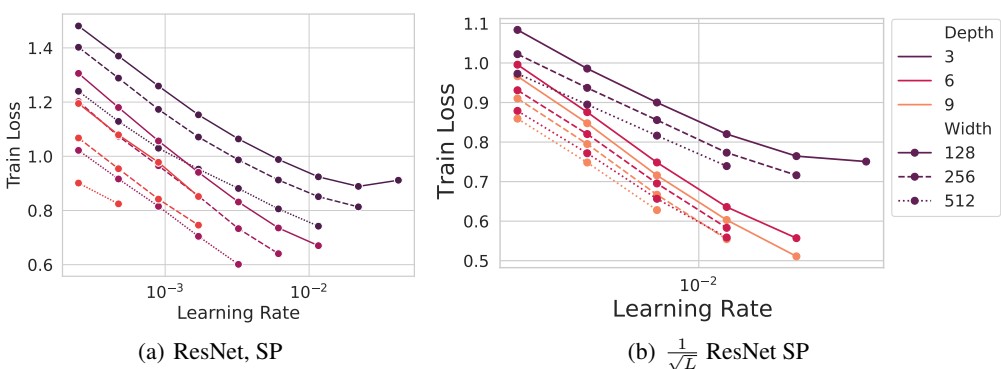

(a) ResNet, SP  (b) $\frac{1}{\sqrt{L}}$ ResNet SP

Figure B.2: Learning rates transfer across both depth and width in our proposed parameterization but not in standard parameterization (SP) or $\mu$P. Loss is plotted after 20 epochs on CIFAR-10. *All the missing datapoints indicate that the corresponding run diverged.* In particular, all depth 30 runs diverged when using $\mu$P without $1/\sqrt{\text{depth}}$ scaling.

## B.2 How Long Should you Train for Accurate Transfer?

An important question is after how many steps the optimal hyperparameters are fixed and can hence be selected without training for longer. In Fig. B.3(a), we plot the training loss profiles for two different depths and 4 different learning rates for a residual convolutional network of width $512$ trained on CIFAR-10. Notice how after the first few epochs it is already possible to decide the optimal learning rate. We confirm this for a ViT trained on Tiny ImageNet in Fig. B.3(b-c), where the optimal learning rate at two selected epochs (3 and 9) is the same.

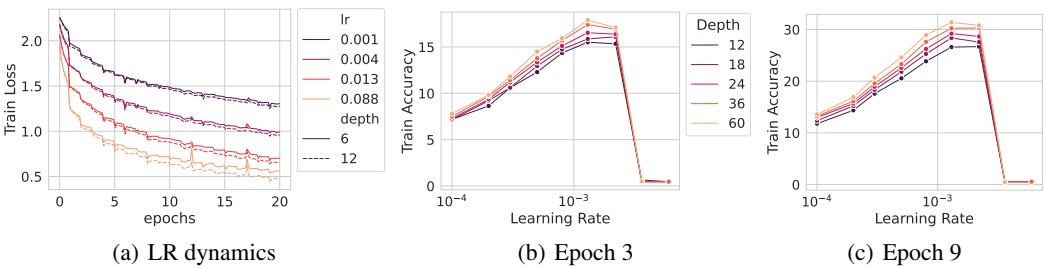

(a) LR dynamics        (b) Epoch 3        (c) Epoch 9

Figure B.3: (a) dynamics of train loss for different depths and learning rates. The network is the residual convolutional network trained on CIFAR-10. (b-c) learning rate tranfer plots for the same ViT at two different checkpoints (epoch 3 and 5). Notice how the optimal learning rate is determined after the first few epochs of training.

## B.3 Other Experiments

In Fig. B.2, we show how the learning rate does not transfer under SP for the residual model considered in this work, both with and without the $1/\sqrt{L}$-scaling. In Fig. B.4, we train models from the ResNets family, replacing BatchNorm with LayerNorm, concluding that the version with $1/\sqrt{L}$ scaling exhibits a better transfer. In Fig. B.6 we show our results on learning rate transfers for ImageNet, and in Fig. B.8, we show how also weight decay transfer under the proposed $1/\sqrt{L}$-scaling and has very consistent dynamics. The model used is the simplified residual convolutional model, trained for 20 epochs on CIFAR-10.

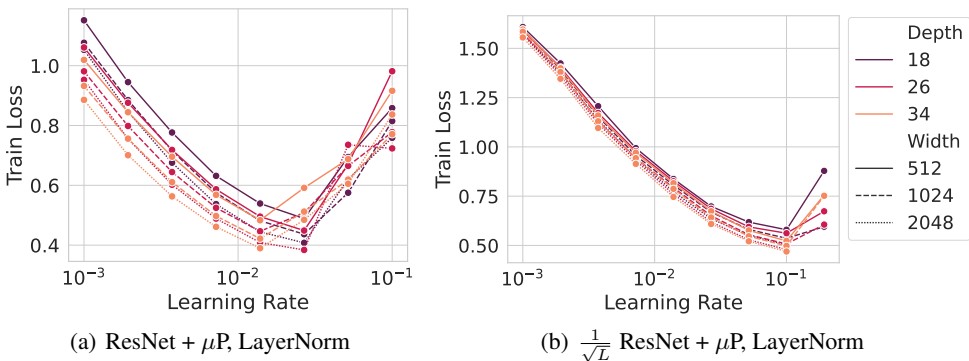

(a) ResNet + $\mu$P, LayerNorm        (b) $\frac{1}{\sqrt{L}}$ ResNet + $\mu$P, LayerNorm

Figure B.4: Effect of normalization layers. The above examples are taken after 20 epochs on CIFAR-10. Normalization layers can slightly improve consistency across depths in standard ResNets, but consistency is much more reliable with the $1/\sqrt{L}$ scaling.

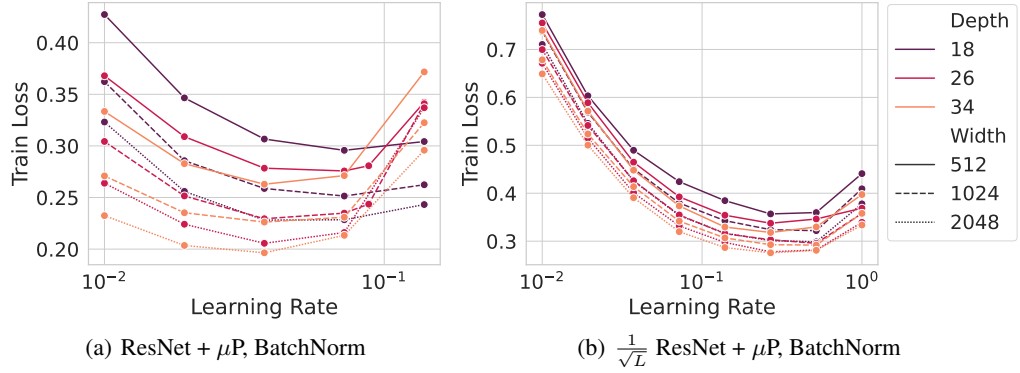

(a) ResNet + $\mu$P, BatchNorm

(b) $\frac{1}{\sqrt{L}}$ ResNet + $\mu$P, BatchNorm

Figure B.5: Effect of normalization layers. The above examples are taken after 20 epochs on CIFAR-10. Normalization layers can slightly improve consistency across depths in standard ResNets, but consistency is much more reliable with the $1/\sqrt{L}$ scaling.

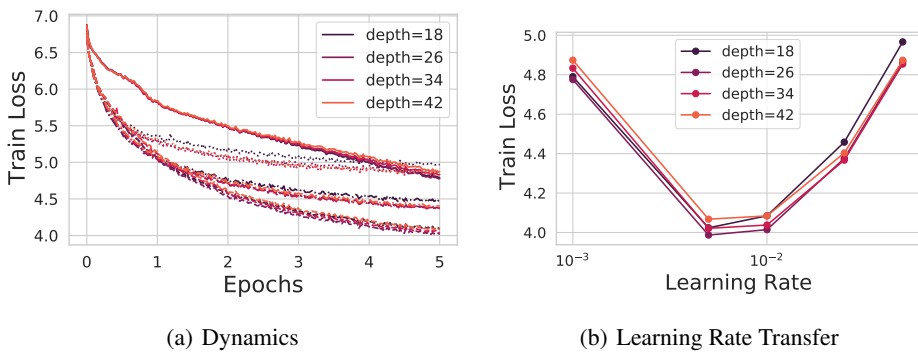

(a) Dynamics

(b) Learning Rate Transfer

Figure B.6: Learning rate + depth sweep on ImageNet for $5$ epochs with depth extrapolations of ResNet-18 *without any normalization layers*. Training is performed with SGD with momentum $0.9$ and batchsize $128$. (a) The loss curves for different learning rates (each learning rate is a distinct linestyle) and for different depths (colors). Dynamics are strikingly consistent across depths for each learning rate. (b) The optimal learning rate is preserved across depths.

## C  A REVIEW OF VARIOUS WIDTH PARAMETERIZATIONS

In this section, we discuss various parameterizations of neural networks with respect to the width of the model. We will discuss and compare their large width behaviors. First we will discuss the Neural-Tangent parameterization (NTP) which gives rise to a kernel limit. Next, we will discuss the Mean Field /$\mu$P parameterization which allows for feature learning at infinite width. In all models, we focus on

$$f = \frac{1}{\sqrt{N}\gamma} \boldsymbol{w}^L \cdot \phi(\boldsymbol{h}^L) \tag{8}$$

The choice of whether one is in NTP or $\mu$P depends on how $\gamma$ is scaled with respect to $N$. When training with a learning rate $\eta = \gamma^2 \eta_0$ (to ensure $\frac{d}{dt}f|_{t=0} = \mathcal{O}_\gamma(1)$), we have the following scale of internal preactivation updates (Bordelon & Pehlevan, 2022b)

$$\frac{d}{dt}\boldsymbol{h}^\ell = \mathcal{O}(\gamma N^{-1/2}). \tag{9}$$

Depending on how $\gamma$ is scaled with $N$, we can either have vanishing, constant, or diverging feature learning with respect to width $N$. In discrete time, a diverging parameterization is *unstable* (Yang & Hu, 2021). Below we discuss two of the popular stable scalings for analysis.

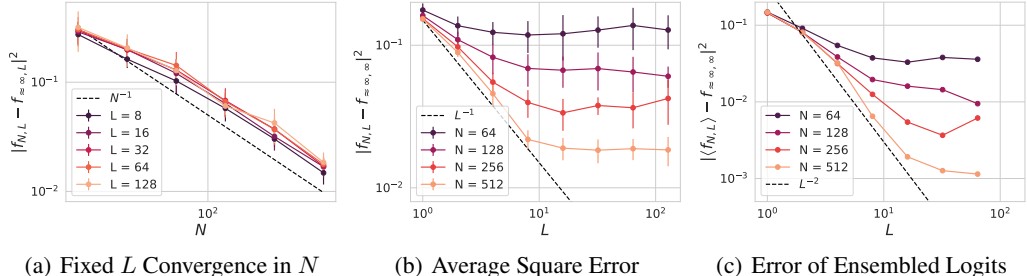

(a) Fixed $L$ Convergence in $N$  (b) Average Square Error  (c) Error of Ensembled Logits

Figure B.7: Further numerical support of our finite $N, L$ error decomposition in Appendix I through ensembling logits over $E = 15$ random seeds in the same experimental setting as Figure 5. (a) At each depth $L$, the finite width $N$ network converges in square error as $\mathcal{O}_{N,L}(N^{-1})$, independent of $L$. (b) The full expected square error from Figure 5 shows a convergence rate of $\mathcal{O}(L^{-1})$ for sufficiently large $N$, consistent with theory. (c) The ensemble averaged logits $\langle f_{L,N} \rangle$ converge to the infinite depth proxy at a faster rate, which is consistent with the theoretical $\mathcal{O}(L^{-2})$ for sufficiently large $N$.

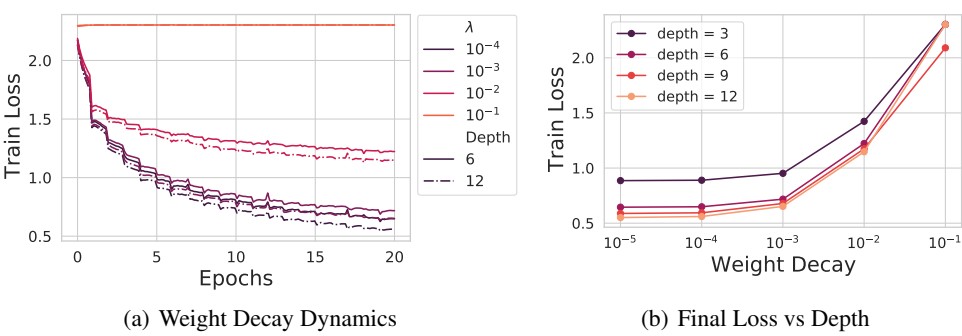

(a) Weight Decay Dynamics  (b) Final Loss vs Depth

Figure B.8: Weight decay dynamics also transfer across depths.

## C.1 Neural Tangent Parameterization

In NTP, the parameter $\gamma = \mathcal{O}_N(1)$ is a constant with respect to width $N$. As a consequence finite width $N$ models only exhibit $\mathcal{O}(N^{-1/2})$ movement of preactivations and $\mathcal{O}(N^{-1})$ movement of kernels. In this parameterization, models at very large width are well approximated by a kernel method (Jacot et al., 2018; Lee et al., 2019).

## C.2 Mean Field/$\mu$P Parameterizations

On the other hand, if $\gamma = \gamma_0 \sqrt{N}$ for some constant $\gamma_0$, then the network converges to a feature learning infinite width limit. This is precisely the mean-field/$\mu$P parameterization (Yang & Hu, 2021; Bordelon & Pehlevan, 2022b;a). The work of Yang & Hu (2021) also gives a precise sense in which this parameterization is *maximal*: each layer is learning features so that the preactivations in the sense that

$$\frac{1}{\sqrt{N}} \left( \frac{d}{dt} \boldsymbol{W}^\ell \right) \phi(\boldsymbol{h}^\ell) = \mathcal{O}(1) \tag{10}$$

Thus, the dynamics for $\frac{d}{dt} \boldsymbol{h}^{\ell+1} = \frac{d}{dt} \boldsymbol{h}^\ell + \frac{1}{\sqrt{N}} \left( \frac{d}{dt} \boldsymbol{W}^\ell \right) \phi(\boldsymbol{h}^\ell) + \frac{1}{\sqrt{N}} \boldsymbol{W}^\ell \left( \frac{d}{dt} \phi(\boldsymbol{h}^\ell) \right) = \mathcal{O}_N(1)$. Because the underlying feature learning updates within the network are approximately constant as width $N$ is increased, this model exhibits more consistent dynamics and better hyperparameter transfer across widths (Yang et al., 2022; Vyas et al., 2023).

We note that the code for $\mu$P released in the paper by Yang et al. (2022) uses an equivalent version of this dynamical system where learning rates do not need to be explictly scaled with $N$.

# D   DERIVATION OF THE $N \to \infty$ LIMIT AT FIXED $L$

In this section, we characterize the feature learning dynamics of the $N \to \infty$ limit of fixed depth $L$ residual networks. Our derivation is an extension of the approach presented in (Bordelon & Pehlevan, 2022b). Before taking the limit, we first identify our order parameters and weight space dynamics. Then we take the $N \to \infty$ limit using a technique known as Dynamical Mean Field Theory (DMFT).

## D.1   FINITE $N, L$ FUNCTION AND WEIGHT DYNAMICS

We start by computing the neural tangent kernel (NTK). For the sake of brevity, we will first write the equations where the read-in $\boldsymbol{W}^0$ weights are held fixed and will add their contribution to the dynamics in Appendix K.3. Excluding these two weight matrices from the present discussion does not alter any of the scalings with width $N$ or depth $L$. The dynamics of the prediction $f$ under gradient flow take the form

$$\frac{d}{dt}f(\boldsymbol{x};t) = \eta_0 \, \mathbb{E}_{\boldsymbol{x}'} K(\boldsymbol{x},\boldsymbol{x}';t)\Delta(\boldsymbol{x}';t) \tag{11}$$

where $\Delta(\boldsymbol{x};t) = -\frac{\partial \mathcal{L}(f(\boldsymbol{x},t))}{\partial f(\boldsymbol{x},t)}$ is an error signal and $K$ is the NTK of this model, which takes the form

$$K(\boldsymbol{x},\boldsymbol{x}') = \gamma_0^2 N \sum_{\ell=1}^{L} \frac{\partial f(\boldsymbol{x})}{\partial \boldsymbol{W}^\ell} \cdot \frac{\partial f(\boldsymbol{x}')}{\partial \boldsymbol{W}^\ell}. \tag{12}$$

Explicitly calculating the gradients of the output with respect to the weights $\boldsymbol{W}^\ell$ gives

$$N\gamma_0 \frac{\partial f(\boldsymbol{x})}{\partial \boldsymbol{W}^\ell} = \frac{1}{\sqrt{NL}} \boldsymbol{g}^{\ell+1}(\boldsymbol{x})\phi(\boldsymbol{h}^\ell(\boldsymbol{x}))^\top \;,\; g_i^\ell \equiv N\gamma_0 \frac{\partial f(\boldsymbol{x})}{\partial h_i^\ell(\boldsymbol{x})} \sim \mathcal{O}_{N,L}(1) \tag{13}$$

Using this above formula, we find that the NTK is $\mathcal{O}(1)$ with respect to $N, L$

$$K(\boldsymbol{x},\boldsymbol{x}') = \frac{1}{L}\sum_{\ell=1}^{L} G^{\ell+1}(\boldsymbol{x},\boldsymbol{x}')\Phi^\ell(\boldsymbol{x},\boldsymbol{x}') \tag{14}$$

where we introduced the $\mathcal{O}(1)$ feature kernel $\Phi^\ell$ and gradient kernels $G^\ell$ which have the form

$$\Phi^\ell(\boldsymbol{x},\boldsymbol{x}') = \frac{1}{N}\phi(\boldsymbol{h}^\ell(\boldsymbol{x})) \cdot \phi(\boldsymbol{h}^\ell(\boldsymbol{x}')) \;,\; G^\ell(\boldsymbol{x},\boldsymbol{x}') = \frac{1}{N}\boldsymbol{g}^\ell(\boldsymbol{x}) \cdot \boldsymbol{g}^\ell(\boldsymbol{x}') \tag{15}$$

The base cases for the feature and gradient kernels are given by

$$\Phi^0(\boldsymbol{x},\boldsymbol{x}') = \frac{1}{D}\boldsymbol{x} \cdot \boldsymbol{x}' \;,\; G^{L+1}(\boldsymbol{x},\boldsymbol{x}') = 1. \tag{16}$$

We next start by computing the dynamics of the weights under gradient flow

$$\frac{d}{dt}\boldsymbol{W}^\ell = \frac{\eta_0 \gamma_0}{\sqrt{LN}} \, \mathbb{E}_{\boldsymbol{x}} \, \Delta(\boldsymbol{x};t)\boldsymbol{g}^{\ell+1}(\boldsymbol{x},t)\phi(\boldsymbol{h}^\ell(\boldsymbol{x},t))^\top. \tag{17}$$

Integrating this equation from time 0 to time $t$, we get the following

$$\boldsymbol{h}^{\ell+1}(\boldsymbol{x},t) = \boldsymbol{h}^\ell(\boldsymbol{x},t) + \frac{1}{\sqrt{NL}}\boldsymbol{W}^\ell(0)\phi(\boldsymbol{h}^\ell)$$
$$+ \frac{\eta_0 \gamma_0}{L}\int_0^t ds \mathbb{E}_{\boldsymbol{x}'}\Delta(\boldsymbol{x}',s)\boldsymbol{g}^{\ell+1}(\boldsymbol{x}',s)\Phi^\ell(\boldsymbol{x},\boldsymbol{x}';t,s) \tag{18}$$

where $\Phi^\ell(\boldsymbol{x},\boldsymbol{x}';t,s) = \frac{1}{N}\phi(\boldsymbol{h}^\ell(\boldsymbol{x},t)) \cdot \phi(\boldsymbol{h}^\ell(\boldsymbol{x}';s))$ is the double time-index generalization of the feature kernel. Similarly, we obtain a backward pass relation of the form

$$\boldsymbol{g}^\ell(\boldsymbol{x},t) = \boldsymbol{g}^{\ell+1}(\boldsymbol{x},t) + \frac{1}{\sqrt{NL}}\dot{\phi}(\boldsymbol{h}^\ell(\boldsymbol{x},t)) \odot \boldsymbol{W}^\ell(0)^\top \boldsymbol{g}^{\ell+1}$$
$$+ \frac{\eta_0 \gamma_0}{L}\dot{\phi}(\boldsymbol{h}^\ell(\boldsymbol{x},t)) \odot \int_0^t ds \mathbb{E}_{\boldsymbol{x}'}\Delta(\boldsymbol{x}',s)G^{\ell+1}(\boldsymbol{x},\boldsymbol{x}';t,s)\phi(\boldsymbol{h}^\ell(\boldsymbol{x}',s)) \tag{19}$$

These equations define update rules for the preactivations $\boldsymbol{h}^\ell$ and the gradients $\boldsymbol{g}^\ell$ in terms of the feature and gradient kernels $\{\Phi^\ell, G^\ell\}$. However the above equations explicitly depend on the random initial weights $\boldsymbol{W}^\ell(0)$ through which time-varying signals propagate. Our ultimate goal is to gain insight into the effective dynamics over random initializations in the large $N$ limit. To do that, we will invoke dynamical mean field theory. In the next section, we characterize how these terms behave at large $N$ using a saddle point argument.

## D.2 INTUITIVE EXPLANATION OF LARGE WIDTH FEATURE LEARNING LIMIT

In the limit of $N \to \infty$ with $\gamma_0$ held fixed, the above equations can be simplified. The two key effects that occur at large width are

- Kernels (feature kernels, gradient kernels, NTK) and output logits concentrate by a law of large numbers.

- At each layer, each of the hidden neuron's activities becomes independent draws from a *single site stochastic process* which is characterized by the kernels.

Once the single site density is known, the kernels can be computed as averages over the single site density. The next section derives this result formally from a Martin-Siggia-Rose Path Integral derivation of the Dynamical Mean Field Theory (Martin et al., 1973).

## D.3 PATH-INTEGRAL AND SADDLE POINT EQUATIONS IN $N \to \infty$ LIMIT

This computation sketches the DMFT derivation of the limiting large $N$ process. Examples of detailed computations of the DMFT action $\mathcal{S}$ can be found in the Appendix of Bordelon & Pehlevan (2022b;a). Our notation is chosen to match the derivations found in their Appendix.

To characterize the effect of the random initial weights, we will attempt to characterize the distribution of $\boldsymbol{h}^\ell(\boldsymbol{x}, t), \boldsymbol{g}^\ell(\boldsymbol{x}, t)$ by tracking the moment generating functional of these fields

$$Z[\{\boldsymbol{j}^h, \boldsymbol{j}_g\}] = \mathbb{E}_{\{\boldsymbol{W}^\ell(0)\}} \exp\left( \sum_{\ell=1}^{L} \int d\boldsymbol{x} \int dt \, [\boldsymbol{j}_h^\ell(\boldsymbol{x}, t) \cdot \boldsymbol{h}^\ell(\boldsymbol{x}, t) + \boldsymbol{j}_g(\boldsymbol{x}, t) \cdot \boldsymbol{g}^\ell(\boldsymbol{x}, t)] \right) \quad (20)$$

Moments of the fields can be obtained with differentiation near zero source, for example

$$\lim_{\{\boldsymbol{j}_h^\ell, \boldsymbol{j}_g^\ell\} \to 0} \frac{\delta}{\delta j_{h,i_1}^\ell(\boldsymbol{x}_1, t_1)} \cdots \frac{\delta}{\delta j_{h,i_k}^\ell(\boldsymbol{x}_k, t_k)} Z[\{\boldsymbol{j}^h, \boldsymbol{j}_g\}] = \left\langle h_{i_1}^{\ell_1}(\boldsymbol{x}_1, t_1) ... h_{i_k}^{\ell_k}(\boldsymbol{x}_k, t_k) \right\rangle. \quad (21)$$

Now, following Bordelon & Pehlevan (2022b), we introduce the following useful auxiliary fields

$$\boldsymbol{\chi}^{\ell+1}(\boldsymbol{x}, t) = \frac{1}{\sqrt{N}} \boldsymbol{W}^\ell(0) \phi(\boldsymbol{h}^\ell(\boldsymbol{x}, t)) \,, \; \boldsymbol{\xi}^\ell(\boldsymbol{x}, t) = \frac{1}{\sqrt{N}} \boldsymbol{W}^\ell(0)^\top \boldsymbol{g}^{\ell+1}(\boldsymbol{x}, t) \quad (22)$$

The choice to introduce these auxiliary fields is apparent once one realizes that the $h$ and $g$ fields can be regarded as functions of $\chi, \xi$ and the kernels $\Phi, G$. We enforce the constraints defining these fields with Dirac-delta functions by repeatedly multiplying by unity

$$1 = \int d\boldsymbol{\chi}^{\ell+1}(\boldsymbol{x}, t) \, \delta \left( \boldsymbol{\chi}^{\ell+1}(\boldsymbol{x}, t) - \frac{1}{\sqrt{N}} \boldsymbol{W}^\ell(0) \phi(\boldsymbol{h}^\ell(\boldsymbol{x}, t)) \right)$$

$$= \int \frac{d\hat{\boldsymbol{\chi}}^{\ell+1}(\boldsymbol{x}, t) d\boldsymbol{\chi}^{\ell+1}(\boldsymbol{x}, t)}{(2\pi)^N} \exp \left( i\hat{\boldsymbol{\chi}}^{\ell+1}(\boldsymbol{x}, t) \cdot \left[ \boldsymbol{\chi}^{\ell+1}(\boldsymbol{x}, t) - \frac{1}{\sqrt{N}} \boldsymbol{W}^\ell(0) \phi(\boldsymbol{h}^\ell(\boldsymbol{x}, t)) \right] \right)$$
$$(23)$$

for all $\ell, \boldsymbol{x}, t$. And similarly for the $\boldsymbol{g}^\ell$ fields. After this insertion, the averages over $\{\boldsymbol{W}^\ell(0)\}$ can be performed with Gaussian integration. Concretely, we obtain the following average over Gaussian

initialization $\boldsymbol{W}^\ell(0)$

$$
\ln \mathbb{E} \exp \left( -\frac{i}{\sqrt{N}} \text{Tr} \, \boldsymbol{W}^\ell(0)^\top \int d\boldsymbol{x} \int dt \left[ \hat{\boldsymbol{\chi}}^{\ell+1}(\boldsymbol{x},t) \phi(\boldsymbol{h}^\ell(\boldsymbol{x},t))^\top + \boldsymbol{g}^{\ell+1}(\boldsymbol{x},t) \boldsymbol{\xi}^\ell(\boldsymbol{x},t)^\top \right] \right)
$$

$$
= -\frac{1}{2} \int dt ds d\boldsymbol{x} d\boldsymbol{x}' \, \hat{\boldsymbol{\chi}}^{\ell+1}(\boldsymbol{x},t) \cdot \hat{\boldsymbol{\chi}}^{\ell+1}(\boldsymbol{x}',s) \Phi^\ell(\boldsymbol{x},\boldsymbol{x}';t,s)
$$

$$
- \frac{1}{2} \int dt ds d\boldsymbol{x} d\boldsymbol{x}' \, \hat{\boldsymbol{\xi}}^\ell(\boldsymbol{x},t) \cdot \hat{\boldsymbol{\xi}}^\ell(\boldsymbol{x}',s) G^{\ell+1}(\boldsymbol{x},\boldsymbol{x}';t,s)
$$

$$
- i \int dt ds d\boldsymbol{x} d\boldsymbol{x}' \, \hat{\boldsymbol{\chi}}^{\ell+1}(\boldsymbol{x},t) \cdot \boldsymbol{g}^{\ell+1}(\boldsymbol{x}',s) \, A^\ell(\boldsymbol{x},\boldsymbol{x}';t,s) \tag{24}
$$

where we introduced the following order parameters

$$
\Phi^\ell(\boldsymbol{x},\boldsymbol{x}';t,s) = \frac{1}{N} \phi(\boldsymbol{h}^\ell(\boldsymbol{x},t)) \cdot \phi(\boldsymbol{h}^\ell(\boldsymbol{x}',s)), \; G^\ell(\boldsymbol{x},\boldsymbol{x}';t,s) = \frac{1}{N} \boldsymbol{g}^\ell(\boldsymbol{x},t) \cdot \boldsymbol{g}^\ell(\boldsymbol{x}',s) \tag{25}
$$

$$
A^\ell(\boldsymbol{x},\boldsymbol{x}';t,s) = -\frac{i}{N} \phi(\boldsymbol{h}^\ell(\boldsymbol{x},t)) \cdot \hat{\boldsymbol{\xi}}^\ell(\boldsymbol{x}',s) \tag{26}
$$

As we did with the $\boldsymbol{\chi}, \boldsymbol{\xi}$ fields, we can enforce the definition of the above order parameters with integral representations of Dirac-delta functions. For example, the $\Phi$ kernels can be enforced by inserting

$$
1 = \int \frac{d\Phi^\ell(\boldsymbol{x},\boldsymbol{x}';t,s) d\hat{\Phi}^\ell(\boldsymbol{x},\boldsymbol{x}';t,s)}{2\pi i N^{-1}} \exp \left( N\Phi^\ell(\boldsymbol{x},\boldsymbol{x}';t,s) \hat{\Phi}^\ell(\boldsymbol{x},\boldsymbol{x}';t,s) \right)
$$

$$
\exp \left( -\hat{\Phi}^\ell(\boldsymbol{x},\boldsymbol{x}';t,s) \phi(\boldsymbol{h}^\ell(\boldsymbol{x},t)) \cdot \phi(\boldsymbol{h}^\ell(\boldsymbol{x}',t')) \right) \tag{27}
$$

where the $\hat{\Phi}$ integral is performed along the imaginary axis $(-i\infty, i\infty)$ in the complex plane. The same trick is performed for $G^\ell$ and $A^\ell$ which have conjugate variables $\hat{G}^\ell, B^\ell$ respectively. After inserting these new variables, we find that the moment generating function can be expressed as an integral over a collection of order parameters $\boldsymbol{q}$,

$$
\boldsymbol{q} = \text{Vec}\{\Phi^\ell(\boldsymbol{x},\boldsymbol{x}';t,s), \hat{\Phi}^\ell(\boldsymbol{x},\boldsymbol{x}';t.s), G^\ell(\boldsymbol{x},\boldsymbol{x}',t,s), \hat{G}^\ell(\boldsymbol{x},\boldsymbol{x}',t,s), A^\ell(\boldsymbol{x},\boldsymbol{x}',t,s) B^\ell(\boldsymbol{x},\boldsymbol{x}',t,s)\}, \tag{28}
$$

which is vectorized over all layers $\ell$ time points $t,s$ and samples $\boldsymbol{x}, \boldsymbol{x}'$. The moment generating function takes the form

$$
Z[\{\boldsymbol{j}_h, \boldsymbol{j}_g\}] = \int d\boldsymbol{q} \exp \left( N\mathcal{S}[\boldsymbol{q}, \boldsymbol{j}_h, \boldsymbol{j}_g] \right) \tag{29}
$$

where $\mathcal{S}$ is the $\mathcal{O}_N(1)$ DMFT action which has the following form

$$
\mathcal{S} = \sum_\ell \int d\boldsymbol{x} d\boldsymbol{x}' dt ds \left[ \Phi^\ell(\boldsymbol{x},\boldsymbol{x}';t,s) \hat{\Phi}^\ell(\boldsymbol{x},\boldsymbol{x}';t.s) + G^\ell(\boldsymbol{x},\boldsymbol{x}',t,s) \hat{G}^\ell(\boldsymbol{x},\boldsymbol{x}',t,s) \right]
$$

$$
- \sum_\ell \int d\boldsymbol{x} d\boldsymbol{x}' dt ds \, A^\ell(\boldsymbol{x},\boldsymbol{x}',t,s) B^\ell(\boldsymbol{x}',\boldsymbol{x},s,t) + \frac{1}{N} \sum_{i=1}^N \mathcal{Z}_i[\boldsymbol{q}, \boldsymbol{j}_h, \boldsymbol{j}_g] \tag{30}
$$

The functions $\mathcal{Z}_i$ are *single-site* moment generating functions (MGFs) which express the distribution of $h_i, g_i$ for a given value of the order parameters $\boldsymbol{q}$ and sources $\boldsymbol{j}_h, \boldsymbol{j}_g$.

$$
\mathcal{Z}_i = \int \prod_{\ell,\boldsymbol{x},t} d\hat{\chi}^\ell(\boldsymbol{x},t) d\chi^\ell(\boldsymbol{x},t) d\hat{\chi}^\ell(\boldsymbol{x},t) d\hat{\chi}^\ell(\boldsymbol{x},t) \exp \left( -\sum_\ell \mathcal{H}_i^\ell[\hat{\chi}, \hat{\xi}, \chi, \xi, j_{h,i}, j_{g,i}] \right) \tag{31}
$$

where $\mathcal{H}_i^\ell$ are the single-site Hamiltonians

$$
\begin{aligned}
\mathcal{H}_i^\ell =& \frac{1}{2}\int d\boldsymbol{x}d\boldsymbol{x}'dtds \left[ \Phi^{\ell-1}(\boldsymbol{x},\boldsymbol{x}';t,s)\hat{\chi}^\ell(\boldsymbol{x},t)\hat{\chi}^\ell(\boldsymbol{x}',s) + G^{\ell+1}(\boldsymbol{x},\boldsymbol{x}';t,s)\hat{\xi}^\ell(\boldsymbol{x},t)\hat{\xi}^\ell(\boldsymbol{x}',s) \right] \\
&+ \int d\boldsymbol{x}d\boldsymbol{x}'dtds \left[ \hat{\Phi}^\ell(\boldsymbol{x},\boldsymbol{x}';t,s)\phi(h^\ell(\boldsymbol{x},t))\phi(h^\ell(\boldsymbol{x}',t')) + \hat{G}^\ell(\boldsymbol{x},\boldsymbol{x}';t,s)g^\ell(\boldsymbol{x},t)g^\ell(\boldsymbol{x}',s) \right] \\
&+ i\int d\boldsymbol{x}d\boldsymbol{x}'dtds \left[ A^{\ell-1}(\boldsymbol{x},\boldsymbol{x}';t,s)\hat{\chi}^\ell(\boldsymbol{x},t)g^\ell(\boldsymbol{x}',s) + B^\ell(\boldsymbol{x},\boldsymbol{x}';t,s)\hat{\xi}^\ell(\boldsymbol{x},t)\phi(h^\ell(\boldsymbol{x}',s)) \right] \\
&- \int d\boldsymbol{x}dt \left[ i\hat{\chi}^\ell(\boldsymbol{x},t)\chi^\ell(\boldsymbol{x},t) + i\hat{\xi}^\ell(\boldsymbol{x},t)\xi^\ell(\boldsymbol{x},t) + j_{h,i}^\ell(\boldsymbol{x},t)h^\ell(\boldsymbol{x},t) + j_{g,i}^\ell(\boldsymbol{x},t)g^\ell(\boldsymbol{x},t) \right]
\end{aligned}
\tag{32}
$$

In the above formulas, the $h, g$ fields should be regarded as functions of the $\chi, \xi$ fields.

Since the moment generating function has the form $Z = \int d\boldsymbol{q}\exp\left(N\mathcal{S}[\boldsymbol{q}]\right)$, the $N \to \infty$ limit can thus be characterized by the steepest-descent method (saddle-point integration over $\boldsymbol{q}$). The result is that all of the order parameters take on definite values (concentrate) at infinite width. The saddle point equations $\frac{\partial \mathcal{S}}{\partial \boldsymbol{q}} = 0$ state that each of the kernels takes on definite values

$$
\Phi^\ell(\boldsymbol{x},\boldsymbol{x}';t,s) = \frac{1}{N}\sum_{i=1}^{N}\left\langle \phi(h^\ell(\boldsymbol{x},t))\phi(h^\ell(\boldsymbol{x}',s)) \right\rangle_i
$$

$$
G^\ell(\boldsymbol{x},\boldsymbol{x}';t,s) = \frac{1}{N}\sum_{i=1}^{N}\left\langle g^\ell(\boldsymbol{x},t)g^\ell(\boldsymbol{x}',s) \right\rangle_i
$$

$$
A^\ell(\boldsymbol{x},\boldsymbol{x}';t,s) = -\frac{i}{N}\sum_{i=1}^{N}\left\langle \phi(h^\ell(\boldsymbol{x},t))\hat{\xi}^\ell(\boldsymbol{x}',s) \right\rangle_i
$$

$$
B^\ell(\boldsymbol{x},\boldsymbol{x}';t,s) = -\frac{i}{N}\sum_{i=1}^{N}\left\langle g^{\ell+1}(\boldsymbol{x},t)\hat{\chi}^{\ell+1}(\boldsymbol{x}',s) \right\rangle_i
$$

$$
\hat{\Phi}^\ell(\boldsymbol{x},\boldsymbol{x}';t,s) = -\frac{1}{2N}\sum_{i=1}^{N}\left\langle \hat{\chi}^\ell(\boldsymbol{x},t)\hat{\chi}^\ell(\boldsymbol{x}',s) \right\rangle_i
$$

$$
\hat{G}^\ell(\boldsymbol{x},\boldsymbol{x}';t,s) = -\frac{1}{2N}\sum_{i=1}^{N}\left\langle \hat{\xi}^\ell(\boldsymbol{x},t)\hat{\xi}^\ell(\boldsymbol{x}',s) \right\rangle_i
\tag{33}
$$

where $\langle\rangle_i$ denotes averaging over a *single site stochastic process* defined by the moment generating function $\mathcal{Z}_i$. At zero source $\boldsymbol{j} \to 0$ all of the single site MGF functions $\mathcal{Z}_i$ are identical so all averages $\langle\cdot\rangle_i$ are also identical. At zero source, the kernels have the simple expressions such as $\Phi^\ell(\boldsymbol{x},\boldsymbol{x}';t,s) = \left\langle \phi(h^\ell(\boldsymbol{x},t))\phi(h^\ell(\boldsymbol{x}',s)) \right\rangle$.

### D.3.1 HUBBARD-STRATONOVICH TRANSFORM

Now that we know the kernels $\Phi^\ell, G^\ell$ take on definite values at $N \to \infty$, we can perform a Hubbard-Stratonovich transformation to introduce Gaussian sources

$$
\exp\left( -\frac{1}{2}\int d\boldsymbol{x}d\boldsymbol{x}'dtds \, \hat{\chi}^\ell(\boldsymbol{x},t)\hat{\chi}^\ell(\boldsymbol{x}',t')\Phi^{\ell-1}(\boldsymbol{x},\boldsymbol{x}';t,s) \right)
$$

$$
= \left\langle \exp\left( -i\int d\boldsymbol{x}dt \, \hat{\chi}^\ell(\boldsymbol{x},t)u^\ell(\boldsymbol{x},t) \right) \right\rangle_{u^\ell\sim\mathcal{GP}(0,\Phi^{\ell-1})}
\tag{34}
$$

$$
\exp\left( -\frac{1}{2}\int d\boldsymbol{x}d\boldsymbol{x}'dtds \, \hat{\xi}^\ell(\boldsymbol{x},t)\hat{\xi}^\ell(\boldsymbol{x}',t')G^{\ell+1}(\boldsymbol{x},\boldsymbol{x}';t,s) \right)
$$

$$
= \left\langle \exp\left( -i\int d\boldsymbol{x}dt \, \hat{\xi}^\ell(\boldsymbol{x},t)r^\ell(\boldsymbol{x},t) \right) \right\rangle_{r^\ell\sim\mathcal{GP}(0,G^{\ell+1})}
\tag{35}
$$

After introducing these Gaussian sources $u^\ell, r^\ell$, we have linearized the single site Hamiltonians $\mathcal{H}_i^\ell$ with respect to $\hat{\chi}$ and $\hat{\xi}$. Integration over these variables gives us the following Dirac-Delta function constraints

$$\chi^\ell(\boldsymbol{x}, t) = u^\ell(\boldsymbol{x}, t) + \int d\boldsymbol{x} ds \, A^{\ell-1}(\boldsymbol{x}, \boldsymbol{x}'; t, s) g^\ell(\boldsymbol{x}', s)$$

$$\xi^\ell(\boldsymbol{x}, t) = r^\ell(\boldsymbol{x}, t) + \int d\boldsymbol{x} ds \, B^\ell(\boldsymbol{x}, \boldsymbol{x}'; t, s) \phi(h^\ell(\boldsymbol{x}', s)) \tag{36}$$

Using similar manipulations, we can simplify the expressions for $A^\ell, B^\ell, \hat{\Phi}, \hat{G}$ at zero source $\boldsymbol{j} = 0$

$$A^\ell(\boldsymbol{x}, \boldsymbol{x}'; t, s) = \left\langle \frac{\delta \phi(h^\ell(\boldsymbol{x}, t))}{\delta r^\ell(\boldsymbol{x}', s)} \right\rangle$$

$$B^\ell(\boldsymbol{x}, \boldsymbol{x}'; t, s) = \left\langle \frac{\delta g^{\ell+1}(\boldsymbol{x}, t)}{\delta u^\ell(\boldsymbol{x}', s)} \right\rangle$$

$$\hat{\Phi}^\ell(\boldsymbol{x}, \boldsymbol{x}'; t, s) = 0 \, , \, \hat{G}^\ell(\boldsymbol{x}, \boldsymbol{x}'; t, s) = 0 \tag{37}$$

Plugging the dynamics for $\chi, \xi$ into the formulas for $h^\ell, g^\ell$, we obtain the following final equations for the stochastic process of $h^\ell, g^\ell$.

$$
\begin{aligned}
h^{\ell+1}(\boldsymbol{x}, t) &= h^\ell(\boldsymbol{x}, t) + \frac{1}{\sqrt{L}} u^{\ell+1}(\boldsymbol{x}, t) + \frac{1}{\sqrt{L}} \int d\boldsymbol{x}' ds \, A^\ell(\boldsymbol{x}, \boldsymbol{x}'; t, s) g^{\ell+1}(\boldsymbol{x}', s) \\
&\quad + \frac{\eta_0 \gamma_0}{L} \mathbb{E}_{\boldsymbol{x}'} \int_0^t ds \, \Delta(\boldsymbol{x}', s) \Phi^\ell(\boldsymbol{x}, \boldsymbol{x}'; t, s) \, g^{\ell+1}(\boldsymbol{x}', s) \, , \, u^\ell \sim \mathcal{GP}(0, \Phi^{\ell-1}) \\
g^\ell(\boldsymbol{x}, t) &= g^{\ell+1}(\boldsymbol{x}, t) + \frac{1}{\sqrt{L}} \dot{\phi}(h^\ell(\boldsymbol{x}, t)) z^\ell(\boldsymbol{x}, t) \, , \\
z^\ell(\boldsymbol{x}, t) &= r^\ell(\boldsymbol{x}, t) + \int d\boldsymbol{x}' ds \, B^\ell(\boldsymbol{x}, \boldsymbol{x}'; t, s) \phi(h^\ell(\boldsymbol{x}', s)) \, , \, r^\ell \sim \mathcal{GP}(0, G^{\ell+1}) \\
&\quad + \frac{\eta_0 \gamma_0}{\sqrt{L}} \mathbb{E}_{\boldsymbol{x}'} \int_0^t ds \, \Delta(\boldsymbol{x}', s) G^{\ell+1}(\boldsymbol{x}, \boldsymbol{x}'; t, s) \phi(h^\ell(\boldsymbol{x}', s))
\end{aligned} \tag{38a}
$$

where the order parameters satisfy the the following saddle point equations

$$
\begin{aligned}
\Phi^\ell(\boldsymbol{x}, \boldsymbol{x}'; t, s) &= \left\langle \phi(h^\ell(\boldsymbol{x}, t)) \phi(h^\ell(\boldsymbol{x}', s)) \right\rangle \, , \, G^\ell(\boldsymbol{x}, \boldsymbol{x}'; t, s) = \left\langle g^\ell(\boldsymbol{x}, t) g^\ell(\boldsymbol{x}', s) \right\rangle \\
A^\ell(\boldsymbol{x}, \boldsymbol{x}'; t, s) &= \left\langle \frac{\delta \phi(h^\ell(\boldsymbol{x}, t))}{\delta r^\ell(\boldsymbol{x}', s)} \right\rangle \, , \, B^\ell(\boldsymbol{x}, \boldsymbol{x}'; t, s) = \left\langle \frac{\delta g^{\ell+1}(\boldsymbol{x}, t)}{\delta u^{\ell+1}(\boldsymbol{x}', s)} \right\rangle \\
\frac{df(\boldsymbol{x}, t)}{dt} &= \eta_0 \mathbb{E}_{\boldsymbol{x}'} K(\boldsymbol{x}, \boldsymbol{x}', t) \Delta(\boldsymbol{x}', t) \, , \, \Delta(\boldsymbol{x}, t) = -\frac{\partial \mathcal{L}}{\partial f(\boldsymbol{x}, t)} \\
K(\boldsymbol{x}, \boldsymbol{x}', t) &= \frac{1}{L} \sum_{\ell=1}^L G^{\ell+1}(\boldsymbol{x}, \boldsymbol{x}'; t, s) \Phi^\ell(\boldsymbol{x}, \boldsymbol{x}'; t, s)
\end{aligned} \tag{39a}
$$

Up to the residual skip connections and the factors of $\frac{1}{\sqrt{L}}$, this agrees with the $N \to \infty$ limit DMFT derived in Bordelon & Pehlevan (2022b). At fixed $L$ this above equation can be solved with a self-consistent Monte Carlo sampling procedure. For deep linear networks, the equations close since all $h, g$ fields are Gaussian. In the next section, we analyze the behavior of the above equations at large depth $L$ so that we can take a large depth limit.

# E  THE $L \to \infty$ LIMIT FROM THE DMFT SADDLE POINT EQUATIONS

In this section, we consider how the infinite width dynamics behaves at large depth. We will show that the limiting dynamics takes the form of a system of ODEs for the kernels in terms of layer time

$\tau = \ell/L$. First, we will show that the response functions need to be rescaled with respect to $L$. Then we will take a continuum limit over layers to arrive at the final ODEs for the $\Phi$ and $G$ correlation functions.

### E.1 Unfolding the Layer Recursion

A useful first step which we will use in the following analysis is "unfolding" the recurrence to get explicit formulas for $h^\ell$ in terms of all of the Gaussian sources $u$ and all of the gradient fields $g^\ell$.

$$
\begin{aligned}
h^\ell(\boldsymbol{x}, t) = & h^1(\boldsymbol{x}, t) + \frac{1}{\sqrt{L}} \sum_{k=0}^{\ell} u^k(\boldsymbol{x}, t) + \frac{1}{\sqrt{L}} \sum_{k=1}^{\ell-1} \int d\boldsymbol{x}' ds\, A^k(\boldsymbol{x}, \boldsymbol{x}'; t, s) g^{k+1}(\boldsymbol{x}', s) \\
& + \frac{\eta_0 \gamma_0}{L} \mathbb{E}_{\boldsymbol{x}'} \int_0^t ds\, \Delta(\boldsymbol{x}', s) \sum_{k=0}^{\ell-1} \Phi^k(\boldsymbol{x}, \boldsymbol{x}', t, s) g^{k+1}(\boldsymbol{x}', s)
\end{aligned}
\tag{40}
$$

and similarly for the backward pass, we have

$$
\begin{aligned}
g^\ell(\boldsymbol{x}, t) = & g^L(\boldsymbol{x}, t) + \frac{1}{\sqrt{L}} \sum_{k=\ell}^{L} \dot{\phi}(h^k(\boldsymbol{x}, t)) r^k(\boldsymbol{x}, t) \\
& + \frac{1}{\sqrt{L}} \sum_{k=\ell}^{L} \dot{\phi}(h^k(\boldsymbol{x}, t)) \int d\boldsymbol{x}' ds\, B^k(\boldsymbol{x}, \boldsymbol{x}'; t, s) \phi(h^k(\boldsymbol{x}', s)) \\
& + \frac{\eta_0 \gamma_0}{L} \sum_{k=\ell}^{L} \dot{\phi}(h^k(\boldsymbol{x}, t)) \mathbb{E}_{\boldsymbol{x}'} \int_0^t ds\, \Delta(\boldsymbol{x}', s) \sum_{k=1}^{\ell-1} G^k(\boldsymbol{x}, \boldsymbol{x}', t, s) \phi(h^k(\boldsymbol{x}', s))
\end{aligned}
\tag{41}
$$

A very useful fact that can be immediately gleaned from the above equations is that

$$
\begin{aligned}
\forall k \leq \ell\,, \quad & \frac{\delta h^\ell(\boldsymbol{x}, t)}{\delta u^k(\boldsymbol{x}, t)} \sim \mathcal{O}\left(\frac{1}{\sqrt{L}}\right) \\
\forall k \geq \ell\,, \quad & \frac{\delta g^\ell(\boldsymbol{x}, t)}{\delta r^k(\boldsymbol{x}, t)} \sim \mathcal{O}\left(\frac{1}{\sqrt{L}}\right)
\end{aligned}
\tag{42}
$$

These facts will be utilized to obtain the scale of the response functions in the large depth limit.

### E.2 Characterizing Response Functions at Large Depth

First, we will study the response functions $A^\ell, B^\ell$. We note that from the saddle point equations,

$$
A^\ell(\boldsymbol{x}, \boldsymbol{x}'; t, s) = \left\langle \frac{\delta \phi(h^\ell(\boldsymbol{x}, t'))}{\delta r^\ell(\boldsymbol{x}', s)} \right\rangle .
\tag{43}
$$

Since $\frac{\delta \phi(h)}{\delta r} = \dot{\phi}(h) \frac{\delta h}{\delta r}$, it suffices to consider the scale of the derivative $\frac{\delta h^\ell(\boldsymbol{x}, t')}{\delta r(\boldsymbol{x}', s)}$

$$
\begin{aligned}
A^\ell(\boldsymbol{x}, \boldsymbol{x}'; t, s) = & \frac{1}{\sqrt{L}} \sum_{k=1}^{\ell} \int d\boldsymbol{x}'' dt'\, A^k(\boldsymbol{x}, \boldsymbol{x}''; t, t') \left\langle \dot{\phi}(h^\ell(\boldsymbol{x}, t)) \frac{\delta g^{k+1}(\boldsymbol{x}'', t')}{\delta r^\ell(\boldsymbol{x}', s)} \right\rangle \\
& + \frac{\eta_0 \gamma_0}{L} \sum_{k=0}^{\ell-1} \mathbb{E}_{\boldsymbol{x}'} \int_0^t dt' \Delta(\boldsymbol{x}', s) \Phi^k(\boldsymbol{x}, \boldsymbol{x}'; t, s) \left\langle \dot{\phi}(h^\ell(\boldsymbol{x}, t)) \frac{\delta g^k(\boldsymbol{x}'', t')}{\delta r^\ell(\boldsymbol{x}', s)} \right\rangle
\end{aligned}
\tag{44}
$$

Next, we use the fact that $\frac{\delta g^k(\boldsymbol{x}'', t')}{\delta r^\ell(\boldsymbol{x}', s)} \sim \mathcal{O}(L^{-1/2})$. Therefore, the second sum (the one which depends on $\eta_0 \gamma_0 L^{-1}$) scales as $\mathcal{O}(\eta_0 \gamma_0 L^{-1/2})$. Solving the $A^\ell$ recursion iteratively from the first layer, we arrive at the conclusion that

$$
A^\ell(\boldsymbol{x}, \boldsymbol{x}'; t, s) \sim \mathcal{O}(\eta_0 \gamma_0 L^{-1/2}).
\tag{45}
$$

Indeed, as a sanity check, plugging this in gives a consistent solution to the above equation. Repeating an identical argument for the backward pass reveals an identical scaling for $B^\ell$ with depth $L$

$$B^\ell(\boldsymbol{x}, \boldsymbol{x}'; t, s) \sim \mathcal{O}(\eta_0 \gamma_0 L^{-1/2}) \tag{46}$$

We therefore need to introduce a rescaled version of the response functions

$$A^\ell(\boldsymbol{x}, \boldsymbol{x}', t, s) \leftarrow \frac{\sqrt{L}}{\eta_0 \gamma_0} A^\ell(\boldsymbol{x}, \boldsymbol{x}', t, s) \tag{47}$$

$$B^\ell(\boldsymbol{x}, \boldsymbol{x}', t, s) \leftarrow \frac{\sqrt{L}}{\eta_0 \gamma_0} B^\ell(\boldsymbol{x}, \boldsymbol{x}', t, s) \tag{48}$$

After this rescaling, we have

$$A^\ell(\boldsymbol{x}, \boldsymbol{x}', t, s) = \frac{\sqrt{L}}{\eta_0 \gamma_0} \left\langle \frac{\delta \phi(h^\ell(\boldsymbol{x}, t))}{\delta r^\ell(\boldsymbol{x}', s)} \right\rangle \tag{49}$$

To calculate this rescaled response function, we can derive a closed set of equations for all of the cross layer correlators

$$\frac{\delta h^\ell(\boldsymbol{x}, t)}{\delta r^{\ell'}(\boldsymbol{x}', s)} = \frac{\eta_0 \gamma_0}{L} \sum_{k=1}^{\ell} \int d\boldsymbol{x}'' ds\, C_h^k(\boldsymbol{x}, \boldsymbol{x}'', t, s) \frac{\delta g^k(\boldsymbol{x}'', s)}{\delta r^{\ell'}(\boldsymbol{x}', s)}$$

$$\frac{\delta g^\ell(\boldsymbol{x}, t)}{\delta r^{\ell'}(\boldsymbol{x}', s)} = \frac{1}{\sqrt{L}} \Theta(\ell' - \ell) \delta(\boldsymbol{x} - \boldsymbol{x}') \delta(t - s) \dot{\phi}(h^{\ell'}(\boldsymbol{x}, t))$$

$$+ \frac{1}{\sqrt{L}} \sum_{k=\ell}^{L} r^k(\boldsymbol{x}, t) \frac{\delta \dot{\phi}(h^k(\boldsymbol{x}, t))}{\delta r^{\ell'}(\boldsymbol{x}', s)}$$

$$+ \frac{\eta_0 \gamma_0}{L} \sum_{k=\ell}^{L} \dot{\phi}(h^k(\boldsymbol{x}, t)) \int d\boldsymbol{x}'' ds C_g^k(\boldsymbol{x}, \boldsymbol{x}'', t, s) \frac{\delta \phi(h^k(\boldsymbol{x}'', s))}{\delta r^{\ell'}(\boldsymbol{x}', s)}$$

$$+ \frac{\eta_0 \gamma_0}{L} \sum_{k=\ell}^{L} \frac{\delta \dot{\phi}(h^k(\boldsymbol{x}, t))}{\delta r^{\ell'}(\boldsymbol{x}', s)} \int d\boldsymbol{x}'' ds C_g^k(\boldsymbol{x}, \boldsymbol{x}'', t, s) \phi(h^k(\boldsymbol{x}'', s)) \tag{50}$$

where we introduced the shorthand

$$C_h^\ell(\boldsymbol{x}, \boldsymbol{x}', t, s) = A^{\ell-1}(\boldsymbol{x}, \boldsymbol{x}', t, s) + \Phi^{\ell-1}(\boldsymbol{x}, \boldsymbol{x}', t, s) \Delta(\boldsymbol{x}', s) p(\boldsymbol{x}')$$

$$C_h^\ell(\boldsymbol{x}, \boldsymbol{x}', t, s) = B^\ell(\boldsymbol{x}, \boldsymbol{x}', t, s) + G^{\ell+1}(\boldsymbol{x}, \boldsymbol{x}', t, s) \Delta(\boldsymbol{x}', s) p(\boldsymbol{x}') \tag{51}$$

Similar equations exist for all derivatives with respect to $u^\ell(\boldsymbol{x}, t)$. Once the above equations are solved, then one can compute the response functions $A^\ell$, $B^\ell$ by averaging the necessary field derivatives.

### E.3 Continuum Layer-time Limit at Infinite Depth

In this section, we argue the infinite depth limit can be viewed as a continuum limiting process defined by ODEs for the kernels and SDEs for the preactivation and gradient fields for a layertime $\tau = \ell/L$. We start by examining the asymptotic structure of the preactivation $h^\ell$ dynamics

$$h^\ell(\boldsymbol{x}; t) \sim h^1(\boldsymbol{x}; t) + \frac{1}{\sqrt{L}} \sum_{k=1}^{\ell} u^k(\boldsymbol{x}; t)$$

$$+ \frac{\eta_0 \gamma_0}{L} \sum_{k=0}^{\ell-1} \int_0^t ds\, \mathbb{E}_{\boldsymbol{x}'} C_h^k(\boldsymbol{x}, \boldsymbol{x}'; t, s)\, g^{k+1}(\boldsymbol{x}', s) \Delta(\boldsymbol{x}'; s) \tag{52}$$

The Gaussian source variables $u^k$ are all uncorrelated across layers. So the first sum is also a mean zero Gaussian with variance

$$\left\langle \left( \frac{1}{\sqrt{L}} \sum_{k=1}^{\ell} u^k(\boldsymbol{x}, t) \right) \left( \frac{1}{\sqrt{L}} \sum_{k=1}^{\ell} u^k(\boldsymbol{x}', s) \right) \right\rangle = \frac{1}{L} \sum_{k=1}^{\ell} \Phi^{k-1}(\boldsymbol{x}, \boldsymbol{x}'; t, s) \sim \mathcal{O}(1) \tag{53}$$

At large depth $L$, this term will behave as integrated Brownian motion since each layer's contribution is independent and order $\sqrt{d\tau} \sim \frac{1}{\sqrt{L}}$. Next, we reason about the scale of feature learning update

$$
\frac{\eta_0^2 \gamma_0^2}{L^2} \sum_{k=0}^{\ell-1} \sum_{k'=0}^{\ell-1} \int_0^t ds' \int_0^t ds \, \mathbb{E}_{\boldsymbol{x}', \boldsymbol{x}''} \, \Delta(\boldsymbol{x}'; s) \Delta(\boldsymbol{x}''; s')
$$
$$
\times C_h^k(\boldsymbol{x}, \boldsymbol{x}'; t, s) \, C_h^{k'}(\boldsymbol{x}, \boldsymbol{x}'; t, s) \, \left\langle g^{k+1}(\boldsymbol{x}', s) g^{k'+1}(\boldsymbol{x}'', s') \right\rangle \sim \mathcal{O}(1) \tag{54}
$$

The above double sum is $\mathcal{O}(1)$ because we have $\left\langle g^k g^{k'} \right\rangle \sim \mathcal{O}(1)$ for all $k, k'$. This fact arises from the skip connections, which generate *long-range correlations* across layers. This behavior is very different from non-residual networks where $g^\ell$ and $g^{\ell'}$ are independent for all $\ell \neq \ell'$ (Bordelon & Pehlevan, 2022b).

To take the continuum limit, we define functions of a continuous layer time variable $\tau \in [0, 1]$,

$$
h(\tau, \boldsymbol{x}, t) = h^\ell(\boldsymbol{x}, t)|_{\ell=\tau L} \, , \, g(\tau, \boldsymbol{x}, t) = g^\ell(\boldsymbol{x}, t)|_{\ell=\tau L}
$$
$$
\Phi(\tau; \boldsymbol{x}, \boldsymbol{x}'; t, s) = \Phi^\ell(\boldsymbol{x}, \boldsymbol{x}'; t, s)|_{\ell=\tau L} \, , \, G(\tau; \boldsymbol{x}, \boldsymbol{x}'; t, s) = G^\ell(\boldsymbol{x}, \boldsymbol{x}'; t, s)|_{\ell=\tau L}. \tag{55}
$$

The key idea when taking the continuum limit is the following relations based on a notion an infinitesimal layer increment $d\tau \sim \frac{1}{L}$. Let $F^\ell$ be a deterministic function of the layer index and let $F(\tau)$ be its continuous analogue which satisfies $F(\tau) = F^\ell|_{\ell=\tau L}$. Then,

$$
\lim_{L \to \infty} \frac{1}{L} \sum_{\ell=1}^{\tau L} F^\ell \to \int_0^\tau d\tau' F(\tau)
$$
$$
\lim_{L \to \infty} L \left[ F^{\tau L+1} - F^{\tau L} \right] \to \frac{\partial}{\partial \tau} F(\tau). \tag{56}
$$

We will use these relations when deriving continuum limits of the DMFT process. We note that standard error analysis for Euler discretizations can be used to argue that finite depth averages converge to the $\tau$ integral at rate

$$
\left| \frac{1}{L} \sum_{\ell=1}^{\tau L} F^\ell - \int_0^\tau d\tau' F(\tau') \right| \sim \mathcal{O}(L^{-1}) \tag{57}
$$

This is merely the approximation error of a Riemann sum which is proportional to the step size (Rudin, 1953), which in this case is $1/L$. This idea can be used to generate a convergence rate of the NTK and the network output predictions at large depth as we discuss in I.

### E.4 PREACTIVATION DYNAMICS

As discussed in the main text, the large $L$ limit for the dynamics of $h^\ell$ can be viewed as a stochastic integral over $\tau$ and over Brownian motion for the $u^k$ sum

$$
h(\tau; \boldsymbol{x}, t) = h(0; \boldsymbol{x}, t) + \int_0^\tau du(\tau', \boldsymbol{x}, t)
$$
$$
+ \eta_0 \gamma_0 \int d\boldsymbol{x}' \int_0^t ds \int_0^\tau d\tau' \left[ \Phi(\tau'; \boldsymbol{x}, \boldsymbol{x}'; t, s) \Delta(\boldsymbol{x}'; s) p(\boldsymbol{x}') + A(\tau', \boldsymbol{x}, \boldsymbol{x}'; t, s) \right] g(\tau', \boldsymbol{x}', s) \tag{58}
$$

where $du$ is Brownian motion term with covariance structure

$$
\langle du(\tau, \boldsymbol{x}, t) du(\tau', \boldsymbol{x}', s) \rangle = \delta(\tau - \tau') \Phi(\tau, \boldsymbol{x}, \boldsymbol{x}', t, s) d\tau. \tag{59}
$$

Similarly, the gradient fields have an analogous structure

$$
g(\tau; \boldsymbol{x}, t) = g(\tau; \boldsymbol{x}, t) + \int_\tau^1 dr(\tau', \boldsymbol{x}, t) \dot{\phi}(h(\tau', \boldsymbol{x}, t))
$$
$$
+ \eta_0 \gamma_0 \mathbb{E}_{\boldsymbol{x}'} \int_0^t ds \int_\tau^1 d\tau' \dot{\phi}(h(\tau', \boldsymbol{x}, t)) G(\tau'; \boldsymbol{x}, \boldsymbol{x}'; t, s) \Delta(\boldsymbol{x}'; s) \phi(h(\tau'; \boldsymbol{x}', s))
$$
$$
+ \eta_0 \gamma_0 \int d\boldsymbol{x}' \int_0^t ds \int_\tau^1 d\tau' \dot{\phi}(h(\tau', \boldsymbol{x}, t)) B(\tau', \boldsymbol{x}, \boldsymbol{x}'; t, s) \phi(h(\tau'; \boldsymbol{x}', s)). \tag{60}
$$

where $dr$ is Brownian motion with covariance structure

$$\langle dr(\tau, \boldsymbol{x}, t) dr(\tau', \boldsymbol{x}', s) \rangle = \delta(\tau - \tau') G(\tau, \boldsymbol{x}, \boldsymbol{x}', t, s) d\tau. \tag{61}$$

We note that, due to the nonlinearities, the $h, g$ fields are *non-Gaussian* at finite $\gamma_0$ as in the case of mean field networks Bordelon & Pehlevan (2022b). We note that an alternative to the above integral formulation is a stochastic differential equations

$$dh(\tau, \boldsymbol{x}, t) = du(\tau, \boldsymbol{x}, t)$$
$$+ d\tau \, \eta_0 \gamma_0 \int d\boldsymbol{x}' \int_0^t ds \, [\Phi(\tau; \boldsymbol{x}, \boldsymbol{x}'; t, s) \Delta(\boldsymbol{x}'; s) p(\boldsymbol{x}') + A(\tau; \boldsymbol{x}, \boldsymbol{x}'; t, s)] g(\tau, \boldsymbol{x}', s) \tag{62}$$

The $\gamma_0 \to 0$ limit of the $h$ equation at $t, s = 0$ coincides with the SDE derived at initialization by Hayou (2023); Hayou & Yang (2023). However, we now see that feature learning term (the term which depends on $\gamma_0$) adds a drift term to the layerwise dynamics compared to the diffusion term from the initial weights.

## E.5 KERNEL LAYERWISE ODES IN $\tau$ IN FEATURE LEARNING REGIME

We can start from the stochastic difference equation

$$dh(\tau, \boldsymbol{x}, t) = du(\tau, \boldsymbol{x}', t) + d\tau \, \eta_0 \gamma_0 \int d\boldsymbol{x}' \int_0^t ds \, C_h(\tau, \boldsymbol{x}, \boldsymbol{x}', t, s) g(\tau, \boldsymbol{x}', s)$$
$$C_h(\tau, \boldsymbol{x}, \boldsymbol{x}', t, s) = A(\tau, \boldsymbol{x}, \boldsymbol{x}', t, s) + \Phi(\tau, \boldsymbol{x}, \boldsymbol{x}', t, s) \Delta(\boldsymbol{x}', s) p(\boldsymbol{x}'). \tag{63}$$

and derive an ODE for the feature kernel at layer-time $\tau$. Applying Ito's lemma (Itô, 1944), we obtain the following feature kernel dynamics

$$\frac{\partial}{\partial \tau} \Phi(\tau, \boldsymbol{x}, \boldsymbol{x}', t, s) =$$
$$\frac{1}{2} \int d\tilde{\boldsymbol{x}} d\tilde{\boldsymbol{x}}' dt' ds' \Phi(\tau, \tilde{\boldsymbol{x}}, \tilde{\boldsymbol{x}}', t', s') \left\langle \frac{\partial^2}{\partial h(\tau, \tilde{\boldsymbol{x}}, t') \partial h(\tau, \tilde{\boldsymbol{x}}', s')} [\phi(h(\tau, \boldsymbol{x}, t)) \phi(h(\tau, \boldsymbol{x}', s))] \right\rangle$$
$$+ \eta_0 \gamma_0 \int d\boldsymbol{x}'' dt' C_h(\tau, \boldsymbol{x}, \boldsymbol{x}'', t') \left\langle g(\tau, \boldsymbol{x}'', t') \dot{\phi}(h(\tau, \boldsymbol{x}, t)) \phi(h(\tau, \boldsymbol{x}', s)) \right\rangle$$
$$+ \eta_0 \gamma_0 \int d\boldsymbol{x}'' ds' C_h(\tau, \boldsymbol{x}', \boldsymbol{x}'', s') \left\langle g(\tau, \boldsymbol{x}'', s') \phi(h(\tau, \boldsymbol{x}, t)) \dot{\phi}(h(\tau, \boldsymbol{x}', s)) \right\rangle \tag{64}$$

The first term on the right hand side comes from signal propagation, which persists even in the lazy limit. The second set of terms come from feature learning corrections. This equation should be integrated from $\tau = 0$ to $\tau = 1$. Similarly, there is a backward pass ODE for the $G$ kernel

$$\frac{\partial}{\partial \tau} G(\tau, \boldsymbol{x}, \boldsymbol{x}', t, s) = - \left\langle \dot{\phi}(h(\tau, \boldsymbol{x}, t)) \dot{\phi}(h(\tau, \boldsymbol{x}', s)) \right\rangle G(\tau, \boldsymbol{x}, \boldsymbol{x}', t, s)$$
$$- \eta_0 \gamma_0 \int d\boldsymbol{x}'' \int_0^{t'} dt' C_g(\tau, \boldsymbol{x}, \boldsymbol{x}'', t, t') \left\langle \dot{\phi}(h(\tau, \boldsymbol{x}, t)) \phi(h(\tau, \boldsymbol{x}', s)) g(\tau, \boldsymbol{x}'', t') \right\rangle$$
$$- \eta_0 \gamma_0 \int d\boldsymbol{x}'' \int_0^{s'} ds' C_g(\tau, \boldsymbol{x}, \boldsymbol{x}'', t, t') \left\langle \phi(h(\tau, \boldsymbol{x}, t)) \dot{\phi}(h(\tau, \boldsymbol{x}', s)) g(\tau, \boldsymbol{x}'', s') \right\rangle$$
$$C_g(\tau, \boldsymbol{x}, \boldsymbol{x}', t, s) = B(\tau, \boldsymbol{x}, \boldsymbol{x}', t, s) + G(\tau, \boldsymbol{x}, \boldsymbol{x}', t, s) \Delta(\boldsymbol{x}', s) p(\boldsymbol{x}') \tag{65}$$

which should be integrated from $\tau = 1$ to $\tau = 0$.

We note again, that convergence to the ODE will occur at rate $\mathcal{O}(L^{-1})$ in absolute error.

## E.6 RESPONSE FUNCTION EQUATIONS IN THE CONTINUUM LIMIT

The response functions can also be determined in the $L \to \infty$ limit using the following relations

$$A(\tau, \boldsymbol{x}, \boldsymbol{x}', t, s) = \frac{1}{\eta_0 \gamma_0} \left\langle \frac{\delta \phi(h(\tau, \boldsymbol{x}, t))}{\delta \, dr(\tau, \boldsymbol{x}', s)} \right\rangle \, , \, B(\tau, \boldsymbol{x}, \boldsymbol{x}', t, s) = \frac{1}{\eta_0 \gamma_0} \left\langle \frac{\delta g(\tau, \boldsymbol{x}, t)}{\delta \, du(\tau, \boldsymbol{x}', s)} \right\rangle \tag{66}$$

The reason that the variation is computed with respect to the differential Brownian motion $du, dr$ is to match the scale of the increment $\frac{1}{\sqrt{L}}$, which appears in the finite $L$ response functions $A^\ell = \frac{\sqrt{L}}{\eta_0 \gamma_0} \left\langle \frac{\delta\phi(h^\ell)}{\delta r^\ell} \right\rangle$. Using our continuum limit equations, we obtain the following field derivatives

$$\frac{\delta h(\tau, \boldsymbol{x}, t)}{\delta du(\tau', \boldsymbol{x}', s)} = \Theta(\tau - \tau')\delta(\boldsymbol{x} - \boldsymbol{x}')\delta(t - s)$$

$$+ \eta_0 \gamma_0 \int d\tau'' \int d\boldsymbol{x}'' \int dt' C_h(\tau'', \boldsymbol{x}, \boldsymbol{x}'', t, t') \frac{\delta g(\tau'', \boldsymbol{x}'', t')}{\delta du(\tau', \boldsymbol{x}', s)}$$

$$\frac{\delta g(\tau, \boldsymbol{x}, t)}{\delta du(\tau', \boldsymbol{x}', s)} = \eta_0 \gamma_0 \int d\tau'' \dot\phi(h(\tau'', \boldsymbol{x}, t)) \int d\boldsymbol{x}'' \int dt' C_g(\tau'', \boldsymbol{x}, \boldsymbol{x}'', t, t') \frac{\delta\phi(h(\tau'', \boldsymbol{x}'', t'))}{\delta du(\tau', \boldsymbol{x}', s)}$$

$$\frac{\delta h(\tau, \boldsymbol{x}, t)}{\delta dr(\tau', \boldsymbol{x}', s)} = \eta_0 \gamma_0 \int d\tau'' \int d\boldsymbol{x}'' \int dt' C_h(\tau'', \boldsymbol{x}, \boldsymbol{x}'', t, t') \frac{\delta g(\tau'', \boldsymbol{x}'', t')}{\delta dr(\tau', \boldsymbol{x}', s)}$$

$$\frac{\delta g(\tau, \boldsymbol{x}, t)}{\delta dr(\tau', \boldsymbol{x}', s)} = \Theta(\tau' - \tau)\delta(\boldsymbol{x} - \boldsymbol{x}')\delta(t - s)\,\dot\phi(h(\tau, \boldsymbol{x}, t))$$

$$+ \eta_0 \gamma_0 \int d\tau'' \dot\phi(h(\tau'', \boldsymbol{x}, t)) \int d\boldsymbol{x}'' \int dt' C_h(\tau'', \boldsymbol{x}, \boldsymbol{x}'', t, t') \frac{\delta\phi(h(\tau'', \boldsymbol{x}'', t'))}{\delta dr(\tau', \boldsymbol{x}', s)} \tag{67}$$

Once these recursions are solved, the response functions can be computed by averaging the necessary field derivatives at each layer time $\tau$

$$A(\tau, \boldsymbol{x}, \boldsymbol{x}', t, s) = \left\langle \frac{\delta\phi(h(\tau, \boldsymbol{x}, t))}{\delta dr(\tau, \boldsymbol{x}', s)} \right\rangle \;,\; B(\tau, \boldsymbol{x}, \boldsymbol{x}', t, s) = \left\langle \frac{\delta g(\tau, \boldsymbol{x}, t)}{\delta du(\tau, \boldsymbol{x}', s)} \right\rangle. \tag{68}$$

### E.7 Final Result: Infinite Width and Depth Limit

In this section we combine the results obtained in the preceeding sections to state our main theoretical result which is the infinite width and depth limit of the learning dynamics in our ResNet. Below we state the result for gradient flow, but minor modifications can also characterize discrete time SGD, momentum, weight decay etc as we discuss in Appendices K.

**Proposition 3** *Consider the $N, L \to \infty$ limit of the ResNet $\mu P$-$\frac{1}{\sqrt{L}}$ model presented in the main text trained with gradient flow on data distribution $p(\boldsymbol{x})$. The preactivation $h(\tau, \boldsymbol{x}, t)$ and gradient $g(\tau, \boldsymbol{x}, t)$ stochastic fields obey the equations*

$$h(\tau, \boldsymbol{x}, t) = h(0, \boldsymbol{x}, t) + \int_0^\tau du(\tau', \boldsymbol{x}, t) + \eta_0 \gamma_0 \int_0^\tau d\tau' \int d\boldsymbol{x}' \int_0^t ds\, C_h(\tau', \boldsymbol{x}, \boldsymbol{x}', t, s) g(\tau', \boldsymbol{x}', s)$$

$$g(\tau, \boldsymbol{x}, t) = g(1, \boldsymbol{x}, t) + \int_\tau^1 \dot\phi(h(\tau', \boldsymbol{x}, t)) dr(\tau', \boldsymbol{x}, t)$$

$$+ \eta_0 \gamma_0 \int_0^\tau d\tau' \dot\phi(h(\tau', \boldsymbol{x}, t)) \int d\boldsymbol{x}' \int_0^t ds\, C_g(\tau', \boldsymbol{x}, \boldsymbol{x}', t, s) \phi(h(\tau', \boldsymbol{x}', s)) \tag{69}$$

*where the zero mean Brownian motions $du(\tau, \boldsymbol{x}, t), dr(\tau, \boldsymbol{x}, t)$ have the following covariance structure*

$$\langle du(\tau, \boldsymbol{x}, t) du(\tau', \boldsymbol{x}', s) \rangle = \Phi(\tau, \boldsymbol{x}, \boldsymbol{x}', t, s)\delta(\tau - \tau')d\tau$$

$$\langle dr(\tau, \boldsymbol{x}, t) dr(\tau', \boldsymbol{x}', s) \rangle = G(\tau, \boldsymbol{x}, \boldsymbol{x}', t, s)\delta(\tau - \tau')d\tau, \tag{70}$$

*the deterministic kernels $C_h$ and $C_g$ have the form*

$$C_h(\tau, \boldsymbol{x}, \boldsymbol{x}', t, s) = \Phi(\tau, \boldsymbol{x}, \boldsymbol{x}', t, s)\Delta(\boldsymbol{x}', s)p(\boldsymbol{x}') + A(\tau, \boldsymbol{x}, \boldsymbol{x}', t, s)$$

$$C_g(\tau, \boldsymbol{x}, \boldsymbol{x}', t, s) = G(\tau, \boldsymbol{x}, \boldsymbol{x}', t, s)\Delta(\boldsymbol{x}', s)p(\boldsymbol{x}') + B(\tau, \boldsymbol{x}, \boldsymbol{x}', t, s) \tag{71}$$

*where the kernels $\Phi, G$ and response functions $A, B$ are computed as averages over the $h, g$ stochastic process*

$$\Phi(\tau, \boldsymbol{x}, \boldsymbol{x}', t, s) = \langle \phi(h(\tau, \boldsymbol{x}, t))\phi(h(\tau, \boldsymbol{x}', s)) \rangle \;,\; G(\tau, \boldsymbol{x}, \boldsymbol{x}', t, s) = \langle g(\tau, \boldsymbol{x}, t)g(\tau, \boldsymbol{x}', s) \rangle$$

$$A(\tau, \boldsymbol{x}, \boldsymbol{x}', t, s) = \frac{1}{\eta_0 \gamma_0} \left\langle \frac{\delta\phi(h(\tau, \boldsymbol{x}, t))}{\delta dr(\tau, \boldsymbol{x}', s)} \right\rangle \;,\; B(\tau, \boldsymbol{x}, \boldsymbol{x}', t, s) = \frac{1}{\eta_0 \gamma_0} \left\langle \frac{\delta g(\tau, \boldsymbol{x}, t)}{\delta du(\tau, \boldsymbol{x}', s)} \right\rangle. \tag{72}$$

*From the feature and gradient kernels, we can compute the dynamical NTK and the dynamics of network predictions*

$$K(\boldsymbol{x}, \boldsymbol{x}', t) = \int_0^1 d\tau \, G(\tau, \boldsymbol{x}, \boldsymbol{x}', t, t) \Phi(\tau, \boldsymbol{x}, \boldsymbol{x}', t, t)$$

$$\frac{d}{dt} f(\boldsymbol{x}, t) = \eta_0 \int d\boldsymbol{x}' p(\boldsymbol{x}') K(\boldsymbol{x}, \boldsymbol{x}', t) \Delta(\boldsymbol{x}', t) \, , \, \Delta(\boldsymbol{x}, t) = -\frac{\partial \mathcal{L}}{\partial f(\boldsymbol{x}, t)}. \tag{73}$$

*This is a closed system of equations relating preactiation and gradient marginals, the kernel dynamics and the network prediction dynamics. In this limit, the kernels, response functions and network outputs are all deterministic.*

## F  INFINITE DEPTH THEN INFINITE WIDTH LIMIT

In this section, we compute the limiting process in the other direction, by directly working with the distribution of order parameters $\boldsymbol{q}$ at finite $N$. We show that at large depth $L$ the DMFT action converges to a limiting object $\mathcal{S}_\infty(\boldsymbol{q})$. We the show that the large $N$ limit of the distribution recovers the same saddle point derived in Appendix E for the large $N$ then large $L$ limit. Thus, this section constitutes a theoretical argument that the width and depth limits commute for the whole course of training dynamics. The derivations are performed at the level of rigor of physics, and we leave open for future work a rigorous result about commutativity of limits throughout training.

### F.1  LARGE $L$ LIMIT AT FIXED $N$

We derived the finite $N, L$ distribution of order parameters in Appendix Section D. The result was that the log-density of the order parameters $\boldsymbol{q}$ over random initialization of network weights was given by the DMFT action $\ln p(\boldsymbol{q}) \propto N\mathcal{S}_L(\boldsymbol{q})$. Suppressing unnecessary (for the purposes of this section) sample and time indices, the action at depth $\mathcal{S}_L$ had the form

$$\mathcal{S}_L = \sum_\ell \left[ \Phi^\ell \hat{\Phi}^\ell + G^\ell \hat{G}^\ell - A^\ell B^\ell \right] + \mathcal{Z}[\boldsymbol{q}] \tag{74}$$

where the single site density $\mathcal{Z}$ has the form

$$\mathcal{Z} = \int \prod_\ell dh^\ell d\hat{h}^\ell dg^\ell d\hat{g}^\ell$$

$$\mathbb{E}_{\{u^\ell, r^\ell\}} \exp \left( i \sum_\ell \hat{h}^\ell \left( h^{\ell+1} - h^\ell - \frac{1}{\sqrt{L}} u^\ell - \frac{1}{\sqrt{L}} A^\ell g^{\ell+1} - \frac{\eta\gamma}{L} \Phi^\ell \Delta g^{\ell+1} \right) \right)$$

$$\exp \left( i \sum_\ell \hat{g}^\ell \left( g^\ell - g^{\ell+1} - \frac{1}{\sqrt{L}} r^\ell \dot{\phi}(h^\ell) - \frac{1}{\sqrt{L}} \dot{\phi}(h^\ell) B^\ell \phi(h^\ell) - \frac{\eta_0\gamma_0}{L} \dot{\phi}(h^\ell) G^{\ell+1} \Delta\phi(h^\ell) \right) \right)$$

$$\exp \left( - \sum_\ell \phi(h^\ell) \hat{\Phi}^\ell \phi(h^\ell) - \sum_\ell g^\ell \hat{G}^\ell g^\ell \right) \tag{75}$$

where the averages are over $u^\ell, r^\ell$ which are Gaussians with covariances $\Phi^{\ell-1}$ and $G^{\ell+1}$ respectively. In the above equation we work with $h, g$ directly rather than working with the intermediate fields $\chi, \xi$ which only depend on the initialization.

To obtain a proper large depth limit, we rescale the conjugate order parameters as

$$\hat{\Phi}^\ell \rightarrow \frac{1}{L} \hat{\Phi}^\ell \, , \, \hat{G}^\ell \rightarrow \frac{1}{L} \hat{G}^\ell \, , \, A^\ell \rightarrow \frac{\eta_0\gamma_0}{\sqrt{L}} A^\ell \, , \, B^\ell \rightarrow \frac{\eta_0\gamma_0}{\sqrt{L}} B^\ell \tag{76}$$

We note that such a rescaling will not alter the distribution of real network observables $\{\Phi, G, K, f, ...\}$ (Bordelon & Pehlevan, 2022b; 2023). After this rescaling, we take $L \rightarrow \infty$ which gives a continuum limit of the resulting layer sums, we obtain the following simplified infinite depth

action in terms of the kernels and features at layer time $\tau \in [0, 1]$

$$\mathcal{S}_\infty = \int d\tau [\Phi(\tau)\hat{\Phi}(\tau) + G(\tau)\hat{G}(\tau) - \eta_0^2\gamma_0^2 A(\tau)B(\tau)] + \ln \mathcal{Z}$$

$$\mathcal{Z} = \int \mathcal{D}\hat{h}\mathcal{D}\hat{g}\mathcal{D}h\mathcal{D}g \, \mathbb{E}_{u(\tau),r(\tau)} \exp\left(i \int \hat{h}(\tau)[dh(\tau) - du(\tau) - \eta_0\gamma_0 C_h(\tau)g(\tau)d\tau]\right)$$

$$\exp\left(i \int \hat{g}(\tau)[-dg(\tau) - dr(\tau)\dot{\phi}(h(\tau)) - \eta_0\gamma_0\dot{\phi}(h(\tau))C_g(\tau)\phi(h(\tau))d\tau]\right)$$

$$\exp\left(-\int d\tau[\phi(h(\tau))\hat{\Phi}(\tau)\phi(h(\tau)) + g(\tau)\hat{G}(\tau)g(\tau)]\right). \tag{77}$$

where $C_h, C_g$ are given in the main text and the $du(\tau), dr(\tau)$ are Brownian motion processes over layer time $\tau$. We therefore find that the distribution of order parameters at finite width $N$ and infinite depth $L \to \infty$ is

$$\lim_{L \to \infty} p_L(\boldsymbol{q}) = p_\infty(\boldsymbol{q}) \propto \exp\left(N\mathcal{S}_\infty(\boldsymbol{q})\right) \tag{78}$$

This constitutes taking the infinite depth limit $L \to \infty$ at fixed $N$. At this stage, the order parameters do not obey a deterministic set of equations, but are samples from the above density. In the next section, we study infinite width of this density and find that we recover the same result that we derived previously in the other direction.

## F.2 SADDLE POINT EQUATIONS ON INFINITE DEPTH ACTION

This section considers the large $N$ limit of the infinite depth stochastic process. In particular, we evaluate the saddle point of $\mathcal{S}_\infty$ and show that it recovers the same limit as we obtained by first taking a saddle point and then taking the large $L$ limit. This shows that the limits commute.

$$\frac{\delta\mathcal{S}_\infty}{\delta\hat{\Phi}(\tau)} = \Phi(\tau) - \langle\phi(h(\tau))\phi(h(\tau))\rangle = 0$$

$$\frac{\delta\mathcal{S}_\infty}{\delta\hat{G}(\tau)} = G(\tau) - \langle g(\tau)g(\tau)\rangle = 0$$

$$\frac{\delta\mathcal{S}_\infty}{\delta A(\tau)} = -\eta_0^2\gamma_0^2 B(\tau) - i\eta_0\gamma_0 \left\langle\hat{h}(\tau)g(\tau)\right\rangle = 0$$

$$\frac{\delta\mathcal{S}_\infty}{\delta B(\tau)} = -\eta_0^2\gamma_0^2 A(\tau) - i\eta_0\gamma_0 \left\langle\hat{g}(\tau)\dot{\phi}(h(\tau))\phi(h(\tau))\right\rangle = 0 \tag{79}$$

For the last two expressions, we recognizing that

$$\frac{\delta}{\delta du(\tau')} \exp\left(-i \int \hat{h}(\tau)du(\tau)\right) = -i\hat{h}(\tau) \exp\left(-i \int \hat{h}(\tau)du(\tau)\right)$$

$$\frac{\delta}{\delta dr(\tau')} \exp\left(-i \int \hat{g}(\tau)\dot{\phi}(h(\tau))dr(\tau)\right) = -i\hat{g}(\tau)\dot{\phi}(h(\tau)) \exp\left(-i \int \hat{g}(\tau)\dot{\phi}(h(\tau))dr(\tau)\right) \tag{80}$$

We can thus express the saddle point equations in terms of the left hand side derivatives with respect to Gaussian sources. Performing integration by parts, we arrive at the usual response function formulas

$$A(\tau) = \eta_0^{-1}\gamma_0^{-1} \left\langle\frac{\delta\phi(h(\tau))}{\delta dr(\tau))}\right\rangle \;,\; B(\tau) = \eta_0^{-1}\gamma_0^{-1} \left\langle\frac{\delta g(\tau)}{\delta du(\tau)}\right\rangle \tag{81}$$

Lastly, we can integrate over $\hat{h}(\tau)$ and $\hat{g}(\tau)$ which give Dirac-Delta functions that define the $h$ and $g$ stochastic processes

$$dh(\tau) = du(\tau) + \eta_0\gamma_0 C_h(\tau)g(\tau) \;,\; dg(\tau) = -dr(\tau)\dot{\phi}(h(\tau)) - \eta_0\gamma_0\dot{\phi}(h(\tau))C_g(\tau)\phi(h(\tau))d\tau \tag{82}$$

The rest of the indices (over samples and training times) can be incorporated into the above argument and the result is easily verified to match that derived in Appendix E from the large width then large depth limit. This exercise thus verifies that the large width and large depth limit continue to commute throughout training, extending the commutativity at initialization proven by Hayou & Yang (2023).

## G  EXACTLY SOLVEABLE CASES

In this section, we give exact solutions to the dynamics which do not require any non-Gaussian integrals. The two cases where this can be achieved are

1. the lazy limit $\gamma_0 \to 0$ of network training where the features are frozen

2. networks with linear activations, which preserve Gaussianity of $h, g$ at any $\gamma_0$.

We discuss these two limits in great detail below.

### G.1  LAZY LIMIT

In the lazy limit $\gamma_0 \to 0$, the flow for the preactivations $h(\tau, \boldsymbol{x}, t) = h(\tau, \boldsymbol{x})$ and the gradient fields $g^\ell(\tau, \boldsymbol{x}, t) = g^\ell(\tau, \boldsymbol{x})$ are constant over training time. Similarly, the kernels $\Phi^\ell, G^\ell$ are also constant throughout the dynamics. We thus have the following forward and backward dynamics

$$h(\tau, \boldsymbol{x}) = h(0, \boldsymbol{x}) + \int_0^\tau du(\tau', \boldsymbol{x}) \tag{83}$$

$$g(\tau, \boldsymbol{x}) = g(1, \boldsymbol{x}) + \int_\tau^1 \dot{\phi}(h(\tau', \boldsymbol{x})) dr(\tau', \boldsymbol{x}) \tag{84}$$

We note that the above equations indicate that $h(\tau, \boldsymbol{x})$ are all jointly Gaussian with zero mean. Since $h$ is Gaussian, it makes sense to characterize its covariance $H(\tau, \boldsymbol{x}, \boldsymbol{x}') = \langle h(\tau, \boldsymbol{x}) h(\tau, \boldsymbol{x}') \rangle$ at layer-time $\tau$. This object satisfies the differential equation

$$\begin{aligned}
\frac{\partial}{\partial \tau} H(\tau, \boldsymbol{x}, \boldsymbol{x}') &= \Phi(\tau, \boldsymbol{x}, \boldsymbol{x}') \\
\Phi(\tau, \boldsymbol{x}, \boldsymbol{x}') &= \langle \phi(h(\tau, \boldsymbol{x})) \phi(h(\tau, \boldsymbol{x}')) \rangle \\
h(\tau, \boldsymbol{x}, \boldsymbol{x}') &\sim \mathcal{N}(0, \boldsymbol{H}(\tau))
\end{aligned} \tag{85}$$

Since $h(\tau, \boldsymbol{x})$ are zero-mean jointly Gaussian random variables, the average to compute $\Phi$ can be computed in closed form for polynomials and several practically used nonlinearities (ReLU, ERF, etc) (Cho & Saul, 2009). Now, to compute the backward pass, we must

$$\frac{\partial}{\partial \tau} G(\tau, \boldsymbol{x}, \boldsymbol{x}') = -\left\langle \dot{\phi}(h(\tau, \boldsymbol{x})) \dot{\phi}(h(\tau', \boldsymbol{x}')) \right\rangle G(\tau, \boldsymbol{x}, \boldsymbol{x}') \tag{86}$$

Again, the bivariate Gaussian integral $\left\langle \dot{\phi}(h(\tau, \boldsymbol{x})) \dot{\phi}(h(\tau', \boldsymbol{x}')) \right\rangle$ has a closed form expression for many regularly used activation functions $\phi$. Given that we have solved for $\Phi(\tau)$ and $G(\tau)$, the dynamics of the predictor is

$$\frac{d}{dt} f(\boldsymbol{x}, t) = \eta_0 \mathbb{E}_{\boldsymbol{x}'} K(\boldsymbol{x}, \boldsymbol{x}') \Delta(\boldsymbol{x}', t) \ , \ \Delta(\boldsymbol{x}, t) = -\frac{\partial \mathcal{L}}{\partial f(\boldsymbol{x}, t)}$$

$$K(\boldsymbol{x}, \boldsymbol{x}') = \int_0^1 d\tau \ G(\tau, \boldsymbol{x}, \boldsymbol{x}') \Phi(\tau, \boldsymbol{x}, \boldsymbol{x}'). \tag{87}$$

This is an exact description of the network predictor dynamics in the $N \to \infty, L \to \infty$ and $\gamma_0 \to 0$ limit. However, the $\gamma_0 \to 0$ limit lacks a fundamental phenomenon of deep neural networks, namely feature learning. To gain analytical insight into the feature learning dynamics, we will study linear networks in the next section.

### G.2  INFINITE DEPTH RESNETS WITH LINEAR ACTIVATIONS

In the case where the activations are linear, we can also close the equations at the level of the kernels (since activations remain Gaussian even after feature learning). To provide an exactly solveable model of the dynamics, we consider computing the kernels after one step of feature learning.

### G.2.1 FULL DMFT EQUATIONS IN DEEP LINEAR NETWORKS

The DMFT equations for the preactivation processes have the form

$$h(\tau, \boldsymbol{x}, t) = h(0, \boldsymbol{x}, t) + \int_0^\tau du(\tau', \boldsymbol{x}, t) + \eta_0 \gamma_0 \int_0^\tau d\tau' \int d\boldsymbol{x}' \int ds \, C_h(\tau', \boldsymbol{x}, \boldsymbol{x}', t, s) g(\tau', \boldsymbol{x}', s)$$

$$g(\tau, \boldsymbol{x}, t) = g(1, \boldsymbol{x}, t) + \int_\tau^1 dr(\tau', \boldsymbol{x}, t) + \eta_0 \gamma_0 \int_\tau^1 d\tau' \int d\boldsymbol{x}' \int ds \, C_g(\tau', \boldsymbol{x}, \boldsymbol{x}', t, s) h(\tau', \boldsymbol{x}', s)$$

$$(88)$$

where $C_h, C_g$ have the usual expressions. The field derivatives (to get the response functions) have the form

$$\frac{\delta h(\tau, \boldsymbol{x}, t)}{\delta du(\tau', \boldsymbol{x}', s)} = \Theta(\tau - \tau')\delta(\boldsymbol{x} - \boldsymbol{x}')\delta(t - s)$$

$$+ \eta_0 \gamma_0 \int_0^\tau d\tau'' \int dx'' \int dt' C_h(\tau'', \boldsymbol{x}, \boldsymbol{x}'', t, t') \frac{\delta g(\tau'', \boldsymbol{x}'', t')}{\delta du(\tau', \boldsymbol{x}', s)}$$

$$\frac{\delta g(\tau, \boldsymbol{x}, t)}{\delta du(\tau', \boldsymbol{x}', s)} = \eta_0 \gamma_0 \int_\tau^1 d\tau'' \int dx'' \int dt' C_g(\tau'', \boldsymbol{x}, \boldsymbol{x}', t, t') \frac{\delta h(\tau'', \boldsymbol{x}'', t')}{\delta du(\tau', \boldsymbol{x}', s)}$$

$$\frac{\delta h(\tau, \boldsymbol{x}, t)}{\delta dr(\tau', \boldsymbol{x}', s)} = \eta_0 \gamma_0 \int_0^\tau d\tau'' \int dx'' \int dt' C_h(\tau'', \boldsymbol{x}, \boldsymbol{x}', t, t') \frac{\delta g(\tau'', \boldsymbol{x}'', t')}{\delta dr(\tau', \boldsymbol{x}', s)}$$

$$\frac{\delta g(\tau, \boldsymbol{x}, t)}{\delta dr(\tau', \boldsymbol{x}', s)} = \Theta(\tau' - \tau)\delta(\boldsymbol{x} - \boldsymbol{x}')\delta(t - s)$$

$$+ \eta_0 \gamma_0 \int_\tau^1 d\tau'' \int dx'' \int dt' C_g(\tau'', \boldsymbol{x}, \boldsymbol{x}', t, t') \frac{\delta h(\tau'', \boldsymbol{x}'', t')}{\delta dr(\tau', \boldsymbol{x}', s)} \qquad (89)$$

Since the activations are linear, these derivatives are no longer stochastic and we have the equal layer time $\tau = \tau'$ derivatives as the response functions

$$A(\tau, \boldsymbol{x}, \boldsymbol{x}', t, s) = \frac{\delta h(\tau, \boldsymbol{x}, t)}{\delta dr(\tau, \boldsymbol{x}', s)} \; , \; B(\tau, \boldsymbol{x}, \boldsymbol{x}', t, s) = \frac{\delta g(\tau, \boldsymbol{x}, t)}{\delta du(\tau, \boldsymbol{x}', s)} \qquad (90)$$

Next, we compute the kernels as simple Gaussian averages

$$H(\tau, \boldsymbol{x}, \boldsymbol{x}', t, s) = \langle h(\tau, \boldsymbol{x}, t)h(\tau, \boldsymbol{x}, t) \rangle \; , \; G(\tau, \boldsymbol{x}, \boldsymbol{x}', t, s) = \langle g(\tau, \boldsymbol{x}, t)g(\tau, \boldsymbol{x}, t) \rangle \qquad (91)$$

In the next section, we show a simple example where these equations become simple linear differential equations.

### G.2.2 ONE STEP ONE SAMPLE FEATURE KERNEL EQUATIONS

Before training, the initial conditions for the kernels $H_0(\tau)$ and $G_0(\tau)$ are

$$\partial_\tau H_0(\tau) = H_0(\tau) \; , \; \partial_\tau G_0(\tau) = -G_0(\tau)$$

$$\implies H_0(\tau) = e^\tau \; , \; G_0(\tau) = e^{-\tau} + 1 - e^{-1} \qquad (92)$$

After a step of gradient descent, the preactivation at layer time $\tau$ has the form

$$h(\tau) = h(0) + \int_0^\tau du(\tau') + \eta_0 \gamma_0 \int_0^\tau d\tau'[A(\tau') + H_0(\tau')y]g_0(\tau') \qquad (93)$$

We let $H(\tau), G(\tau)$ represent the kernels after one step. The response function after one step has satisfies the differential equation

$$\partial_\tau A(\tau) = A(\tau) + H_0(\tau)y \; , \; A(0) = 0 \implies A(\tau) = \tau e^\tau \, y \qquad (94)$$

The kernel $H(\tau)$ after one step therefore satisfies

$$H(\tau) = H(0) + \int_0^\tau d\tau' H(\tau')$$

$$+ \eta_0^2 \gamma_0^2 \int_0^\tau \int_0^\tau d\tau' d\tau'' G_0(\tau', \tau'')[A(\tau') + H_0(\tau')y][A(\tau'') + H_0(\tau'')y]$$

$$= H(0) + \int_0^\tau d\tau' H(\tau') + \eta_0^2 \gamma_0^2 y^2 \int_0^\tau \int_0^\tau d\tau' d\tau'' G_0(\tau', \tau'') e^{\tau' + \tau''}[\tau' + 1][\tau'' + 1]. \quad (95)$$

where $G_0(\tau, \tau') = \langle g_0(\tau) g_0(\tau') \rangle$. Differentiating both sides with respect to $\tau$ gives the integro-differential equation

$$\partial_\tau H(\tau) = H(\tau) + 2\eta_0^2 \gamma_0^2 y^2 e^\tau [\tau + 1] \int_0^\tau d\tau' G_0(\tau, \tau') e^{\tau'} [\tau' + 1] \tag{96}$$

Note that $G_0(\tau, \tau') = G_0(\tau) = e^{-\tau} + 1 - e^{-1}$ for $\tau' < \tau$. So we can pull this out of the integral and we are left with

$$\int_0^\tau d\tau' e^{\tau'} [\tau' + 1] = \tau e^\tau \tag{97}$$

We are finally left with the ODE

$$\partial_\tau H(\tau) = H(\tau) + 2\eta_0^2 \gamma_0^2 y^2 \ e^{2\tau} [e^{-\tau} + 1 - e^{-1}][\tau^2 + \tau] \tag{98}$$

Integrating gives

$$H(\tau) = e^\tau H(0) + 2\eta_0^2 \gamma_0^2 e^\tau \left[ \frac{1}{3}\tau^3 + \frac{1}{2}\tau^2 + (1 + e^{-1})(\tau^2 e^\tau - \tau e^\tau + e^\tau - 1) \right] \tag{99}$$

This solution gives the profile of the feature kernels after a single step of gradient descent. This is verified numerically in Figure 7. At small $\tau$ this goes as $H(\tau) - H_0(\tau) \sim \eta_0^2 \gamma_0^2 [1 + (1 + e^{-1})]\tau^2$.

## H   FINITE WIDTH CONVERGENCE: FLUCTUATIONS DO NOT SCALE WITH $L$

At fixed $L$, one can compute systematic asymptotic corrections in $\mathcal{O}_N(N^{-1})$ to the statistical properties of $\boldsymbol{q}$. This follows closely from the finite width analysis around mean field dynamics performed by Bordelon & Pehlevan (2023). The idea of their expansion is to extract from $\mathcal{S}$ information about finite size deviations from the infinite width DMFT process. The key fact which appears in the next to leading order terms is that at finite width $N$, the order parameters $\boldsymbol{q}$ become random with covariance

$$\text{Cov}(\boldsymbol{q}) \sim \frac{1}{N} \boldsymbol{\Sigma}^L + \mathcal{O}_N(N^{-2}) \ , \ \boldsymbol{\Sigma}^L = \left[ -\nabla^2 \mathcal{S}(\boldsymbol{q}_\infty) \right]^{-1} \tag{100}$$

The matrix $\frac{1}{N} \boldsymbol{\Sigma}^L$ gives the leading order (in powers of $1/N$) covariance of the order parameters over random initialization at width $N$ depth $L$. The goal of this section is to show that the entries of this covariance matrix $\boldsymbol{\Sigma}^L$ that correspond to the NTK and (consequently the predictor $f$) do not diverge with $L$ for the architecture of this paper (residual networks with $1/\sqrt{L}$ branch scaling), but rather approach a well defined limit at rate $\mathcal{O}(L^{-1})$. We again stress that this is unique to this architecture. Non-residual deep networks networks have propagator entries which scale linearly with $L$ since variances accumulate rather than average out over layers (Bordelon & Pehlevan, 2022a; 2023). We note that the predictor $f$ can be included in the set of order parameters $\boldsymbol{q}$ at depth $L$, so that the finite width deviation scales as

$$\langle (f_{N,L} - f_{\infty,L})^2 \rangle \sim \frac{1}{N} \Sigma_f^L \tag{101}$$

where averages are taken over random initialization.

### H.1   COMPONENTS OF THE HESSIAN

Since we are primarily interested in the scaling of the finite width effects with respect to $N$ and $L$, we will suppress the sample and time indices of the kernels and instead focus on the dependence of the fluctuations on layer index $\ell$. This will not change the analysis of the scaling but will save us from tracking a tremendous amount of indices. We will let $\Phi^\ell$ to represent the feature kernel at layer $\ell$ and $G^\ell$ the gradient kernel at layer $\ell$ with no time or sample indices. Under this simplified notation, the Hessian has entries of the form

$$\frac{\partial^2 \mathcal{S}}{\partial \hat{\Phi}^\ell \partial \hat{\Phi}^{\ell'}} = \left\langle \phi(h^\ell) \phi(h^\ell) \phi(h^{\ell'}) \phi(h^{\ell'}) \right\rangle - \Phi^\ell \Phi^{\ell'} \equiv \kappa_\Phi^{\ell, \ell'}$$

$$\frac{\partial^2 \mathcal{S}}{\partial \hat{\Phi}^\ell \partial \Phi^{\ell'}} = \delta_{\ell, \ell'} - \frac{\partial}{\partial \Phi^{\ell'}} \left\langle \phi(h^\ell) \phi(h^\ell) \right\rangle \equiv \delta_{\ell, \ell'} - \frac{1}{L} D_\Phi^{\ell, \ell'}$$

$$\frac{\partial^2 \mathcal{S}}{\partial \Phi^\ell \partial \Phi^{\ell'}} = 0 \tag{102}$$

First, we note that since the $h^\ell$ fields have *long-range correlations*, the $\kappa^{\ell,\ell'}$ is non-vanishing for all $\ell, \ell'$. We next note that $D_\Phi^{\ell,\ell'} \sim \mathcal{O}(1)$ since $\frac{\partial h^\ell}{\partial \Phi^{\ell'}} \sim \mathcal{O}(L^{-1})$ and by Price's theorem,

$$D_\Phi^{\ell,\ell'} = \frac{1}{2} \left\langle \frac{\partial^2}{\partial u^{\ell'} \partial u^{\ell'}} [\phi(h^\ell)\phi(h^\ell)] \right\rangle + 2L \left\langle \dot{\phi}(h^\ell)\phi(h^\ell) \frac{\partial h^\ell}{\partial \Phi^{\ell'}} \right\rangle \sim \mathcal{O}(1) \tag{103}$$

Similarly, we also need to calculate the terms involving $G^\ell$

$$\frac{\partial^2 \mathcal{S}}{\partial \hat{G}^\ell \partial \hat{G}^{\ell'}} = \left\langle g^\ell g^\ell g^{\ell'} g^{\ell'} \right\rangle - G^\ell G^{\ell'} \equiv \kappa_G^{\ell,\ell'}$$

$$\frac{\partial^2 \mathcal{S}}{\partial \hat{G}^\ell \partial G^{\ell'}} = \delta_{\ell,\ell'} - \frac{\partial}{\partial G^{\ell'}} \left\langle g^\ell g^\ell \right\rangle \equiv \delta_{\ell,\ell'} - \frac{1}{L} D_G^{\ell,\ell'}$$

$$\frac{\partial^2 \mathcal{S}}{\partial G^\ell \partial G^{\ell'}} = 0 \tag{104}$$

Lastly, we must consider the cross terms involving both $\{\Phi^\ell\}$ and $\{G^\ell\}$

$$\frac{\partial^2 \mathcal{S}}{\partial \hat{\Phi}^\ell \partial \hat{G}^{\ell'}} = \left\langle \phi(h^\ell)\phi(h^\ell) g^{\ell'} g^{\ell'} \right\rangle - \Phi^\ell G^{\ell'} \equiv \kappa_{\Phi,G}^{\ell,\ell'}$$

$$\frac{\partial^2 \mathcal{S}}{\partial \hat{\Phi}^\ell \partial G^{\ell'}} = -\frac{\partial}{\partial G^{\ell'}} \left\langle \phi(h^\ell)\phi(h^\ell) \right\rangle \equiv -\frac{1}{L} D_{\Phi,G}^{\ell,\ell'}$$

$$\frac{\partial^2 \mathcal{S}}{\partial \hat{G}^\ell \partial \Phi^{\ell'}} = -\frac{\partial}{\partial \Phi^{\ell'}} \left\langle g^\ell g^\ell \right\rangle \equiv -\frac{1}{L} D_{G,\Phi}^{\ell,\ell'}$$

$$\frac{\partial^2 \mathcal{S}}{\partial \Phi^\ell \partial G^{\ell'}} = 0 \tag{105}$$

The $\kappa$ tensor represents an instantaneous source of variance for the kernels, while the $D$ tensor represents the sensitivity of the kernel at layer $\ell$ to a perturbation in the kernel at layer $\ell'$. All entries of each $D$ matrix are $\mathcal{O}(1)$. Now, to compute the propagator, we construct block matrices

$$\kappa = \begin{bmatrix} \kappa_\Phi & \kappa_{\Phi,G} \\ \kappa_{G,\Phi} & \kappa_G \end{bmatrix} \, , \, D = \begin{bmatrix} D_\Phi & D_{\Phi,G} \\ D_{G,\Phi} & D_G \end{bmatrix} \tag{106}$$

Now, to obtain the propagator $\Sigma$, we must compute the inverse of the Hessian $\nabla^2 \mathcal{S}$

$$\Sigma = - \begin{bmatrix} \mathbf{0} & (I - \frac{1}{L}D)^\top \\ I - \frac{1}{L}D & \kappa \end{bmatrix}^{-1} = \begin{bmatrix} (I - \frac{1}{L}D)^{-1} \kappa \left[ (I - \frac{1}{L}D)^{-1} \right]^\top & -(I - \frac{1}{L}D)^{-1} \\ - \left[ (I - \frac{1}{L}D)^{-1} \right]^\top & \mathbf{0} \end{bmatrix} \tag{107}$$

The observables of interest are the top-left block which gives the joint covariance over all $\{\Phi^\ell, G^\ell\}$ variables. We introduce the following $2 \times 2$ block matrices

$$\kappa^{\ell,\ell'} = \begin{bmatrix} \kappa_\Phi^{\ell,\ell'} & \kappa_{\Phi G}^{\ell,\ell'} \\ \kappa_{G\Phi}^{\ell,\ell'} & \kappa_G^{\ell,\ell'} \end{bmatrix} \, , \, \Sigma^{\ell,\ell'} = \begin{bmatrix} \Sigma_\Phi^{\ell,\ell'} & \Sigma_{\Phi G}^{\ell,\ell'} \\ \Sigma_{G\Phi}^{\ell,\ell'} & \Sigma_G^{\ell,\ell'} \end{bmatrix} \, , \, D^{\ell,\ell'} = \begin{bmatrix} D_\Phi^{\ell,\ell'} & D_{\Phi G}^{\ell,\ell'} \\ D_{G\Phi}^{\ell,\ell'} & D_G^{\ell,\ell'} \end{bmatrix} \tag{108}$$

The above equation gives us we get the following $2 \times 2$ matrix equations

$$\Sigma^{\ell,\ell'} - \frac{1}{L} \sum_k D^{\ell,k} \Sigma^{k,\ell'} - \frac{1}{L} \sum_k \Sigma^{\ell,k} [D^{\ell',k}]^\top + \frac{1}{L^2} \sum_{kk'} D^{\ell,k} \Sigma^{k,k'} [D^{k',\ell'}]^\top = \kappa^{\ell,\ell'} \tag{109}$$

We see that the solutions $\Sigma^{\ell,\ell'}$ to the above equation will be $\mathcal{O}_L(1)$. This is precisely due to the factors of $1/L$ which appear in the above sums. One way to see this is to consider a large $L \gg 1$ limit where the above layer sums converge to integrals over $\tau$

$$\Sigma(\tau, \tau') - \int d\tau'' D(\tau, \tau'') \Sigma(\tau'', \tau') - \int d\tau'' \Sigma(\tau'', \tau') D(\tau, \tau'')^\top$$

$$+ \int d\tau'' d\tau''' D(\tau, \tau'') \Sigma(\tau'', \tau''') D(\tau', \tau''')^\top = \kappa(\tau, \tau') \tag{110}$$

So we find that the covariance of kernels at each pair of layers $\Sigma(\tau, \tau')$ is $\mathcal{O}_L(1)$.

## H.2 Variance of the NTK and Predictor

Using the above fact that $\Sigma^{\ell,\ell'} \sim \mathcal{O}_L(1)$, we can reason about the variance of the neural tangent kernel $K$

$$\text{Var}(K) = \frac{1}{NL^2} \sum_{\ell\ell'} \Sigma_\Phi^{\ell,\ell'} G^{\ell+1} G^{\ell'+1} + \frac{1}{NL^2} \sum_{\ell\ell'} \Sigma_G^{\ell,\ell'} \Phi^{\ell-1} \Phi^{\ell'-1}$$
$$+ \frac{2}{NL^2} \sum_{\ell\ell'} \Sigma_{\Phi G}^{\ell,\ell'} G^{\ell+1} \Phi^{\ell'-1} \sim \mathcal{O}_{L,N}(N^{-1}). \tag{111}$$

This demonstrates that the NTK will have initialization variance that scales only with the width $N$ but not the depth $L$. Using the fact that $\frac{\partial f}{\partial t} = K\Delta$, we see that $f$ will inherit fluctuations on the same scale of $\mathcal{O}_{N,L}(N^{-1/2})$. The depth $L$ NTK and predictor covariance converge to the infinite depth propagator with rate $\mathcal{O}(L^{-1})$.

# I   Finite Width And Depth Error Analysis

In this section, we combine the results of the previous sections. We consider the following error decomposition where we introduce the norm $||z|| = \sqrt{\langle z^2 \rangle}$ and $\langle\rangle$ denotes an average over random initializations of the network. Applying the triangle inequality on this norm, we have

$$\begin{aligned} ||f_{N,L} - f_{\infty,\infty}|| &\leq ||f_{N,L} - \langle f_{N,L}\rangle|| && \text{(Finite Width Variance)} \\ &+ ||\langle f_{N,L}\rangle - f_{\infty,L}|| && \text{(Mean Predictor Finite Width Error)} \\ &+ ||f_{\infty,L} - f_{\infty,\infty}|| && \text{(Finite Depth Error at Infinite Width)} \end{aligned} \tag{112}$$

From the preeceeding section H, the first term has asymptotic behavior

$$\left\langle (f_{N,L} - \langle f_{N,L}\rangle)^2 \right\rangle \sim \frac{1}{N}\Sigma_f^L + \mathcal{O}(N^{-2}) = \frac{1}{N}\Sigma_f^\infty + \mathcal{O}\left(\frac{1}{NL} + \frac{1}{N^2}\right) \tag{113}$$

where the last term comes from the fact that $\Sigma^L$ approximates the infinite depth propagator $\Sigma^\infty$ with error $1/L$. Next, we can apply results developed from Bordelon & Pehlevan (2023) which show that the mean predictor satisfies $\langle f_{N,L}\rangle = f_{\infty,L} + \mathcal{O}(N^{-1})$ to find that mean finite width error scales as

$$|\langle f_{N,L}\rangle - f_{\infty,L}|^2 \sim \mathcal{O}(N^{-2}) \tag{114}$$

Lastly, the finite depth error in the infinite width DMFT arises from the $1/L$ discretization effect. Thus the square error goes as

$$|f_{\infty,L} - f_{\infty,\infty}|^2 \sim \mathcal{O}(L^{-2}). \tag{115}$$

which follows from the discretization error of the limiting large depth process for the NTK and thus the predictor. Altogether, we have

$$||f_{N,L} - f_{\infty,\infty}|| \sim \sqrt{\frac{1}{N}\Sigma_f^\infty + \mathcal{O}\left(\frac{1}{NL}\right)} + \mathcal{O}\left(\frac{1}{N}\right) + \mathcal{O}\left(\frac{1}{L}\right) \tag{116}$$

We see that at large width and depth this is dominated by the first term which comes from the finite width fluctuations. However, if the model is ensembled, then we expect a faster rate of convergence from the last two sources of error. Indeed, we provide an experiment verifying that ensembling improves the rate of convergence in Figure B.7.

# J   Why Parameterization Influences Hyperparameter Transfer at Finite Widths and Depths

One may wonder why other parameterizations which also admit a scaling limit (such as Neural Tangent Parameterization) do not transfer as well as networks in $\mu$P or why our proposal transfers over depth but others do not. To achieve successful transfer across finite widths, it is insufficient to reason about the infinite width/depth scaling limit, rather one needs to identify a parameterization

that *minimizes the approximation error between finite networks and the scaling limit* (of course subject to the obvious constraints of training in finite time, learning features at the desired rate etc).

Alternative parameterizations that do not keep feature updates constant, can be thought of as choosing a function $\gamma_0(N, L)$ which describes how much the features evolve with $N, L$. Suppose further that this has a limiting value $\lim_{N,L\to\infty} \gamma_0(N, L) = \gamma_0^\star$. For example, neural tangent parameterization in our $\frac{1}{\sqrt{L}}$ ResNet corresponds to $\gamma_0 = \frac{1}{\sqrt{N}}$ with $\gamma_0^\star = 0$. We let the logits for a finite width/depth model in this parameterization be $f_{N,L}^{\gamma_0(N,L)}$ represent a finite width $N$ and depth $L$ network predictions with feature learning scale $\gamma_0(N, L)$. The approximation error this $(N, L)$ network and another $(N', L')$ network has two potential sources of error

$$||f_{N,L}^{\gamma_0(N,L)} - f_{N',L'}^{\gamma_0(N',L')}|| \leq ||f_{N,L}^{\gamma_0(N,L)} - f_{\infty,\infty}^{\gamma_0(N,L)}|| + ||f_{N',L'}^{\gamma_0(N,L)} - f_{\infty,\infty}^{\gamma_0(N',L')}||$$
$$+ ||f_{\infty,\infty}^{\gamma_0(N,L)} - f_{\infty,\infty}^{\gamma_0(N',L')}|| \tag{117}$$

Heuristically, one can eliminate the final approximation error by choosing a parameterization where $\gamma_0(N, L) = \gamma_0$ is a width/depth-independent constant. Then one is left with finite approximation errors to an infinite model with the same rate of feature learning.

# K  ARCHITECTURAL + OPTIMIZATION EXTENSIONS

In this section, we explore various extensions of the model written in the main text. Most of these extensions do not modify the scaling rule or the procedural ideas which we used to characterize the limit. We will explore convolutions, multiple layers per block, training read-in and read-out matrices, discrete time training dynamics (finite step size SGD), momentum, and LayerNorm.

## K.1  CONVOLUTIONAL MODELS

A DMFT for convolutional ResNets can be easily obtained by introducing spatial coordinates $\mathfrak{a}, \mathfrak{b}$ of each layer's representation (Bordelon & Pehlevan, 2022b). The preactivation vectors $\boldsymbol{h}_{\mathfrak{a}}^{\ell}(\boldsymbol{x}, t) \in \mathbb{R}^N$ represent the activity of all $N$ channels in layer $\ell$ at spatial position $\mathfrak{a}$. The trainable weights take the form $\boldsymbol{W}_{\mathfrak{a}}^{\ell} \in \mathbb{R}^{N \times N}$ describe the filter at a spatial displacement $\mathfrak{a}$ from its center. The recursion of the ResNet is

$$\boldsymbol{h}_{\mathfrak{a}}^{\ell+1}(\boldsymbol{x}, t) = \boldsymbol{h}_{\mathfrak{a}}^{\ell}(\boldsymbol{x}, t) + \frac{1}{\sqrt{NL}} \sum_{\mathfrak{b}} \boldsymbol{W}_{\mathfrak{b}}^{\ell} \phi(\boldsymbol{h}_{\mathfrak{a}+\mathfrak{b}}^{\ell}(\boldsymbol{x}, t)) \tag{118}$$

where the sum over $\mathfrak{b}$ runs over all valid spatial positions in the filter. The relevant order parameters are still feature-feature correlations, but now with additional spatial indices

$$\Phi_{\mathfrak{a},\mathfrak{b}}^{\ell}(\boldsymbol{x}, \boldsymbol{x}', t, s) = \frac{1}{N} \phi(\boldsymbol{h}_{\mathfrak{a}}^{\ell}(\boldsymbol{x}, t)) \cdot \phi(\boldsymbol{h}_{\mathfrak{b}}^{\ell}(\boldsymbol{x}', s)) \tag{119}$$

A similar gradient-gradient kernel is also crucial $G_{\mathfrak{a},\mathfrak{b}}^{\ell}(\boldsymbol{x}, \boldsymbol{x}', t, s)$. From these objects, one can construct a limiting DMFT at $N \to \infty$

## K.2  MULTIPLE LAYERS PER RESIDUAL BLOCK

Many residual neural networks, including the popular ResNet-18 and ResNet-50 models, there are more than a single convolution layer on a residual branch. In this section, we show that this does not alter the scaling rule or the convergence to a well defined large width and depth limit provided the number of branches $L$ goes to infinity with number of layers per block $K$ fixed. We therefore work with the case of $K \sim \mathcal{O}(1)$ hidden layers on each branch and still take the branch number $L$

to infinity. The recurrence of interest is

$$f = \frac{1}{\gamma_0 N} \boldsymbol{w}^L \cdot \phi(\boldsymbol{h}^L)$$

$$\boldsymbol{h}^{\ell+1} = \boldsymbol{h}^\ell + \frac{1}{\sqrt{L}} \tilde{\boldsymbol{h}}^{\ell,K} , \ \ell \in \{1, ..., L-1\}$$

$$\tilde{\boldsymbol{h}}^{\ell,k+1} = \frac{1}{\sqrt{N}} \boldsymbol{W}^{\ell,k} \phi(\tilde{\boldsymbol{h}}^{\ell,k}) , \ k \in \{1, ..., K-1\}$$

$$\tilde{\boldsymbol{h}}^{\ell,1} = \frac{1}{\sqrt{N}} \boldsymbol{W}^{\ell,0} \phi(\boldsymbol{h}^\ell)$$

$$\boldsymbol{h}^1 = \frac{1}{\sqrt{D}} \boldsymbol{W}^0 \boldsymbol{x} \tag{120}$$

The goal is to verify that gradient based learning on $\{\boldsymbol{W}^{\ell,k}\}_{\ell\in[L],k\in[K]}$ gives a well defined infinite width and depth limit $N \to \infty$ and $L \to \infty$. The weight dynamics are

$$\frac{d}{dt} \boldsymbol{W}^{\ell,k} = \frac{\eta_0 \gamma_0}{\sqrt{NL}} \mathbb{E}_{\boldsymbol{x}} \Delta(\boldsymbol{x}, t) \tilde{\boldsymbol{g}}^{\ell,k+1}(\boldsymbol{x}, t) \phi\left(\tilde{\boldsymbol{h}}^{\ell,k}(\boldsymbol{x}, t)\right)^\top , \ k \in \{1, ..., K-1\}$$

$$\frac{d}{dt} \boldsymbol{W}^{\ell,K} = \frac{\eta_0 \gamma_0}{\sqrt{NL}} \mathbb{E}_{\boldsymbol{x}} \Delta(\boldsymbol{x}, t) \boldsymbol{g}^{\ell+1}(\boldsymbol{x}, t) \phi\left(\tilde{\boldsymbol{h}}^{\ell,k}(\boldsymbol{x}, t)\right)^\top \tag{121}$$

where we have defined the gradient fields

$$\tilde{\boldsymbol{g}}^{\ell,k}(\boldsymbol{x}, t) = N\gamma_0 \sqrt{L} \frac{\partial f(\boldsymbol{x}, t)}{\partial \tilde{\boldsymbol{h}}^{\ell,k}(\boldsymbol{x}, t)} \sim \mathcal{O}_{N,L}(1)$$

$$\boldsymbol{g}^\ell(\boldsymbol{x}, t) = N\gamma_0 \frac{\partial f(\boldsymbol{x}, t)}{\partial \boldsymbol{h}^\ell(\boldsymbol{x}, t)} \sim \mathcal{O}_{N,L}(1) \tag{122}$$

We next examine the scale of the updates to preactivation features $\tilde{\boldsymbol{h}}^{\ell,k}$ and $\boldsymbol{h}^\ell$

$$\tilde{\boldsymbol{h}}^{\ell,k+1}(\boldsymbol{x}, t) = \frac{1}{\sqrt{N}} \boldsymbol{W}^{\ell,k}(0) \phi(\tilde{\boldsymbol{h}}^{\ell,k}) + \frac{\eta_0 \gamma_0}{\sqrt{L}} \mathbb{E}_{\boldsymbol{x}'} \int_0^t ds \, \Delta(\boldsymbol{x}', s) \Phi^{\ell,k}(\boldsymbol{x}, \boldsymbol{x}'; t, s) \tilde{\boldsymbol{g}}^{\ell,k+1}(\boldsymbol{x}', s)$$

$$\boldsymbol{h}^{\ell+1}(\boldsymbol{x}, t) = \boldsymbol{h}^\ell(\boldsymbol{x}, t) + \frac{1}{\sqrt{LN}} \boldsymbol{W}^{\ell,K}(0) \phi(\tilde{\boldsymbol{h}}^{\ell,K-1})$$

$$+ \frac{\eta_0 \gamma_0}{L} \mathbb{E}_{\boldsymbol{x}'} \int_0^t ds \, \Delta(\boldsymbol{x}', s) \Phi^{\ell,K}(\boldsymbol{x}, \boldsymbol{x}', t, s) \boldsymbol{g}^{\ell+1}(\boldsymbol{x}', s) \tag{123}$$

We see that the $\tilde{\boldsymbol{h}}^{\ell,k}$ features will be uncorrelated across $k$ since there are no skip connections within a block. However, the $\boldsymbol{h}^\ell$ exhibit long range correlations as before. To understand the cumulative impact of the feature learning corrections to the preactivations, we must reason about how correlated the gradient signals $\boldsymbol{g}^\ell$ and $\tilde{\boldsymbol{g}}^{\ell,k}$ are across different layers. We

$$\tilde{\boldsymbol{g}}^{\ell,k}(\boldsymbol{x}, t) = \frac{1}{\sqrt{N}} \dot{\phi}(\tilde{\boldsymbol{h}}^{\ell,k}(\boldsymbol{x}, t)) \odot \boldsymbol{W}^{\ell,k}(t)^\top \tilde{\boldsymbol{g}}^{\ell,k+1}(\boldsymbol{x}, t)$$

$$\boldsymbol{g}^\ell(\boldsymbol{x}, t) = \boldsymbol{g}^{\ell+1}(\boldsymbol{x}, t) + \frac{1}{\sqrt{NL}} \dot{\phi}(\boldsymbol{h}^\ell(\boldsymbol{x}, t)) \odot \left[\boldsymbol{W}^{\ell,0}(t)^\top \tilde{\boldsymbol{g}}^{\ell,1}(\boldsymbol{x}, t)\right] \tag{124}$$

We see that the skip connections keep long range correlations in the $\boldsymbol{g}^\ell$ signals, however, the $\tilde{\boldsymbol{g}}^{\ell,k}$ are uncorrelated with one another, as in a standard feedforward network.

### K.2.1 THE LIMITING BEHAVIOR OF MULTI-LAYER PER BLOCK MODEL

Based on the correlation structures, of the $\boldsymbol{g}^\ell$ and $\tilde{\boldsymbol{g}}^{\ell,k}$ gradient featuers, we have the following DMFT equations at infinite width and large depth

$$h^{\ell+1}(\boldsymbol{x}, t) = h^1(\boldsymbol{x}, t) + \frac{1}{\sqrt{L}} \sum_{\ell'=1}^{\ell+1} u^{\ell'}(\boldsymbol{x}, t)$$

$$+ \frac{\eta_0 \gamma_0}{L} \sum_{\ell'=1}^{\ell+1} \mathbb{E}_{\boldsymbol{x}'} \int_0^t ds \, \Delta(\boldsymbol{x}', s) \Phi^{\ell'-1,K}(\boldsymbol{x}, \boldsymbol{x}', t, s) g^{\ell'}(\boldsymbol{x}', s) + \mathcal{O}(L^{-1/2}) \tag{125}$$

where the Gaussian source has the form

$$u^{\ell+1}(\boldsymbol{x}, t) \sim \mathcal{GP}(0, \boldsymbol{\Phi}^{\ell,K}) \tag{126}$$

where $\Phi^{\ell,K}$ is determined through the *lazy within-block recursion*

$$\Phi^{k,\ell}(\boldsymbol{x}, \boldsymbol{x}', t, s) = \left\langle \phi(u^{\ell,k}(\boldsymbol{x}, t))\phi(u^{\ell,k}(\boldsymbol{x}, t))\right\rangle_{u^{\ell,k} \sim \mathcal{N}(0, \boldsymbol{\Phi}^{\ell,k-1})} , \ k \in \{1, ..., K-1\}. \tag{127}$$

Though all of the kernels are dynamical, it is as if the weights $\boldsymbol{W}^{\ell,k}$ are static for $k \in \{1, ..., K-1\}$. This is a consequence of the decorrelation of $\tilde{g}^{\ell,k}$ across $k$. We leave as a future problem whether it is possible to reparameterize the intermediate weights so that each of the internal block weight's dynamics move by $\mathcal{O}_L(1)$ and contribute additional feature learning terms in the dynamics.

## K.3  EFFECT OF TRAINING READ-IN WEIGHTS

If in addition to training the weights of the ResNet "body" we also train the readout and readin weights, they will also contribute to the total NTK

$$K(\boldsymbol{x}, \boldsymbol{x}') = \frac{1}{L}\sum_{\ell=1}^{L} G^{\ell+1}(\boldsymbol{x}, \boldsymbol{x}')\Phi^{\ell}(\boldsymbol{x}, \boldsymbol{x}') + G^1(\boldsymbol{x}, \boldsymbol{x}')K^x(\boldsymbol{x}, \boldsymbol{x}'). \tag{128}$$

where $K^x(\boldsymbol{x}, \boldsymbol{x}') = \frac{1}{D}\boldsymbol{x} \cdot \boldsymbol{x}'$ is the input kernel. We see that the addition of these two layers does not change the fact that the NTK is $\mathcal{O}_L(1)$ in this parameterization. Now, we need to verify that the first layer weights move by the appropriate scale

$$\frac{d}{dt}W_{ij}^0(t) = \frac{\eta_0\gamma_0}{\sqrt{D}}\mathbb{E}_{\boldsymbol{x}}\Delta(\boldsymbol{x}, t)g_i^1(\boldsymbol{x}, t)x_j \sim \mathcal{O}_{L,N}(1) \tag{129}$$

These equations lead to the following base case equations for the fields $g^L(\boldsymbol{x}, t)$ and $h^1(\boldsymbol{x}, t)$

$$h^1(\boldsymbol{x}, t) = u^1(\boldsymbol{x}, t) + \eta_0\gamma_0\mathbb{E}_{\boldsymbol{x}'}\int_0^t ds\Delta(\boldsymbol{x}', s)g^1(\boldsymbol{x}', s)K^x(\boldsymbol{x}, \boldsymbol{x}')$$

$$g^L(\boldsymbol{x}, t) = \dot{\phi}(h^L(\boldsymbol{x}, t))r^L(\boldsymbol{x}, t) + \eta_0\gamma_0\dot{\phi}(h^L(\boldsymbol{x}, t))\mathbb{E}_{\boldsymbol{x}'}\int_0^t ds \ \Phi^L(\boldsymbol{x}, \boldsymbol{x}', t, s)\Delta(\boldsymbol{x}', s) \tag{130}$$

These fields will therefore evolve by $\mathcal{O}_{N,L}(1)$ in this parameterization.

## K.4  DISCRETE TIME UPDATES

DMFT for feedforward networks can also be modified for discrete time (Bordelon & Pehlevan, 2022b; 2023). A straightforward extension of these results will give discrete time dynamics for the field dynamics $h^\ell, g^\ell$ for our ResNet. The one subtlety about discrete time is that the dynamical neural tangent kernel (NTK) no longer governs the evolution of the predictor $f$. Instead,

$$f(\boldsymbol{x}, t) = \eta_0\sum_{s<t}\mathbb{E}_{\boldsymbol{x}' \sim \mathfrak{B}_s}\Delta(\boldsymbol{x}', s)\Phi^L(\boldsymbol{x}, \boldsymbol{x}', t, s) + \eta_0\int d\boldsymbol{x}'\sum_{s<t}A^L(\boldsymbol{x}, \boldsymbol{x}', t, s), \tag{131}$$

where $\mathfrak{B}_s$ is the minibatch at timestep $s$ and $A^L(\boldsymbol{x}, \boldsymbol{x}', t, s) = \frac{\sqrt{L}}{\eta_0\gamma_0}\left\langle \frac{\delta\phi(h^L(\boldsymbol{x}, t))}{\delta r^L(\boldsymbol{x}', s)}\right\rangle$ is the final layer response function at depth $L$. These are now evaluated for $t, s$ integer rather than real numbers. In the infinite width and depth limit $N, L \to \infty$, we have the following discrete time update equations for the preactivations

$$h(\tau, \boldsymbol{x}, t) = \int_0^\tau du(\tau', \boldsymbol{x}, t) + \eta_0\gamma_0\int_0^\tau \int d\boldsymbol{x}'\sum_{s<t}C_h(\tau', \boldsymbol{x}, \boldsymbol{x}', t, s)g(\tau', \boldsymbol{x}', s)$$

$$C_h(\tau, \boldsymbol{x}, \boldsymbol{x}', t, s) = A(\tau, \boldsymbol{x}, \boldsymbol{x}', t, s) + \frac{1}{|\mathfrak{B}_s|}\sum_{\boldsymbol{x}_s \in \mathfrak{B}_s}\delta(\boldsymbol{x}' - \boldsymbol{x}_s)\Delta(\boldsymbol{x}', s)\Phi(\tau, \boldsymbol{x}, \boldsymbol{x}', t, s) \tag{132}$$

Notice that this is almost identical to the dynamics for gradient flow except that the integrals over time become sums over discrete time and the population density $p(\boldsymbol{x})$ is approximated at each step by a uniform density on minibatch $\mathfrak{B}_t$. An analogous discrete time equation holds for $g(\tau)$.

### K.5 MOMENTUM

Momentum can be easily handled within our framework. Let $\boldsymbol{\theta} = \text{Vec}\{\boldsymbol{W}^0, \boldsymbol{W}^1, ..., \boldsymbol{W}^{L-1}, \boldsymbol{w}^L\}$ represent the concatenation of all trainable network parameters. The momentum update is controlled by a momentum parameter $\alpha$ which controls the exponential moving average of gradients

$$\boldsymbol{\theta}(t+1) = \boldsymbol{\theta}(t) + \eta_0 \gamma_0 \boldsymbol{v}(t)$$
$$\boldsymbol{v}(t) = \alpha \, \boldsymbol{v}(t-1) - (1-\alpha) N \gamma_0 \nabla_{\boldsymbol{\theta}} \mathbb{E}_{\boldsymbol{x}} \mathcal{L}[f(\boldsymbol{x}, \boldsymbol{\theta}(t))]. \tag{133}$$

Writing this out for a particular hidden layer yields

$$\boldsymbol{W}^\ell(t+1) = \boldsymbol{W}^\ell(t) + \eta_0 \gamma_0 \boldsymbol{V}^\ell(t)$$
$$\boldsymbol{V}^\ell(t) = \alpha \boldsymbol{V}^\ell(t-1) + (1-\alpha) \frac{1}{\sqrt{LN}} \mathbb{E}_{\boldsymbol{x}} \Delta(\boldsymbol{x}, t) \boldsymbol{g}^{\ell+1}(\boldsymbol{x}, t) \phi(\boldsymbol{h}^\ell(\boldsymbol{x}, t))^\top \tag{134}$$

We can unfold the recurrence for $\boldsymbol{V}^\ell(t)$ to get the expression

$$\boldsymbol{V}^\ell(t) = \frac{(1-\alpha)}{\sqrt{LN}} \sum_{s=0}^t \alpha^{t-s} \, \mathbb{E}_{\boldsymbol{x}} \Delta(\boldsymbol{x}, s) \boldsymbol{g}^{\ell+1}(\boldsymbol{x}, s) \phi(\boldsymbol{h}^\ell(\boldsymbol{x}, s))^\top \tag{135}$$

Plugging this into the update equations for $\boldsymbol{W}^\ell(t)$ we get

$$\boldsymbol{W}^\ell(t+1) = \boldsymbol{W}^\ell(t) + \frac{\eta_0 \gamma_0 (1-\alpha)}{\sqrt{LN}} \sum_{s=0}^t \alpha^{t-s} \, \mathbb{E}_{\boldsymbol{x}} \Delta(\boldsymbol{x}, s) \boldsymbol{g}^{\ell+1}(\boldsymbol{x}, s) \phi(\boldsymbol{h}^\ell(\boldsymbol{x}, s))^\top \tag{136}$$

From this equation, we can again unfold the recurrence for $\boldsymbol{W}^\ell(t)$ and write the recurrence equations for $\boldsymbol{h}^\ell, \boldsymbol{g}^\ell$, verifying that we get feature learning updates of the correct scale

$$\boldsymbol{h}^{\ell+1}(\boldsymbol{x}, t) = \boldsymbol{h}^\ell(\boldsymbol{x}, t) + \frac{1}{\sqrt{L}} \boldsymbol{\chi}^{\ell+1}(\boldsymbol{x}, t)$$
$$+ \frac{\eta_0 \gamma_0 (1-\alpha)}{L} \sum_{t' < t} \sum_{s < t'} \alpha^{t'-s} \mathbb{E}_{\boldsymbol{x}'} \Phi^\ell(\boldsymbol{x}, \boldsymbol{x}', t, s) \Delta(\boldsymbol{x}', s) \boldsymbol{g}^{\ell+1}(\boldsymbol{x}', s) \tag{137}$$

where $\boldsymbol{\chi}^{\ell+1}(\boldsymbol{x}, t) = \frac{1}{\sqrt{N}} \boldsymbol{W}^\ell(0) \phi(\boldsymbol{h}^\ell(\boldsymbol{x}, t))$. We see that this update has the correct scaling structure, the random (uncorrelated across layers) field $\boldsymbol{\chi}$ has a scale of $\frac{1}{\sqrt{L}}$ and will behave as a Brownian motion and the feature learning update with the gradient field (strongly correlated across layers) has the correct $\frac{1}{L}$ scaling. The normal DMFT procedure can now be carried out on the $\boldsymbol{\chi}$ fields and the corresponding $\boldsymbol{\xi}$ fields for the backward pass.

## L WEIGHT DECAY

Weight decay dynamics can also be analyzed with DMFT. The gradient flow dynamics with weight decay with parameter $\lambda$ have the form

$$\frac{d}{dt} \boldsymbol{W}^\ell(t) = \frac{\eta_0 \gamma_0}{\sqrt{LN}} \, \mathbb{E}_{\boldsymbol{x}} \, \Delta(\boldsymbol{x}, t) \boldsymbol{g}^{\ell+1}(\boldsymbol{x}, t) \phi(\boldsymbol{h}^\ell(\boldsymbol{x}, t))^\top - \eta_0 \lambda \boldsymbol{W}^\ell(t) \tag{138}$$

The dependence of $\boldsymbol{W}^\ell(t)$ on the initial condition can be isolated with a simple integrating factor

$$\boldsymbol{W}^\ell(t) = e^{-\eta_0 \lambda t} \boldsymbol{W}^\ell(0) + \frac{\eta_0 \gamma_0}{\sqrt{NL}} \int ds \, e^{-\eta_0 \lambda (t-s)} \mathbb{E}_{\boldsymbol{x}} \, \Delta(\boldsymbol{x}, s) \boldsymbol{g}^{\ell+1}(\boldsymbol{x}, s) \phi(\boldsymbol{h}^\ell(\boldsymbol{x}, s))^\top \tag{139}$$

Computing the forward pass recursion we find

$$\boldsymbol{h}^{\ell+1}(\boldsymbol{x}, t) = \boldsymbol{h}^\ell(\boldsymbol{x}, t) + \frac{1}{\sqrt{L}} e^{-\eta_0 \lambda t} \boldsymbol{\chi}^\ell(\boldsymbol{x}, t)$$
$$+ \frac{\eta_0 \gamma_0}{L} \int ds \, e^{-\eta_0 \lambda (t-s)} \mathbb{E}_{\boldsymbol{x}'} \, \Delta(\boldsymbol{x}', s) \boldsymbol{g}^{\ell+1}(\boldsymbol{x}', s) \Phi^\ell(\boldsymbol{x}, \boldsymbol{x}', t, s) \tag{140}$$

This clearly will admit an infinite width and depth limit following the arguments in Appendices D & E. The difference is the presence of factors of $e^{-\eta_0 \lambda t}$, which suppress the influence of the initial condition at late time.

# M  OTHER LARGE DEPTH LIMITS

It is possible to construct other large depth limits of residual networks. In general we can consider

$$\boldsymbol{h}^{\ell+1} = \boldsymbol{h}^\ell + \beta_0 L^{-\alpha} N^{-1/2} \boldsymbol{W}^\ell \phi(\boldsymbol{h}^\ell) \tag{141}$$

Well-behaved signal propagation at initialization and throughout feature learning are preserved for any $\alpha \geq 1/2$ provided the learning rate is scaled as $\eta = \eta_0 \gamma_0^2 L^{2\alpha-1} N$. This ensures that $\frac{d}{dt}\boldsymbol{h}^\ell = \mathcal{O}_{N,L}(1)$ and $\frac{d}{dt}f = \mathcal{O}(1)$ in the large width and depth limit $N, L \to \infty$. The DMFT approach can also be used to analyze these limits, though we leave this to future work.

We show an example of $\alpha \in \{1, \frac{1}{2}\}$ parameterizations in Figure M.1. In that model, we train ResNets with $K = 4$ layers per residual block and start with zero initialization of the readouts from each block so that the Brownian motion terms vanish for both parameterizations. This is similar to the Res-Zero initialization (Bachlechner et al., 2021), however we note that even with zero initialization of the branch readout, the parameterization (the depth-scaling exponent $\alpha$) still makes a difference in the dynamics. We find that both parameterizations show approximate transfer (runs diverge for same learning rates at large enough depth). However, the ODE scaling $\alpha = 1$ shows monotone improvement with depth $L$ while the SDE scaling $\alpha = \frac{1}{2}$ shows the opposite at early time.

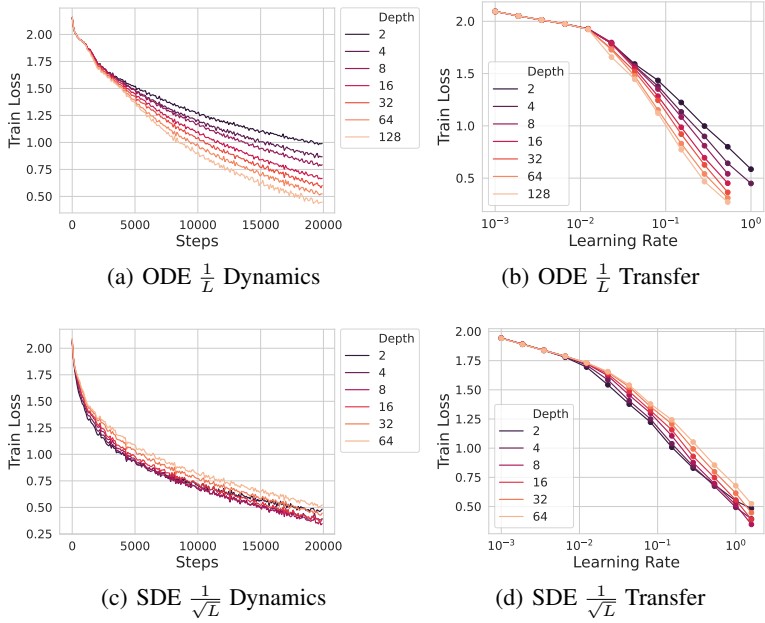

Figure M.1: Hyperparameters transfer for both SDE and ODE scalings for multi-layer-per-block model $K = 4$ models. Depth $L$ is varied for both models. Increasing $L$ leads to monotone improvements in the ODE but not SDE when initializing with zero readout weights for each block. In (d) all runs diverge at the next biggest learning rate.

A more in-depth characterization of the differences in transferability and convergence rates of models in each of these parameterizations will be left to future work.

