# OpenReview forum: "Depthwise Hyperparameter Transfer in Residual Networks: Dynamics and Scaling Limit"
_ICLR.cc/2024/Conference — ICLR 2024 poster_

### Official Review · Reviewer_J7oQ · 2023-10-17

**Soundness:** 3 good
**Presentation:** 3 good
**Contribution:** 3 good
**Rating:** 8
**Confidence:** 4

**Summary:**

The authors study hyperparameter transfer between models at different scales with an emphasis on the hyperparameter transfer of ResNets (or ResNets sub-components) with depth. Following parts of the rational behind muP parameterization, they argue that good parameterizations are those having stable infinite width/depth limits. They then consider one such recent limit ("1/\sqrt{depth}" scaling) and extend prior theoretical results on ResNets at initialization to ResNets trained with gradient flow. Specifically, they establish that network updates are also stable at this limit. The manuscript contains several experimental results on CIFAR-10,Tiny ImageNet, and ImageNet showing successful hyper-parameter transfer based on this "1/\sqrt{depth}" scaling. As an additional theoretical side, the DMFT equations are solved for an infinite widh&depth linear network acting on a single data point.

**Strengths:**

The work offers a good combination of theory and experiment.

It expands a useful and timely avenue of research, hyperparameter transfer, where theory can actually aid practitioners.

It studies relevant complex architecture acting on realistic datasets as opposed to linear or shallow networks.

**Weaknesses:**

Several gaps exist between the stated goal of the theory and its practical implementation. One such gap, which the authors themselves cite, is that the theory predicts stable results following a scale-up. Naturally, however, spending more compute and getting equal performance, is of little practical value. In practice, one expects to get better results which seems to be the case at least for the larger datasets they consider. This means that in examples where HP-transfer is worthwhile, one is by definition well away from the scaling limit.

Another gap, of a similar kind, is that, as far as I could tell, their DMFT framework is based on gradient flow but it is then used to transfer the learning rate. While gradient flow may be a good approximation for SGD in the asymptotically low learning rate regime, typically the optimal learning rates lay far away from this regime.

For CIFAR-10, the 1/\sqrt{depth} scaling leads to poorer performance compared to muP scaling.

Reference to current literature could be broadened and sharpened. For instance, muP parameterization is also a unique parameterization having maximal updates, not just a stable limit (This fuller characterization also partially escapes the first gap described above). Citing further alternatives to muP scaling and adaptive kernel approaches to feature learning is also desirable.

**Questions:**

1. Can the authors rationalize why the above two gaps could be excused?

2. Can the authors explain why for CIFAR-10 their parameterization seems non-optimal?

---

> ### Author Response · Authors · 2023-11-13
>
> We thank the reviewer for their careful reading and valuable feedback, to which we respond below.
>
> ### **Weakness 1 = Gap 1**
>
> Our article neither seeks nor claims to reduce the computational cost of training a large model with a fixed setting of hyperparameters. The core contribution of our work is a methodology for significantly recuding the cost of hyperparameter optimization itself.
>
> Our claim was that this could be achieved by using our parameterization to optimize hyperparameters in much smaller models. We indeed found in all of our experiments that optimal hyperparameters settings were stable over many orders of magnitude with our procedure. To validate this empirically, however, it was necessary to train rather large models.
>
> We certainly agree that it is desirable to obtain a target model accuracy while minimizing compute. However, our experiments consistently showed that larger/deeper models result in better performance. This is consistent with the literature on scaling laws and with the common practice of training very large models to obtain best SOTA performace. That said, the marginal gain from using larger models is sometimes very small (especially when very close to the limiting dynamics) and perhaps not worth the computational cost. An understanding of when this happens is not something we know how to predict.
>
>
> ### **Weakness 2 = Gap 2**
>
> The reviewer is entirely correct that we've presented in the main text the DMFT equations using optimization with gradient flow, rather than discrete time (S)GD. However, as in the prior work on DMFT (Bordelon & Pehlevan, 2022), discrete time SGD also leads to a consistent scaling limit for training dynamics and accompanying DMFT equations. These are included in the Appendix (now App. K.4). The discrete time dynamics of $h, g$ are almost identical to their continuous time counterparts but with sums instead of integrals over time. The dynamics for the predictor is no longer governed by the dynamical NTK but rather is given by the last layer feature kernel and the final layer response function (see equation 131).
>
> ### **Weakness 3 = CIFAR10 Performance**
>
> The reviewer is right that in Figure 1 the best learning rates found in our proposed parameterization reach somewhat higher values of training loss than the best learning rates we found with a vanilla $\mu P$ parameterization. Thank you for suggesting that we address this issue directly.
>
> We have done so in our revision by emphasizing in the introduction that the discrepancy in training loss is actually predicted by our theory. Indeed, note that the $\mu P$ runs diverged at large depth (that is why the orange line corresponding to depth 30 doesn't appear at all in Figure 1(a)). Our theory predicts that increasing depth at a fixed learning rate in the $\mu P$ parameterization corresponds to an effectively higher learning rate. Thus, since our experiments were run for a constant but moderate number of epochs, $\mu P$ runs result in models that are closer to convergence. This is true until the effective learning rate is so large that the runs diverge.
>
> We also note that Figure 1 seeks to illustrate hyperparameter transfer, which clearly fails for $\mu P$ but works well with our parameterization. In particular, we didn't attempt to obtain the best possible test loss for each architecture, which would have requred optimizing other hyperparameters (e.g. notably the lr schedule, how to do data augmention, momentum, batch size etc). In this sense, we do not believe that the discrepancy in test loss in Figure 1 takes away from the merit of our core proposal.
>
> We hope that this addresses the reviewer's concern and welcome any further feedback on this point. Also, we are in the process of running several followup experiments to illustrate the first bullet point and will post then once they have finished running.
>
>
> ### **Weakness 4 = Literature**
>
> We thank the reviewer for their valuable suggestions to improve our coverage of the literature and have made the following improvements in the revision:
> * We have added Appendix C outlining the original work on $\mu P$ and have included a discussion of maximal vs. stable updates as well as alternative parameterizations.
> * We have included a new Appendix section (App. M) discussing other scalings which give stable feature learning infinite depth limits. We also provide more experiments showing hyperparameter transfer for zero-init block readouts (similar to rezero) and $\text{depth}^{-\alpha}$ scalings for residual branches with $\alpha=\frac{1}{2},1$. The $\alpha=1$ case gives a Neural ODE like infinite depth limit rather than the SDE-like limit at $\alpha=1/2$ (which is the focus of the rest of the paper). We hope this shows that our general methods can be applicable to derive and characterize many large depth limits which can show transfer.
> * We have added to the Related Works section in our revision references to recent work on using adaptive kernel methods.

---

> ### Comment · Reviewer_J7oQ · 2023-11-19
> **Reply to reply**
>
> I thank the authors for addressing my comments.
>
> Regarding the authors' response to "Gap I", I think there may have been some misunderstanding worth clearing.
>
> My main focus in this gap is not on the matter of compute. Rather it is on the fact that, as I understand it, in the scaling limit the authors offer the DMFT equations become independent of depth and width. This seems to imply that performance should also be scale invariant. In contrast, however, in their experiments, where they employ this scaling the performance does change. For instance, in Fig. 1b and width=1024, train loss changes from about 0.3 to 0.2 as depth increases. Shouldn't this number be fixed if one is indeed in the asymptotic scaling regime? This seems to mean that in the typical regime of interest (a regime where increasing scale/compute improves performance) sizable corrections to the asymptotic still exist. These corrections could, in principle, scale differently with hyper-parameters thereby augmenting the optimal transfer of parameters based on keeping the asymptotic theory the same.
>
> Notwithstanding, while I'm curious about this I don't consider this a major drawback. Potentially what could be going on is that the asymptotic theory is stable/invariant under scaling in early training stages with corrections to the asymptotic entering in only at later times. If so one can base the logic here on the common practice of tuning parameters based on early epochs. This is roughly consistent with the authors' Supp. figure B3a where the difference between the original and scale-up network becomes larger only at later epochs.

---

> > ### Author Response · Authors · 2023-11-19
> > **Response to Follow up Questions**
> >
> > These are important questions and sorry if we misinterpreted the original critique. Thank you for allowing us a chance to try providing better explanations.
> >
> > 1. First, we are in complete agreement that if the models are *very close* to the DMFT limit (like in Figure 5a) then the performances should be very close. Thus the pure DMFT equations cannot explain benefits in performance to increasing width/depth or the consistency of optimal hyperparameters across finite widths/depths.
> > 2. To have an asymptotic theory which can explain how large the finite width/depth effects are at fixed hyperparameters, we need to go beyond the DMFT limit and we provide some of the tools to do this in Appendices H and I, which gives finite width/depth error rates that we verify empirically. The dominant error effect *near the DMFT limit* is that at finite width/depth, the order parameters (kernels, predictions, etc) are noisy (initialization dependent) versions of their infinite width analogue with a noise that depends on $N,L$. The mean dynamics of the outputs is dominated by the infinite width/depth version of the dynamics which have the same dependence on hyperparameters. The variance in output logits is larger for small width/depth networks, which causes their performance to be slightly worse. For examples of this type of theory explaining why "wider is better" near the limit with fixed hyperparameters, please see the recent work https://arxiv.org/abs/2304.03408 on which we base our finite size analysis. Indeed those authors find, as we do, that these finite size effects grow over training time (like in Supp B3).
> > 3. As the reviewer points out, it is theoretically possible that in the range of finite widths/depths that we consider that both optimal hyperparameters and performance shift with $N, L$ or that optimal learning rates would shift over time. The fact that these are stable at this range of $N,L$ and training times is fortunate but we have not provided a theoretical proof that that this would work at this scale, motivating our suite of experiments.
> >
> > We will add more comments on these points in the Appendix where we carefully explain how explaining transfer theoretically will require going beyond the DMFT limit and characterizing how finite width/depth effects influence performance and optimal hyperparameters. A goal of such a future analysis would be to explain why optimal HPs appear stable across scales while performances are not.

---

### Official Review · Reviewer_SFFE · 2023-10-22

**Soundness:** 3 good
**Presentation:** 3 good
**Contribution:** 3 good
**Rating:** 6
**Confidence:** 4

**Summary:**

The authors' contribution in this work lies in their introduction of a straightforward parameterization for residual networks. Through empirical observations, they have demonstrated the remarkable consistency of hyperparameter transfer across various dimensions of network architecture, such as width and depth, as well as across diverse hyperparameters and datasets. This empirical evidence is further substantiated by theoretical analysis, which reveals that the behaviors of hidden layers remain intricate, unaffected by changes in network dimensions, and do not approach vanishing limits.

**Strengths:**

This paper demonstrates that a straightforward adaptation of the $\mu P$  parameterization enables the transfer of learning rates across both depth and width in residual networks, incorporating $1/\sqrt{depth}$ scaling on the residual branches. The authors conduct extensive experiments. The theory is also nice.

I think these findings have the potential to significantly reduce the computational expenses associated with hyperparameter tuning, thereby enabling practitioners to train large, deep, and wide models just once while achieving near-optimal hyperparameters.

**Weaknesses:**

1. Since that the $\mu P$ parameterization is not a commonly used method, the authors should provide a more comprehensive introduction.

2. The $1/\sqrt{depth}$ trick has been extensively investigated in previous works [1-3]. The novelty of the proposed method appears to be limited. Please clarify the differences. Moreover, many recent works propose initializing ResNet using "rezero"-like methods, which achieve similar performance to the $1/\sqrt{depth}$ trick. The authors should provide some comparisons. In particular, I would like to see some discussions on the scaling parameter. Why use $1/\sqrt{depth}$? What would happen if the scaling parameter is set as $1/depth$ or 0?

3. As the authors admitted in the "Limitation" paragraph, most of the experiments are confined to 10-20 epochs. This limitation makes the empirical evidence less convincing. In fact, I don't understand the difficulty in training a ResNet with more epochs if one can train it with 10-20 epochs.

[1] Tarnowski W, Warchoł P, Jastrzȩbski S, et al. Dynamical isometry is achieved in residual networks in a universal way for any activation function. The 22nd International Conference on Artificial Intelligence and Statistics. PMLR, 2019: 2221-2230.

[2] Yang G, Schoenholz S. Mean field residual networks: On the edge of chaos. Advances in neural information processing systems, 2017, 30.

[3] Zhang H, Dauphin Y N, Ma T. Fixup initialization: Residual learning without normalization. arXiv preprint arXiv:1901.09321, 2019.

**Questions:**

Please see  weaknesses.

---

> ### Author Response · Authors · 2023-11-13
>
> We thank the reviewer for their careful reading and valuable feedback, to which we respond below. We hope in light of our responses the reviewer will consider raising their score.
>
> ### **Weakness 1**
>
> We agree that it makes sense to add a brief description of the $\mu P$ parameterization. Thank you for the suggestion. This is now done in the Appendix C and C.2.
>
> ### **Weakness 2**
>
> Regarding the $1/\sqrt{\mathrm{depth}}$ scaling of residual branches:
> * We agree that our work is certainly not the first to propose the $1/\sqrt{\text{depth}}$ scaling of residual branches and have emphasized this in our revised review of literature.
> * As far as we are aware, ours is the first work which shows that this is precisely the residual branch scaling that gives rise to training dynamics that are both consistent across depth/width and also are capable of feature learning.
> * We have included a new Appendix section (App. M) discussing other possible large depth limits which give stable feature learning infinite depth limits. We also provide more experiments showing hyperparameter transfer for zero-initialized block readouts (similar to the initialization of re-zero) and $\text{depth}^{-\alpha}$ scalings for residual branches with $\alpha = \frac{1}{2},1 $. See Figure M.1. The $\alpha=1$ case gives a Neural ODE like infinite depth limit rather than the SDE-like limit at $\alpha=1/2$ (which is the focus of the rest of the paper). We hope this shows that our general methods can be applicable to derive and characterize many large depth limits which can exhibit transfer. We note that even with the zero initialization trick, the depth exponent $\alpha$ of the model is still important in controlling the training dynamics, with some different behaviors for different values of $\alpha$. We plan to explore this in greater detail in future work.
>
> ### **Weakness 3**
>
> We are starting to run experiments where we train models for longer and still see good hyperparameter transfer and larger "deeper is better" effects. We plan on adding additional experiments where we train these models longer (see for instance some additional experiments here https://imgur.com/a/GA6qepR).
>
> In our original submission, we chose to train for 10 - 20 epochs in most of our experiments for these reasons:
> * Our experiments were rather computationally expensive, at least with the compute resources available to us because
>     * The focus of our work is hyperparameter transfer over both depth and width. We were thus bottlenecked by the computational cost of training our largest (deepest and widest) architectures, which in many experiments were Convolutional ResNets with depth 34, width 2048 and roughly 50M parameters.
>     * Each experiment involved re-training the same architecture over many -- often 10 or more -- different learning rates (or other hyperparameter settings)
> * We believe directly checking hyperparameter transfer after model convergence would likely require an entirely different suite of experiments, rather than simply training longer. Indeed, hyperparameter transfer for models after covergence is most interesting only after optimizing over not only the learning rate but also the learning rate schedule, the batch size, momentum coefficient, data augmention, etc. We have presented a fair amount of evidence indicating that in our proposed paramaterization all these hyperparameters individually transfer (see Figure 3 for learning rate schedules and normalization layer and Figure 4 for momentum and feature learning rate). But running large scans jointly over all these parameters was not computationally feasible.
>
> ### **Further Citations**
>
> We thank the reviewer for citations [1]-[3]. We have now added [2] to our list of references ([1] and [3] were already present). We have also included in our revised Related Works section the point that the main difference between [1]-[3] and our work is threefold:
> * our article derives DMFT equations that govern training dynamics, rather than only the behavior at initialization
> * our analysis is combined $\mu P$-type initialization in the final layer, whereas most of this prior work considered standard parameterization
> * we focus on empirical investigation of whether hyperparameters transfer in this parameterization, while prior work was mainly concerned with the stability of the forward and backward pass at initialization

---

> > ### Comment · Reviewer_SFFE · 2023-11-14
> > **Relpy to authors**
> >
> > Thank you for your answer, and most of my concerns have been addressed. I am inclined to accept this paper.

---

### Official Review · Reviewer_fkzv · 2023-11-01

**Soundness:** 3 good
**Presentation:** 3 good
**Contribution:** 2 fair
**Rating:** 8
**Confidence:** 4

**Summary:**

Paper proposes a novel parameterisation of deep residual networks that tackles width- and depth-dependent cost of hyper-parameter tuning, an issue, exacerbated by recent increase in SOTA models' sizes. A novel $1/L$ extension of $\mu P$ parameterization of residual networks is proposed. Paper argues that hyperparameter transfer is consistent across both width and depth for a variety of architectures, hyperparameters, and datasets.
The work is primarily empirical whose experimental evidence is supported by statistical physics and NTK theory. In particular, the paper advances a width-invariant feature updates present in $\mu P$ parameterisation and in the same spirit derives depth scaling to ensure feature updates invariance over depth using dynamical mean field theory (DMFT).

Exact solutions of the DMFT dynamics in the rich (i.e.non-kernel) regime for a simple deep linear network are provided. The suggested parameterisation is verified empirically by well selected range of experiments, mostly presented in appendices and  summarised in the Section 3. The exposition of the challenges, method and main results is well and accessibly written rounded by limitations and future directions.

**Strengths:**

+ Rounded, well written and easy to follow timely paper presenting valuable empirical evidence put into NTK theory context.
+ Future directions section proposes the method as suitable to study depth-width trade-offs while scaling neural networks since hyperparameter dependencies are consistent across widths/depths. In my opinion this can be very valuable addition to community even if the method does not turn out practically useful and adds positively to my overall rating.
+ Convincing experiments (with a caveat, that I hope will be addressed in the rebuttal, see Weaknesses, ad 1)) cover well selected range of settings and architectures

**Weaknesses:**

1. Fig. 1 Loss levels reached seem to be higher in case of proposed parameterization compared to alternatives. Could authors add some comments on the topic? Especially, is a proposed parameterization capable of reaching an optimal value of hyper parameters in scope, e.g., learning rate in practical settings? I rate paper 'accepted' conditioned on this issue will be alleviated in the camera-ready version.

2. Limited applicability(?) - Could authors argue otherwise? How could it be improved in the paper? Could authors elaborate on computational costs of proposed method vs. alternatives?

3. Paper heavily depends on $\mu P$ parameterisation properties derived in Yang & Hu (2021). In my view a short primer on $\mu P$ parameterisation (even in Appendix) would improve self-consistency and readability of the paper.

4. Proposition 2, Assumption $\gamma_0 \rightarrow 0$ renders ODE from Proposition 1 solvable due to time invariant kernels (and thus given by Gaussian initialization) and the same goes for the second example where linearity of network makes time invariance explicit. To what extend are conclusion transferable to any realistic non-linear scenario?

**Questions:**

See section Weaknesses.

---

> ### Author Response · Authors · 2023-11-13
>
> We thank the reviewer for their careful reading and valuable feedback, to which we respond below.
>
> ### **Weakness 1**
>
> The reviewer is certainly right that in Figure 1 the best learning rates found in our proposed parameterization reach somewhat higher values of training loss than the best learning rates we found with a vanilla $\mu P$ parameterization. Thank you for suggesting that we address this issue.
>
> We have done so in our revision by emphasizing in the introduction that the discrepancy in training loss is actually predicted by our theory. Indeed, note that $\mu P$ runs diverged at large depth (that is why the orange line corresponding to depth 30 doesn't appear at all in Figure 1(a)). Our theory predicts that increasing depth at a fixed learning rate in the $\mu P$ parameterization corresponds to an effectively higher learning rate (see explanation in global response). Thus, since our experiments were run for a constant but moderate number of epochs, $\mu P$ runs result in models that are closer to convergence. This is true until the effective learning rate is so large that the runs diverge. We add a comment about this near the top of page 2 when we reference Figure 1.
>
> We are performing new experiments where the parameterizations coincide in their dynamics at a large depth so that all SP/$\mu$P models in this range of depths are stable over the range of learning rates we consider. This can be achieved, for example by choosing a smaller $O(1)$ value for $\beta$ in the SP/$\mu$P experiments (or larger $\beta_0 = \beta \sqrt{L}$ for our proposed parameterization). In these experiments, we expect to see that the shallower SP/$\mu$P networks will train **slower** than their counterparts in our $\frac{1}{\sqrt L}$ parameterization, resulting in higher train loss. We will revise our sumbission and post them here if/when these expts finish during the rebuttal period.
>
> We also note that Figure 1 seeks to illustrate hyperparameter transfer, which clearly fails for $\mu P$ but works well with our parameterization. In particular, we didn't attempt to obtain the best possible test loss for each architecture, which would have requred optimizing other hyperparameters (e.g. notably the lr schedule, how to do data augmention, momentum, batch size etc). In this sense, we do not believe that the discrepancy in test loss in Figure 1 takes away from the merit of our core proposal.
>
> We hope that this addresses the reviewer's concern and welcome any further feedback on this point. Also, we are in the process of running several followup experiments to illustrate the first bullet point and will post then once they have finished running.
>
> ### **Weakness 2**
>
> While our theory was derived for a certain class of ResNets (Appendix K) and not fully theoretically analyzed for the case of transformers etc, we tried to verify experimentally the broad applicability of our proposed parameterization in at least the following ways:
> - Experiments on CIFAR10 with Wide ResNets. Here, while the dataset is not very complicated, we check that not only the learning rate but also the momentum coefficient and regularization strength transfer (e.g. Figure 4).
> - Experiments on both ImageNet and Tiny ImageNet with Vision Transformers. Here, we checked not only that the learning rate but indeed also the learning schedule transfers (e.g. Figure 3).
> - Empirical validation of our theory in linear MLPs and ResNets in which we have analytic formulas for the NTK.
> - In terms of computational savings we find, for instance, that already depth 6 and width 128, we can estimate optimal learning rates for models at depth 30 and width 1024 (Figure 1). This results in a cost savings of roughly 320x in each forward and backward pass used for hyperparameter tuning. In appendix B.2 (see Figure B.3), we saw that hyperparemeter turning after training ViTs on CIFAR-10 for only 3 epochs gives the same results as training for 9 epochs. So all together we get a computaitonal savings in this case of approximately 1000x. Moreover, we made no serious attempt to systematically investigate the minimal width and depth at which to perform hyperparameter turning.
>
> ### **Weakness 3**
>
> We agree that it makes sense to add a brief description of the $\mu P$ parameterization. Thank you for the suggestion. This is now done in the new Appendix C and C.2.
>
> ### **Weakness 4**
>
> This is an excellent question! The final Figure 7 which shows the linear network solution is actually in the feature learning regime $\gamma_0 > 0$. Figure 7b plots the *change in the kernels* after one step of feature learning for $\gamma_0 > 0$. The linear activations make analysis of feature learning much more tractable. As far as we are aware no one has been able to find explicit solutions for the evolution of the NTK and related feature and gradient kernels in nonlinear networks in regimes where $\gamma_0 > 0$. We hope to return to this in future work but for now have included this point in the discussion.

---

> ### Comment · Reviewer_fkzv · 2023-11-19
>
> Thanks to authors for their "fair enough" responses, that are acceptable if the gaps are properly addressed in Discussion and Appendices, as mentioned in rebuttals. Especially, comments on computational savings are encouraging. I have no further comments and think it's an interesting paper that should be accepted. Thank you.

---

### Official Review · Reviewer_vjdW · 2023-11-02

**Soundness:** 3 good
**Presentation:** 3 good
**Contribution:** 3 good
**Rating:** 8
**Confidence:** 3

**Summary:**

The paper presents a parameterization for Resnet neural networks which is claimed to enable the transfer of hyperparameters such as the learning rate over a large range of network widths and depths. The motivation for this work is to enable engineers to tune hyperpaprameters on small models and then apply them to large models, avoiding a costly fine tuning on large models.

The method extends the muP parameterization which, according to the authors, enables hyperparameter transfers across different widths but not different depths. The proposed parameterization consists of zero-mean unit-variance parameter initialization and scaling factors for the residual branches, output pre-activations and learning rate that depend on both the width and depth of the model, and no normalization layers.

The proposed parameterization is derived by theoretically analyzing the behavior of a (simplified) neural network in the limit of  infinite width and infinite depth, and choosing a parameterization which satisfy certain desiderata. Experiments on a small dataset CFAR-10 are used to support the analysis on realistic neural architectures.

**Strengths:**

- The method tackles an important practical problem: the efficient tuning of hyperparameters.
- Good presentation

**Weaknesses:**

- The experimental section is quite limited: only a single task and a very simple dataset are considered.

**Questions:**

N/A

---

> ### Author Response · Authors · 2023-11-13
>
> We thank the reviewer for their feedback and positive assessment of our work! We agree that additional experiments on larger dataset, larger models, and longer training times would be useful to explore the limits of hyperparameter transfer. However, we also wanted to respectfully point out that our original submission already included experiments on Tiny ImageNet and ImageNet (Figure 3). We plan on adding additional experiments where we train these models longer (see for instance some new experiments here where we train on Tiny-ImageNet for 100 epochs and CIFAR for 40 epochs https://imgur.com/a/GA6qepR). We plan to add these to the paper once we get the corresponding SP/$\mu$P baselines without the branch scalings.

---

### Author Response · Authors · 2023-11-13
**Global Response to Reviewers 1**

We thank the reviewers for their careful reading and their useful questions. Below we address some of the recurring points which were raised in multiple reviews. We also respond to each reviewer separately, addressing all of their concerns.

### **Why is Loss Lower in Fig 1 for SP/$\mu$P without branch scaling compared to our $1/\sqrt{L}$ scaling?**

Several reviewers asked why deeper models in SP/$\mu$P have lower loss at the same learning rate compared to models which have stable $\frac{1}{\sqrt L}$ parameterizations. This effect is actually predicted by our theory, which shows that increasing depth in the usual parameterizations ($\mu P$ or SP) has a similar effect to increasing the learning rate. At early time in $\mu$P/SP the function moves at a rate $f(t+1) - f(t)= \mathcal{O}(\eta L)$ while in our parameterization it moves as $f(t+1) - f(t)=\mathcal{O}(\eta)$. Thus, since we are training all $\mu$P/SP models for a fixed number of epochs, as depth increases the train loss at fixed learning rate will be lower as depth increases until the models diverge due to instability (such as the $\mu P$ models at depth 30 in Figure 1(a)). On the other hand, in our parameterization, the scale of feature updates is approximately depth-independent at fixed learning rate, giving stable and consistent optimal learning rates across depths.

We are currently performing new experiments where the parameterizations coincide in their dynamics at a large depth (rather than at the smallest depth) so that all SP/$\mu$P models in this range of depths are stable over the range of learning rates we consider. This can be achieved, for example by choosing a smaller $O(1)$ value for $\beta$ in the SP/$\mu$P experiments (or larger $\beta_0 = \beta \sqrt{L}$ for our proposed parameterization). In these experiments, we expect to see that the shallower SP/$\mu$P networks will train **slower** than their counterparts in our $\frac{1}{\sqrt L}$ parameterization, resulting in higher train loss. We will revise our sumbission and post them here if/when these expts finish during the rebuttal period.

### **Including a review of $\mu$P and width scalings would be useful**

We agree with this point and we added a new Appendix section (the new Appendix C) which outlines the difference between stable and unstable parameterizations and gives a brief explanation of why $\mu$P is *maximal* (in the sense that all layers contribute to feature learning updates).

### **References to Relevant Literature and other Scalings**

We have added additional references to other depth scaling studies (such as those that study signal propagation at initialization), other adaptive kernel approaches to deep learning theory including recursive kernel machines and Bayesian adaptive NNGP approaches, and perturbative finite width analyses to feature learning in the related work section.

To distinguish our work from prior studies on signal propagation we added the following to the related work section

*The present work differs in three ways. First, we derive a DMFT that captures training dynamics, rather only the behavior at init. Second, we use $\mu P$-type init in the final layer, whereas prior work mostly considered standard parameterization. Third, we provide ample on empirical evidence of hyperparameter transfer throughout training.*

On the question of other large depth limits, we have started experimenting with other residual branch scalings that also give a large depth limit. We added Appendix M which discusses alternative scalings, including the Neural ODE scaling $1/L$. We show in Figure M.1 that this type of model can also achieve transfer. More research needs to be done to compare/contrast these different scaling limits, which we leave to future work.

### **Can we Handle Discrete Time (S)GD?**

Our submission had an Appendix section (now Appendix K.4) which discusses how our theory can be easily adapted to discrete time SGD using the techniques of Bordelon & Pehlevan 2022. We expanded this section to write how the preactivation dynamics throughout training would look at infinite width+depth in discrete time.

---

> ### Author Response · Authors · 2023-11-13
> **Global Response Part 2**
>
> ### **How much utility/applicability does our proposed transfer scheme provide?**
>
> Reviewers probed the utility/applicability of our proposed transfer scheme in several ways:
> * computational savings and breadth of applicability (Reviewer fkzv)
> * hyperparameter transfer beyond 10 - 20 epochs of training (Reviwer SFFE)
> * utility of training large (deep + wide) models in which the marginal gain in train error was small (Reviewer J7oQ)
>
> We briefly address each of these below.
>
> #### **Computational Savings**
>
> We believe a core contribution of our work is a reliable way to find computational savings in hyperparameter optimization. We find, for instance that already depth 6 and width 128, we can estimate optimal learning rates for models at depth 30 and width 1024 (Figure 1). This results in a cost savings of roughly 320x in each forward and backward pass used for hyperparameter tuning. In appendix B.2 (see Figure B.3), we saw that hyperparemeter turning after training ViTs on CIFAR-10 for only 3 epochs gives the same results as training for 9 epochs. So all together we get a computaitonal savings in this case of approximately 1000x. Moreover, we made no serious attempt to systematically investigate the minimal width and depth at which to perform hyperparameter turning.
>
> #### **Hyperparameter transfer beyond 10 - 20 epochs**
>
> We have included in our revision experiments that check hyperparameter for models close to convergence (trained for 100's of epochs) and have found:
> * For ViTs trained on Tiny ImageNet the gap in performance can be much more substantial at large depth if we train for longer. See https://imgur.com/a/GA6qepR.
> * Also attached in the link above are CIFAR-10 runs for 40 epochs where more dramatic gaps in loss can be seen.
>
> Further, in our original submission, we chose to train for 10 - 20 epochs in most of our experiments for these reasons:
> * Our experiments were rather computationally expensive, at least with the compute resources available to us because
>     * The focus of our work is hyperparameter transfer over both depth and width. We were thus bottlenecked by the computational cost of training our largest (deepest and widest) architectures, which in many experiments were Convolutional ResNets with depth 34, width 2048 and roughly 50M parameters.
>     * Each experiment involved re-training the same architecture over many -- often 10 or more -- different learning rates (or other hyperparameter settings)
> * We believe directly checking hyperparameter transfer after model convergence would likely require an entirely different suite of experiments, rather than simply training longer. Indeed, hyperparameter transfer for models fully trained models is most interesting only after optimizing over not only the learning rate but also the learning rate schedule, the batch size, momentum coefficient, data augmention, etc jointly. We have presented a fair amount of evidence indicating that in our proposed paramaterization all these hyperparameters individually transfer (see Figure 3 for learning rate schedules and normalization layer and Figure 4 for momentum and feature learning rate). But running large scans jointly over all these parameters was not computationally feasible.
>
>
> #### **Utility of Training Deeper Models**
>
> We certainly agree that whether training a larger model is worth the computational cost depends on the task. Indeed, our experiments consistently showed that larger/deeper models result in better performance, but the marginal gain from using larger models was sometimes small. In part, this was because we were not trying to reach the best possible accuracy with a given architecture in two senses:
>
> * we did not optimize jointly all hyperparmeters (e.g. lr schedule, momentum, batch size, regularization, etc)
> * we typically only trained for 10 - 20 epochs in our experiment. (Since the submission deadline, we have run several experiments https://imgur.com/a/GA6qepR, which suggest that the gap in performance can be much more substantial at large depth if we train for longer. We plan to add more of these experiments as they finish.)
>
>
> We chose to set up our experiments in this way because the main focus of this paper is hyperparameter transfer, which we found could be achieved by using our parameterization to optimize hyperparameters in much smaller models. But to validate this empirically, regardless of the gains in model performance, required checking that in all of our experiments optimal hyperparameters settings were stable over many orders of magnitude in model size.

---

### Author Response · Authors · 2023-11-16
**Update to Global Response**

Here we provide an update to: **Why is Loss Lower in Fig 1 for SP/$\mu$P without branch scaling compared to our
 scaling?**

As explained in the first global response, the model under $1/\sqrt{L}$ parametrization has a feature movement of $\mathcal{O}(\eta)$ compared to $\mathcal{O}(L \eta)$ of the standard parametrization / $\mu$P, resulting in a slower training when SP is trainable. Hence, we chose to increase the feature learning by scaling the residual branches by a $\mathcal{O}(1)$ constant $\beta = 3$. This choice implies that the dynamics of our $1/\sqrt{L}$ parametrization are identical to a $9$ layers $\mu$P Resnet. As we show in Figure 1b of the revised version of the manuscript (see also here: https://imgur.com/a/Kg61xXI), this choice of $\beta$ allows for learning rate transfer while achieving trainability and lower training loss at larger depths.

We are currently running other experiments applying the same logic of upscaling feature learning with larger $\beta$ to the case of Batch Norm (this will result in updated Figure 2).

---

### Author Response · Authors · 2023-11-18
**Update to Global Response #2**

Dear reviewers,

We have performed more experiments to improve the raw performances in a realistic Resnet18-type architecture (with batch norm) at large depths (Figure 2). The revised version of the paper now contains this plot: https://imgur.com/riDmCbd. We performed this experiment to have further confirmation that:

1. Learning rate transfers extend to very large depths (we trained networks with more than 300 layers) under $1/\sqrt{L}$ scaling with batch norm. On the other hand, without the right scaling there is a significant and incremental shift of the optimal learning rate.

2. The proposed scaling shows consistent improvement with depth, outperforming standard and widely adopted Resnet-type architectures at large depths.

We hope that these additional experiments further convince the reviewers of the significance and novelty of our results.

---

### Meta-Review · Area_Chair_5c33 · 2023-12-03

**Metareview:**

The paper provides an extension of mu-parameterization for res-nets and vision transformers to be able to scale those architectures in width and depth, with the property that the optimal hyperparameters are constant across scales. The paper provides both empirical and theoretical results.

All reviewers are supportive of publication (6,8,8,8). I fully agree with the reviewers that the extension of mu-parameterization (from scaling in width to scaling in width and depth) is a valuable contribution. It is very interesting and valuable to develop parameterizations so that a network can be scaled without having to do again perform hyperparameter optimization.

**Justification For Why Not Higher Score:**

The contribution is valuable, but since the parameterization is relatively close to the existing mu-parameterization and since the empirical results are somewhat limited, I do not recommend an oral or spotlight.

**Justification For Why Not Lower Score:**

It is very valuable to be able to scale in width and depth while keeping hyperparameters constant; this paper is a step in the direction of enabling that.

---

### Decision · Program_Chairs · 2024-01-16

Accept (poster)